

# Estimates of $CO_2$ fluxes over the City of Cape Town, South Africa, through Bayesian inverse modelling

Alecia Nickless[1,2], Peter J. Rayner[3], Francois Engelbrecht[4], Ernst-Günther Brunke[5], Birgit Erni[1], and Robert J. Scholes[6]

[1]Department of Statistical Sciences, University of Cape Town, Cape Town, 7701, South Africa
[2]Nuffield Department of Primary Care Health Sciences, University of Oxford, Oxford, OX2 6GG, UK
[3]School of Earth Sciences, University of Melbourne, Melbourne, VIC 3010, Australia
[4]Climate Studies and Modelling and Environmental Health, CSIR, Pretoria, 0005, South Africa
[5]South African Weather Service c/o CSIR, P.O. Box 320, Stellenbosch, 7599, South Africa
[6]Global Change Institute, University of the Witwatersrand, Johannesburg, 2050, South Africa
*Correspondence to:* Alecia Nickless alecia.nickless@phc.ox.ac.uk

**Abstract.** The results of a high resolution Bayesian inversion over the City of Cape Town, South Africa, are presented, which used observations of atmospheric $CO_2$ from sites at Robben Island and Hangklip lighthouses collected over a sixteen month period from March 2012 until June 2013. A Lagrangian particle dispersion model driven by the regional climate model Conformal Cubic Atmospheric Model (CCAM) was used to provide the sensitivities of the observations to the surface sources

and boundary concentrations. This regional climate model was dynamically coupled to the CABLE (Community Atmosphere Biosphere Land Exchange) model, which provided prior estimates of the biogenic fluxes. Prior estimates of the fossil fuel emissions were obtained from an inventory analysis specifically carried out for this inversion exercise, making use of vehicle count data, population census data, fuel usage at industrial point sources, and aviation and shipping vessel counts. The inversion solved for the actual concentration measurements at each site, which was made possible by the use of the Cape Point back-

ground site to provide information on the boundaries, and was necessary due to the effect of topography on the atmospheric transport, affecting particularly the sensitivity of the Robben Island site to the surface fluxes. Night-time observations were included, but allocated much larger errors compared to the daytime observations.

    The inversion was able to substantially improve the agreement between the modelled and observed concentrations, and able to better represent the diurnal cycle in the concentrations compared with the prior modelled concentrations. The mean

bias in the modelled concentrations was reduced from -2.9 ppm, with interquartile range -9.1 to 3.7, for the prior modelled concentrations, to 0.5, with interquartile range -1.5 to 1.5, for the posterior modelled concentrations at Robben Island, and from a bias of 2.4 ppm in the prior modelled concentrations at the Hangklip site, with interquartile range -2.3 to 6.5, to a bias of 0.04, with interquartile range -1.1 to 0.8. The standard deviations of the posterior residuals at both sites were reduced to values below that of the observed concentrations.

The inversion solved for working week and weekend fossil fuel emissions, and weekly biogenic fluxes, each split into day and night contributions, for each month; therefore six surface sources per week within each of the 10,201 surface pixels. The inversion was also allowed to solve for each of the four boundary concentrations (north, east, south and west), but these were





provided with tight constraints provided by the background site. The inversion tended to reduce fossil fuel emissions over all months. During the warmer, drier months, the inversion increased the biogenic fluxes, but reduced the biogenic emissions during the cooler, wetter months. The uncertainty reduction in the total estimate for the domain over each month ranged between 8.6 to 40.0% for the biogenic fluxes and between 0.4 to 16.4% for the fossil fuel fluxes. Model assessment by means of the $\chi^2$ statistic indicated that the mean statistic was 1.48 over all months, indicating that either the prior values for the model errors or the uncertainty in the fluxes was not specified high enough for some months. A companion paper on sensitivity analyses will address different options for the specification of the correlations between errors in the modelled concentrations, how these prior errors are determined, how correlations are determined between the prior fluxes, and how the state vector is specified. Greater confidence is given to the inversion's ability to correct the total flux within each pixel, rather than the individual flux estimates.

## 1 Introduction

Cities are under pressure to reduce their carbon dioxide emissions. In the last 10 years (2006 to 2015), the mean annual increase in carbon dioxide ($CO_2$) concentrations in the global atmosphere has been 2.11 ppm per year (Dlugokencky and Tans, 2016) (NOAA/ESRL 2016), a sharper rise in $CO_2$ emissions than the preceding decades (IPCC , 2014). Of current anthropogenic greenhouse gas emissions, $CO_2$ contributes 76% (IPCC 2014). While cities cover a mere 2% of the global land surface area, they are responsible for 70% of anthropogenic greenhouse gas emissions (UN–Habitat, 2011), and between 71 and 76% of $CO_2$ emissions from global final energy use (Seto et al 2014). Annual urban $CO_2$ emissions are more than double the net terrestrial or ocean carbon sinks (Le Quéré et al., 2013).

South Africa is the single largest emitter of $CO_2$ on the continent of Africa, and the 13th largest emitter in the world (Boden et al., 2011). South African cities are home to 63% of the present population (Statistics South Africa, 2011), and by 2030 this is predicted to be 71%. The City of Cape Town (CT) saw its population increase from 2,563,095 in 1996 to 3,740,026 in 2011, an overall increase of 46% (City of Cape Town, 2011). Although cities are by far the largest contributors of anthropogenic emissions, they are also seen as having the greatest potential to provide solutions for emissions reduction and climate change mitigation (Seto et al., 2014; Wu et al., 2016). By mitigating the $CO_2$ impact of cities, cities play a pivotal role in decreasing their own climate vulnerability. But there are also additional co-benefits which include improving air-quality, energy access, public health, city liveability, and developing the economy and job creation through advances in green technology (Seto et al., 2014).

Formal climate action plans are developed by governments and city managers, whereby the roadmap for implementing greener policies is provided, such as encouraging and developing public transport which makes use of low emission technologies, mass and rapid transport systems, and building retrofits (Sugar and Kennedy, 2013; Erickson and Tempest, 2014). Many cities are taking it on themselves to respond to the climate crisis, reacting to limited international and national policy progress, which is viewed to be moving too slowly to address the required need for mitigation against climate change (Hutyra et al., 2014). To determine if the plans implemented are having the anticipated effect of lowering $CO_2$ emissions, monitor-



ing is required. Emissions need to be known at baseline, and monitored through time. Monitoring, reporting and verification (MRV) is a concept which is fundamental to most market and policy-based mechanisms in climate economics (Bellassen and Stephan, 2015). In order for emission reduction strategies to be properly implemented and assessed, an MRV approach should be adopted so that emission reduction claims can be validated in a consistent and reliable manner. Currently, the primary source

of this information for cities is by means of emissions inventories. This relies on the collection of activity data to provide an inventory of emissions from different sectors or specific point sources. These inventories are not perfect representations of $CO_2$ emissions, heavily dependent on accurate reporting, emission factors, and on assumptions regarding temporal or spatial disaggregation of emissions (Andres et al., 2012). As the resolution of the inventory analysis increases, so too do the errors associated with these emission estimates (Andres et al., 2014). As the importance of these inventories increases with the need to

quantify emissions and assess emission targets, it has become necessary to verify the accuracy of these estimates (NRC, 2010). Adequate MRV implementation requires transparency, quality and comparability of information, with narrow uncertainty estimates (Wu et al., 2016). Currently, uncertainties associated with urban emissions far exceed emission reduction goals, and therefore verification remains challenging. The large amount of uncertainty is due to factors such as incomplete data, inconsistency in reporting between different institutions or facilities, fugitive emissions from point sources such as those caused by

gas leaks, and methodology which is rarely checked against scientific standards and procedures (Hutyra et al., 2014). A way of verifying inventory data for a city, and reducing uncertainty of inventory estimates, is by means of the Bayesian atmospheric inversion technique.

Originally implemented to determine global, large scale sources and sinks of $CO_2$ (e.g. Chevallier et al. (2010)), regional or mesoscale scale atmospheric inversions are becoming more common (Lauvaux et al., 2008, 2009, 2012; Schuh et al., 2013),

and more recently city scale inversion studies are being conducted in Europe and North America (Strong et al., 2011; Duren and Miller, 2012; McKain et al., 2012; Brioude et al., 2013; Kort et al., 2013; Lauvaux et al., 2013; Bréon et al., 2015; Turnbull et al., 2015; Boon et al., 2016). These top down approaches make use of an atmospheric transport model to relate observations of $CO_2$ in the atmosphere to the emissions from the domain of interest (Lauvaux et al., 2012). This method applies corrections to the inventory data, which enters the inversion calculation by means of the prior estimates. This paper reports the results for

such an inverse study for Cape Town, South Africa.

Atmospheric top down approaches to determining $CO_2$ emissions have the simultaneous advantage and disadvantage of capturing information from all sources and sinks, some of which may have been excluded from the inventory analysis of the domain. All emissions are observed as an aggregated total, therefore all emission sources are accounted for, but it is challenging to separate out these $CO_2$ emissions into different components of the total $CO_2$ budget without additional measurements, such

as $\Delta^{14}C$ and $\delta^{13}C$ isotope measurements (Newman et al., 2016) or confidence about spatial and temporal patterns of emission (Shiga et al., 2014). At the moment background conditions are not sufficiently characterised in order to use isotope tracers to differentiate between fossil fuel and biogenic sources, as these measurements are far rarer than atmospheric measurements of $CO_2$ mole fractions (Turnbull et al., 2015). Even then it would not be possible to assign anthropogenic $CO_2$ emissions to each sector. Therefore it is necessary to conduct an atmospheric inversion study in conjunction with detailed $CO_2$ inventory analysis

if no such inventory exists, where all the main contributors to the anthropogenic $CO_2$ budget are considered. This analysis





underpins the assumptions of human behaviour driving the anthropogenic emissions. It is important to verify these assumptions regarding human activity in order to assess if mitigation interventions are having the desired impact (Strong et al., 2011). Better understanding of the underlying processes at the urban scale and improved quantification will provide information contributing towards the policy decisions made by urban practitioners, and improve understanding of urban dynamics and inform future

scenarios (Hutyra et al., 2014). An example of this is the detailed street level inventory analysis undertaken in the Hestia project for U.S. cities Indianapolis, Los Angeles, Phoenix and Salt Lake City (Gurney et al., 2012; Davis et al., 2017). Preceding these inventories was the Vulcan inventory which covers the contiguous U.S. (Gurney et al., 2009). These detailed inventories have made possible atmospheric inversion exercises or other top down methods for obtaining urban $CO_2$ flux estimates for these cities (Strong et al., 2011; Brioude et al., 2013; Bréon et al., 2015; Lauvaux et al., 2016). Such a detailed inventory analysis

is not available for any South African city, and therefore a detailed spatially and temporally disaggregated inventory analysis of direct $CO_2$ emissions was undertaken for CT specifically for the use of this atmospheric inversion exercise (Nickless et al., 2015a).

Atmospheric inversion models have their own sources of uncertainty, which include atmospheric transport modelling errors (particularly at night when the planetary boundary layer is very shallow); incorrect characterisation of prior errors (which in-

cludes errors from the inventory analysis); atmospheric measurement errors; and representation errors which occur as emissions are coerced into homogeneous grid cells across different data sources. In the case of cities, atmospheric transport modelling is further complicated by small-scale turbulence, highly heterogeneous surface characteristics, and urban heat island effects (Hutyra et al., 2014; Bréon et al., 2015).

Previous studies on estimating $CO_2$ emission for cities have found that errors in atmospheric transport modelling are a

significant contributor to the overall uncertainty of emission estimates (Lauvaux et al., 2013, 2016; Bréon et al., 2015), and therefore more work is required to refine these models so that they can perform more reliably during these periods of high uncertainty before they can be used to infer emission estimates at all times of the day. The atmospheric transport modelling in this study is provided by the Conformal Cubic Atmospheric Model (CCAM) (McGregor and Dix, 2008) at the resolution of $1\,\mathrm{km} \times 1\,\mathrm{km}$. This model at a slightly lower resolution was previously used for a regional network design study, making use of

the same Bayesian inverse methodology (Nickless et al., 2015b), and has been verified over southern Africa at relatively low resolutions through to ultra high resolution ($1\,\mathrm{km} \times 1\,\mathrm{km}$) (Engelbrecht et al., 2009, 2011).

To be able to verify emissions from underlying processes, higher resolution inverse modelling systems are needed to better understand and quantify emissions from different sectors. Lauvaux et al. (2016) considered sector specific anthropogenic emissions but ignored biogenic emissions. This was possible due to the selection of the dormant period when biogenic emis-

sions would have been at a minimum. When considering longer periods or for cities in regions which may not have a dormant vegetation period, this assumption will not be valid, particularly for a medium-sized city, where natural processes can be a significant contributor to the carbon budget. Such would be the case for South African cities, such as CT and Johannesburg, where large national parks and other natural areas are located near or within city limits and within city vegetation growth is non-negligible. Additionally, CT is surrounded by large agricultural sector consisting of vineyards and fruit orchards. Ironi-

cally, there are features of cities which allow for better plant growth. For example, the urban heat island effect leads to a longer





growing season for plants, and reduced wind within cities leads to less plant stress resulting in better plant growth (Buyantuyev and Wu 2012). In addition, nitrogen deposition within cities leads to increased nutrient availability, and particularly in arid regions, cities cause augmented water availability for plants (Hutyra et al., 2014). If allowed growing space, plants can make a significant contribution to the carbon budget of a city.

Therefore for a city like CT, biogenic emissions should not be ignored, and within atmospheric inversion studies are usually accounted for by means of a land surface exchange model (Bréon et al., 2015; Staufer et al., 2016). Bréon et al. (2015) and Staufer et al. (2016) made use of the C-TESSEL land atmosphere scheme which is used in the ECMWF forecasting system. In this study we have made use of the CABLE (Community Atmosphere Biosphere Land Exchange) model to represent the biogenic $CO_2$ fluxes in the $CO_2$ budget (Kowalczyk et al., 2006). CABLE was dynamically coupled to CCAM, so that the

land-atmosphere exchange model was run at the same resolution as meteorology.

We present the results of an atmospheric inversion which was used to model fluxes over the CT using atmospheric $CO_2$ concentration data that was collected over a sixteen month period. The domain considered was a $100\,\mathrm{km} \times 100\,\mathrm{km}$ region with CT at the centre. The spatial resolution of the atmospheric transport model was $1\,\mathrm{km} \times 1\,\mathrm{km}$, and the surfaces sources were set to match this spatial resolution. Fluxes were solved for at a weekly time step, separately for day and night, with fossil fuel and

biogenic fluxes solved separately and fossil fuel fluxes separated by week and weekend.

## 2   Methods

### 2.1   Bayesian Inverse Modelling Approach

The Bayesian synthesis inversion method, as described by Tarantola (2005) and Enting (2002), was used to solve for the sources in this study. This method has been employed for global inversions (Bousquet et al., 1999; Kaminski et al., 1999; Rayner et al.,

1999; Gurney et al., 2002; Peylin et al., 2002; Gurney et al., 2003; Law et al., 2003; Baker et al., 2006; Rayner et al., 2008; Ciais et al., 2010), as well as for many of the city scale inversions (Lauvaux et al., 2016; Bréon et al., 2015). This methodology was employed over South Africa in a previous optimal network design study (Nickless et al., 2015b). The observed concentration ($c$) at a measurement station at a given time can be expressed as the sum of different contributions from the surface fluxes, from the boundaries and from the initial concentration at the site. A linear relationship can be used to describe the relationship

between the modelled concentrations and the contribution from the sources (surface fluxes and boundary inflow):

$$\mathbf{c}_{mod} = \mathbf{Hs} \qquad (1)$$

The vector of the modelled concentrations $\mathbf{c}_{mod}$ is a result of the contribution from the sources $\mathbf{s}$, described by the transport or sensitivity matrix $\mathbf{H}$ . $\mathbf{H}$ is the Jacobian matrix representing the first derivative of the modelled concentration at the observa-





tional site and dated with respect to the coefficients of the source components (Enting, 2002). If we assume a Gaussian error distribution for the surface fluxes and concentrations we obtain the following cost function for our least squares problem:

$$J(\mathbf{s}) = \frac{1}{2}\left((\mathbf{c}_{mod} - \mathbf{c})^T\mathbf{C_c}^{-1}(\mathbf{c}_{mod} - \mathbf{c}) + (\mathbf{s} - \mathbf{s_0})^T\mathbf{C_{s_0}}^{-1}(\mathbf{s} - \mathbf{s_0})\right) \tag{2}$$

5  where $\mathbf{C_c}$ is the error covariance matrix of the observations, $\mathbf{s_0}$ is the vector of prior flux estimates, $\mathbf{s}$ is the vector of predicted sources and $\mathbf{C_{s_0}}$ is the prior error covariance matrix of the sources. The Bayesian cost function minimises both the difference between modelled concentrations and measurements and the difference between prior source estimates and predicted sources.

The posterior source estimates can then be solved for as:

$$\mathbf{s} = \mathbf{s_0} + \mathbf{C_{s_0}}\mathbf{H}^T\left(\mathbf{H}\mathbf{C_{s_0}}\mathbf{H}^T + \mathbf{C_c}\right)^{-1}(\mathbf{c} - \mathbf{H}\mathbf{s_0}) \tag{3}$$

The posterior covariance matrix can be calculated as follows (Tarantola, 2005):

$$\begin{aligned}\mathbf{C_s} &= \left(\mathbf{H}^T\mathbf{C_c}^{-1}\mathbf{H} + \mathbf{C_{s_0}}^{-1}\right)^{-1} \tag{4}\\ &= \mathbf{C_{s_0}} - \mathbf{C_{s_0}}\mathbf{H}^T\left(\mathbf{H}\mathbf{C_{s_0}}\mathbf{H}^T + \mathbf{C_c}\right)^{-1}\mathbf{H}\mathbf{C_{s_0}} \tag{5}\end{aligned}$$

## 2.2  State vector - s

The state vector which describes the total contribution to the concentration observed during an hour at a site from a particular surface source grid during a given week is:

20  $$\mathbf{s} = \mathbf{s}_{\text{ff week day}} + \mathbf{s}_{\text{ff week night}} + \mathbf{s}_{\text{ff weekend day}} + \mathbf{s}_{\text{ff weekend night}} + \mathbf{s}_{\text{NEE day}} + \mathbf{s}_{\text{NEE night}} \tag{6}$$

The surface source is made up of the fossil fuel emissions for a week emitted during the working week and weekend separately for day and night, and the biogenic emissions for the full week, also separated into day and night fluxes. Each of these sources is solved for separately in the inversion solution. Finally, this total surface source contribution is added to the contribution from

25  the mean boundary concentration for the week, which is solved for in the inversion rather than imposed, to give the observed concentration at the receptor site.

As the reference inversion which we present in this paper, we ran the inversion solution for the full month and solved for each week separately, four weeks in total. Therefore for a given surface source we solve for $2\times4 = 8$ biogenic weekly fluxes





(four day and four night) and $4\times4 = 16$ fossil fuel fluxes (four day working week, four night working week, four day weekend, and four night weekend). As a sensitivity analysis, presented in a companion paper, we examined two alternative methods of solving for the weekly flux. We considered solving for a mean weekly flux for that month. In this case for a surface source we solved for two biogenic mean weekly fluxes (day and night) and four fossil fuel mean weekly fluxes (day and night week, day

and night weekend). Secondly we ran the inversion separately for each week. In this case only the concentration measurements for one week were used and the individual week fluxes (two biogenic and four fossil fuel) were solved for, and this was repeated for each of the four weeks in the month, resulting in 24 individual fluxes solved for every month. The benefit these two alternative methods provide is that the resulting $\mathbf{C_{s_o}}$ matrix is much smaller compared to the reference case.

In the reference case, for each month, the prior fluxes consisted of $101\times101\times4\times24 = 979{,}296$ surface sources for each

month plus the $4\times2\times4 = 32$ boundary contributions, a large number of sources for which to solve compared to previous inversions of this kind. When solving for only one week, or a mean weekly flux for a particular month, the number of surface sources reduced to $101\times101\times24 = 244{,}824$. Solving for individual weeks required $4\times2$ additional boundary concentrations to be added to the state vector, and when solving for the mean weekly flux for the month, we allowed the boundary concentrations to differ for each week, and therefore we still solved for the 32 boundary concentrations as in the reference case.

## 2.3   Concentration measurements - c

The existence of the Cape Point Global Atmospheric Watch (GAW) station made CT an ideal candidate for a city scale inversion exercise. The Cape Point station is located approximately $60\,\mathrm{km}$ south of CT within a nature reserve, situated on the southern-most tip of the Cape Peninsula at a latitude of $34°21'12.0''$ south and longitude of $18°29'25.2''$ east. The inlet is located on top of the $30\,\mathrm{m}$ measurement tower, which is located on a cliff $230\,\mathrm{m}$ above sea level. The station observes

background measurements of $CO_2$ when observing maritime air derived directly from the south-western Atlantic Ocean. This is an extensive region stretching from $20°$ (sub-equatorial) to $80°$ (Antarctic region) (Brunke et al., 2004). Therefore background measurements at Cape Point are well representative of the background $CO_2$ signal influencing the Cape Peninsula. The background signal at Cape Point, obtained from a percentile filtering technique, was used as the prior estimate of background concentrations. The uncertainty in the background concentration is based on the hourly measurements contributing towards the

weekly mean. As the variability in the background $CO_2$ in the southern hemisphere is very small, much smaller than for the northern hemisphere, this results in a tight constraint on the prior background $CO_2$ concentrations.

Two monitoring sites were established at Robben Island and Hangklip lighthouses. Due to the dominant wind directions in CT (Fawcett et al., 2007), either from the south or north west, the location of the Robben Island and Hangklip stations were well suited for observing contributions from the area of interest, particularly from CT. One site would usually observe mainly

background influence whereas the other would be measuring enhancements from CT. The location of these sites in relation to the domain are shown in Figure 1.





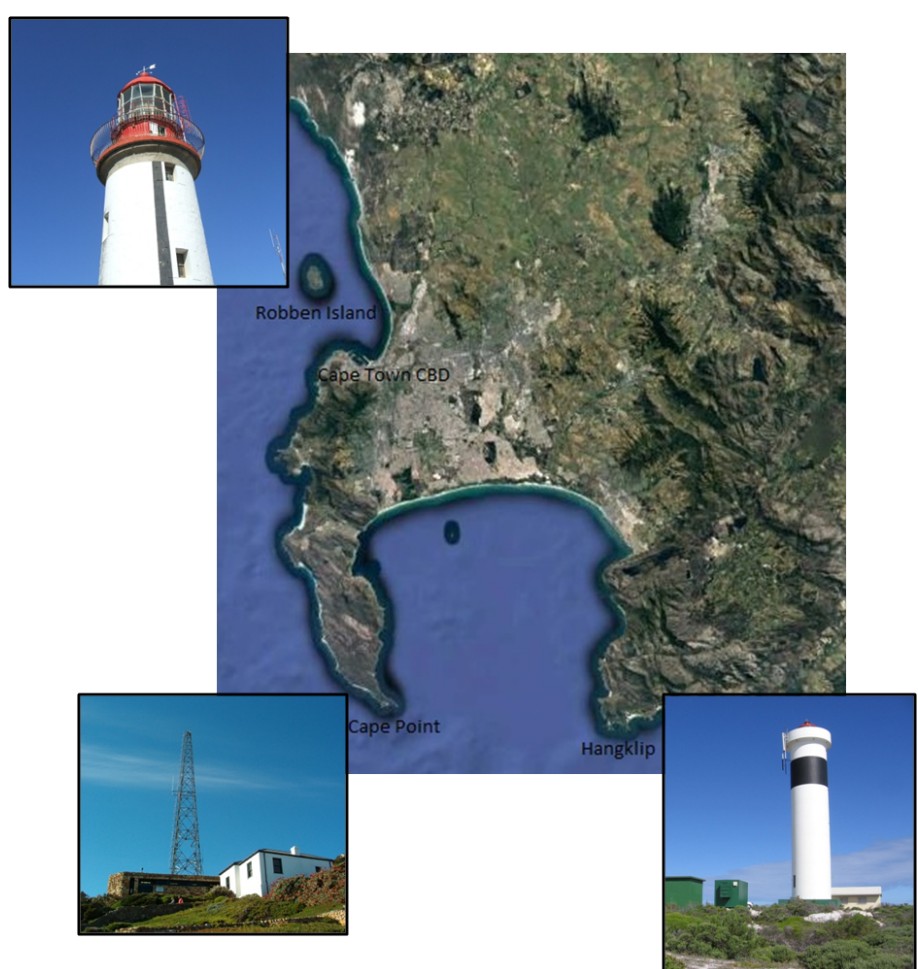

**Figure 1.** Google Earth image of the domain, where Cape Town is located at the centre. The locations of the measurement sites and the Cape Point GAW station background site are indicated together with images of these sites (Photo credits: Ernst Brunke and Alecia Nickless).





Each site was equipped with a Picarro Cavity Ring-down Spectroscopy (CRDS) (Picarro G2301) instrument. This instrument measures $CO_2$, methane ($CH_4$), and water vapour ($H_2O$) simultaneously, every five seconds, producing a precision of better than 0.05 parts-per-million volume (ppmv) for $CO_2$, 0.07 parts-per-billion volume (ppbv) for $CH_4$, and 100 ppmv for $H_2O$. This instrument maintains high linearity, precision, and accuracy over changing environmental conditions, requiring only minimal calibration, and is recognised as the one of the highest precision instruments for measurement of the top three greenhouse gasses (Crosson, 2008).

The inlet of the measurement tube at each site was located at the top of the lighthouse, and had a Gelman filter to prevent contamination of the instrument through aerosols or water droplets. The inlet tube led to a VICI rotary valve which directed the sampled air stream to the Picarro instrument. Approximately every four days the rotary valve switched to a calibration line which allowed the flow of calibration gas through the instrument for a period of half an hour.

The Robben Island lighthouse is an 18 m tall circular masonry tower, and the height of the focal plane of the light is 47 m above the high water level. The location of the lighthouse is $33°48'52.20''$ south and $18°22'29.25''$ east. The Hangklip lighthouse is a 22 m tall concrete tower, where the focal plane of the light is 34 m above the high water level. It is located at $34°23'11.40''$ south and $18°49'42.30''$ east. It is located on the tip of False Bay, opposite to Cape Point.

## 2.4 System Meteorology

CCAM is the variable-resolution global atmospheric model developed by the Commonwealth Scientific and Industrial Research Organisation (CSIRO) (McGregor, 1996; McGregor and Dix, 2001; McGregor, 2005a, b; McGregor and Dix, 2008). It employs a semi-implicit semi-Lagrangian method to solve the hydrostatic primitive equations. The Geophysical Fluid Dynamics Laboratory (GFDL) parameterisations for long-wave and short-wave radiation are used (Lacis and Hansen, 1974; Schwarzkopf and Fels, 1991), with interactive cloud distributions determined by the liquid and ice-water scheme of Rotstayn (1997). Total-variation-diminishing vertical advection is applied to solve for the advective process in the vertical. A stability-dependent boundary layer scheme based on Monin Obukhov similarity theory is employed (McGregor, 1993), together with the non-local treatment of Holtslag and Boville (1993). A canopy scheme is included, as described by Kowalczyk et al. (1994), having 6 layers for soil temperatures and soil moisture (solving Richard's equation) and 3 layers for snow. The cumulus convection scheme uses a mass-flux closure (McGregor, 2003), and includes downdrafts, entrainment and detrainment. Gravity wave drag is parameterised following (Chouinard et al., 1986).

CCAM may be applied in stretched-grid mode to function as a regional climate model, thereby providing a flexible framework for downscaling reanalysis data or global circulation model simulations to high resolution over an area of interest. Stretched grids are obtained using the Schmidt (1977) transformation. Here the model is applied to downscale reanalysis data of the National Centres for Environmental Prediction (NCEP; (Kalnay et al., 1996)) to ultra-high resolution over the CT study area. A multiple-nudging approach was followed to downscale the 250 km resolution reanalysis data to a resolution of 60 km over southern Africa, 8 km over the south western Cape and subsequently to a 1 km resolution over Cape Point. The 8 km resolution domain stretched over an area of about $1300 \times 1300 \, km^2$, whilst the 1 km resolution domain centred over False Bay stretched over an area of about $160 \times 160 \, km^2$. Output was stored at a time resolution of 1 hour. CCAM was spectrally nudged



with the synoptic-scale forcing reanalysis data at 6-hourly intervals for the period 1979-2013 using a scale-selective Gaussian filter (Thatcher and McGregor, 2009, 2010). This forcing was applied from $900\,\mathrm{hPa}$ higher up into the atmosphere. Sea-surface temperatures from the NCEP data set were used as lower boundary forcing.

## 2.5 Lagrangian Particle Dispersion Model (LPDM) - H

In order to generate the sensitivity matrix, $\mathbf{H}$, for the inversion procedure, which maps the surface fluxes and boundary inflows to the concentrations observed at the receptor sites, a Lagrangian particle dispersion model (LPDM) is run in backward mode. An LPDM simulates the release of a large number of particles from arbitrary emissions sources and records the location of the particles at fixed time steps (Uliasz, 1993, 1994). The model implemented in this study was developed by Marek Uliasz (1993), which will be referred to as LPDM. LPDM is driven by the three-dimensional fields of mean winds ($u$, $v$, $w$), potential temperature and turbulent kinetic energy (TKE), which are obtained from the CCAM model. When LPDM is run backward in time, in receptor-orientated mode, the particle counts can be used to generate $\mathbf{H}$ for a given receptor site, as described in Ziehn et al. (2014) and Nickless et al. (2015b) following Seibert and Frank (2004).

We use the LPDM originally proposed by Uliasz (1994), which we run in reverse mode for each measurement station in the observation network. The sensitivity matrix for the four week period during each month of the study is generated by allowing the LPDM model run over a 2 month period, and particle counts are extracted for the weeks of interest. Particles were released every 20 seconds and each particle's position was recorded at 1 minute intervals. Particles that were near the surface were allocated to a surface grid cell and the total count within each of these was obtained to determine the surface influence or sensitivity. These counts depended on the dimensions and position of these surface grid boxes. The particle counts were used to calculate the source–receptor ($s$–$r$) relationship, or influence functions, which form the sensitivity matrix $\mathbf{T}$. Here, we followed Seibert and Frank (2004) to derive the elements of that matrix. As described in Ziehn et al. (2014), we modified the approach of Seibert and Frank (2004) to account for the particle counts which were produced by our LPDM as opposed to the mass concentrations which were outputted by the atmospheric transport model in their study. The resulting $s$–$r$ relationship between the measurement site and source $i$ at time interval $n$, which provide the elements of the matrix $\mathbf{H}$, is:

$$\frac{\partial \bar{\chi}}{\partial \dot{q}_{\mathrm{in}}} = \frac{\Delta T \mathrm{g}}{\Delta P} \overline{\left(\frac{N_{\mathrm{in}}}{N_{\mathrm{tot}}}\right)} \frac{29}{12} \times 10^6, \tag{7}$$

where $\bar{\chi}$ is a volume mixing ratio (receptor) expressed in ppm and $\dot{q}_{in}$ is a mass flux density (source), $N_{in}$ the number of particles in the receptor surface grid from source grid $i$ released at time interval $n$, $\Delta T$ is the length of the time interval, $\Delta P$ is the pressure difference in the surface layer, g is the acceleration due to gravity, and $N_{tot}$ the total number of particles released during a given time interval.

In this inversion setup weekly fluxes of carbon were separated into day and night-time contributions, into fossil fuel and NEE contributions, and in the case of fossil fuels, into working week and weekend contributions. The total flux for a source region is the sum of all these contributions, and therefore, the mass flux density $\dot{q}_{in}$ in Eqn. (7) has units of $\mathrm{g\,C\,m^{-2}\,week^{-1}}$ for the day and similarly for the night. In the case of the NEE contributions the particle count $N_{in}$ is the sum over one week





($\Delta T$=1 week (day/night)). For fossil fuel sources, the particle count is separated into the contribution from the working week and from the weekend, separately for day and night.

The surface layer height was set to $50\,\mathrm{m}$ which corresponds to approximately $595\,\mathrm{Pa}$ ($\Delta P$). If we assume well mixed conditions, then the $s–r$ relationship should be independent of the thickness of the surface layer, as long as the layer is not too
deep, as the particle count will be adjusted proportional to the volume of the grid box. Under stable conditions, this may not be the case (Seibert and Frank, 2004).

The spatial resolution of the surface flux grid boxes was set to be the same as that of the atmospheric transport model, resulting in a grid consisting of $101\times101$ grid boxes (a resolution of approximately $0.01° \times 0.01°$ or $1\,\mathrm{km}\times1\,\mathrm{km}$). As the land surface model CABLE was dynamically coupled to CCAM, the biospheric fluxes were at the same resolution as the
atmospheric transport model. This high spatial resolution was deliberately maintained throughout the inversion to avoid any unnecessary aggregation error (Kaminski et al., 2001; Nickless et al., 2015b), and allowed for small scale transport features to be maintained in $\mathbf{H}$.

Since the surface sources are expressed as fluxes of carbon, the contribution to the concentration at the measurement site is expressed in the amount of carbon seen at the measurement site from a particular source. In the case of the boundary sources
(or contributions from outside of the domain) which are given as concentrations, their contributions to the concentration at the measurement site are expressed as a proportion of their concentration, dependent on their influence at the receptor site. Ziehn et al. (2014) shows that by calculating the Jacobian which provides the sensitivities of observed concentrations to boundary concentrations, the boundary contribution can then be written as:

$$\mathbf{c}_b = \mathbf{H}_B\mathbf{c}_B \tag{8}$$

where $\mathbf{H}_B$ is the Jacobian, $\mathbf{c}_B$ the boundary concentrations and $\mathbf{c}_b$ the contributions from the boundary. The row elements of $\mathbf{H}_B$ sum to one.

For the network design, four boundaries (north, south, east and west) were used and we calculated the sensitivity of hourly observed concentrations to weekly boundary concentrations (separate day and night weekly concentrations).

**2.6  Inventory of anthropogenic emissions**

An inventory analysis was conducted specifically for this atmospheric inversion exercise (Nickless et al., 2015a). The population of CT was 3,740,025, as reported in the 2011 census (Statistics South Africa, 2011). At that time the energy use of the city was determined to be 50% due transport, 18% due to residential, 16% due to commercial, 14% due to industrial and 1% due to government use. But of the carbon emissions due to energy usage, only 27% were attributed to the transport sector as a
result of the carbon intensive usage of coal for electricity generation to provide almost all of the energy to the residential and commercial sectors in South Africa, which emit approximately 29% and 28%, respectively, of the total carbon emissions of CT (City of Cape Town, 2011).



The anthropogenic emissions were subdivided into those due to road transport, airport and harbour emissions, residential lighting and heating, and industrial point sources. Road transport emissions were derived from modelled values of vehicle kilometres for each section of the road network, derived from observed vehicle count data. The vehicle kilometres were scaled for each hour of the day, and separately for working week days and weekend days. Therefore the vehicle emissions for day and
night are distinctive for the week / weekend and day / night periods.

Airport emissions were derived from landing and takeoff cycles, as reported by Airports Company South Africa for each month. We used the IPCC reported average emission factors for domestic and international fleets (IPCC , 2000), and these were used to convert the airport activity data into emissions of $CO_2$. Emissions were expected to be concentrated during the hours of 6:00 and 22:00, and so the monthly emission was divided evenly between these hours. Harbour emissions were derived for port
activity published by the South African Ports Authority for each month. Based on the gross tonnage of vessels which docked at the port during the month, emissions could be derived as described in DEFRA (2010). The monthly emissions were divided equally between all hours of the month, as it was assumed that harbour activities would be continuous.

Residential emissions for lighting and heating were derived from population count data obtained for each of the municipal wards in 2011 (Statistics South Africa, 2011). The South African government reports on the fuel used for domestic heating
and lighting (South African Department of Energy, 2009). This was divided between the total population, and then allocated to each ward depending on the population residing in that area. The fuel usage was scaled according to the proportion of fuel used for cooking, lighting and heating, where 75% of the annual heating fuel usage was assumed to take place during the winter months (March to August). It was assumed that 75% of the annual energy consumed was used for heating, 20% for cooking and 5% for lighting.

CT provided monthly fuel usage by the largest industrial emitters. The reported fuel usage for the top fuel users were converted directly into $CO_2$ emissions by multiplying these figures with the Defra greenhouse gas emission factors (DEFRA, 2013a). The fuel types that were considered included heavy fuel oil, coal, diesel, paraffin and fuel gas which were divided into liquid petroleum gas and refinery fuel gas. As no information was available about when the activity was occurring at these facilities, the emissions were divided equally between all hours of the month.

**2.7   Biogenic emissions**

CCAM was dynamically coupled to the land surface model CABLE, which allows for feedbacks between land surface and climate processes, such as leaf area feedback on maximal canopy conductance and latent heat fluxes (Zhang et al., 2013). This type of coupling has successfully been implemented in CSIRO's national earth system modelling scheme (Australian Community Climate and Earth System Simulator or ACCESS) and describes land-atmosphere exchanges of energy, carbon, and
water using biogeochemical, vegetation-dynamic and disturbance processes (Law et al., 2012). Several studies have validated CABLE under different ecosystems and parameters using both global model simulations (e.g. Zhang et al. (2009)), and site level offline CABLE simulations (Zhang et al., 2013).





The model produces hourly estimates of net ecosystem exchange (NEE), which were aggregated into weekly (day and night) estimates, and used as the prior estimate of biogenic fluxes over the land. The spatial resolution of these prior NEE fluxes were kept at a 1 km resolution.

The $CO_2$ fluxes over the ocean were obtained from Gregor and Monteiro (2013). This study characterised the seasonal cycle
of air-sea fluxes of $CO_2$ in the southern Benguela upwelling system off the South African west coast. A time series of $pCO_2$, derived from total alkalinity and dissolved inorganic carbon and scatterometer-based wind, was obtained from six monthly cross-shelf cruises in the St. Helena Bay region during 2010. Daily $CO_2$ fluxes were derived from these $pCO_2$. These fluxes were applied as prior estimates to the ocean surface grids within the domain. Therefore, an assumption was made that ocean $CO_2$ fluxes are relatively homogeneous in space near the south western coast of South Africa, but the inversion was given the
ability to differentially adjust each of the ocean sources in the posterior estimates.

## 2.8 Boundary concentrations

For this atmospheric inversion study, the selection of CT for this exercise was determine by its location and the presence of the Cape Point GAW station. This site provides a long term record of background $CO_2$ concentrations for the area, and provided the tightly constrained prior estimates of the background concentrations for the inversion. These continuous measurements
of the background $CO_2$ levels meant that we were not dependent on the atmospheric transport model to produce estimate of background $CO_2$ levels, which are prone to large errors (Lauvaux et al., 2016). This also meant that we did not have to depend on a gradient approach to estimate the fluxes, which results in measurements being used only when the wind direction and wind speed are within narrow limits (Bréon et al., 2015; Lauvaux et al., 2016; Staufer et al., 2016). The percentile-based statistical filter applied to the Cape Point $CO_2$ measurements to obtain background levels of atmospheric $CO_2$ selects approximately
75% of the measurements (Brunke et al., 2004). The mean weekly background concentration, separate for day and night, is determined from the percentile filtered measurements at the site, and was used as the prior boundary concentration for each of the four cardinal directions. The inversion was then allowed to adjust these concentrations slightly. The prior variance assigned to the boundary concentrations was equal to the variance of the measured hourly concentrations for that period. The daytime weekly background concentrations are shown in Figure 2. The uncertainty in the background $CO_2$ concentrations ranged
between 0.32 and 0.90 ppm, with a mean of 0.62 ppm.




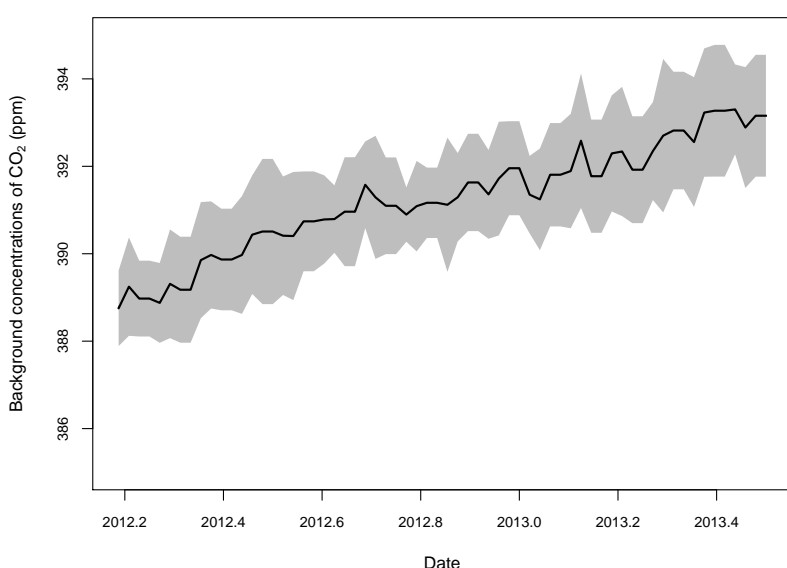

**Figure 2.** Weekly mean background concentrations of $CO_2$ (ppm) as measured at Cape Point GAW station, with 95% confidence interval represented by the grey shaded area. The mean concentrations are calculated from percentile filtered observations, extracting only those observations considered to be representative of background conditions.



## 2.9 *A priori* covariance matrix - $\mathbf{C_{s_0}}$

The variances and covariances in the *a priori* covariance matrix provide the uncertainties in the prior estimates of the sources $\mathbf{s_0}$. The decision of whether to include non-zero covariances in $\mathbf{C_{s_0}}$ determines in part how much freedom the inversion has to modify the source estimates based on the observed concentrations $\mathbf{c}$. If the prior covariance elements are significant then the

estimate for a particular source will be more dependant on the prior estimates of the surrounding sources compared with an inversion where the covariance terms are set to zero. Covariance terms in the prior flux covariance matrix spread the influence of concentration measurements on the posterior flux estimates, subject to sensitivity matrix which determines which covariance terms contribute to the estimation of the flux.

### 2.9.1 Fossil fuel emissions

Error propagation techniques were used to estimate the uncertainties in the sector specific fossil fuel emissions. This is described in Nickless et al. (2015a). DEFRA (2013b) provides estimates of activity data error and emission factor error under each fuel type for industrial sources. These values were used to estimate the uncertainties in the industrial point source emissions using the following formula:

$$\delta\mathbf{s}_0 = |\mathbf{s}_0| \times \sqrt{(\frac{\delta\mathbf{activity\ data}}{\mathbf{fuel\ usage}})^2 + (\frac{\delta\mathbf{emission\ factor}}{\mathbf{emission\ factor}})^2} \tag{9}$$

In the case of the vehicle emissions, which relied on count data, Poisson errors were assumed, and propagated together with the uncertainty in the conversion factors for the different vehicle types. For airport and harbour emissions, vessel counts were assumed to be correct, and therefore the uncertainty in the emissions contained within the emission factors for the different vessel types and activities. For aircraft, these errors are assumed to be 34% for the international fleet and 28% for the domestic

fleet (IPCC , 2000). The error estimate for berth and manoeuvring activities of shipping vessels is reported to be between 20% and 30%, and therefore a conservative estimate of 30% was used (DEFRA, 2010). For domestic heating and lighting, the estimates relied on population census, which had a reported omission rate of 15%. There was no information available on the variability in fuel usage between households, and therefore the uncertainty in the domestic emissions was set at 30% as a relatively arbitrary, but conservative level.

The resulting uncertainty estimates ranged between 6.7% to 71.7% of the prior fossil fuel emission estimate, with a median percentage of 34.9% to 38.4% depending on the month. These values are in general slightly more conservative compared to uncertainties that were determined by Bréon et al. (2015) for the AirParif inventory, which were set at 20% throughout. The spatial distribution of the fossil fuel fluxes during the month of March 2012 are mapped in Figure 3. The daytime fossil fuel emissions have a mean of $0.006 \, \text{kg CO}_2 \, \text{m}^{-2} \, \text{week}^{-1}$ and go up to $3.4 \, \text{kg CO}_2 \, \text{m}^{-2} \, \text{week}^{-1}$. The mean went down to $0.004 \, \text{kg}$

$\text{CO}_2 \, \text{m}^{-2} \, \text{week}^{-1}$ during the summer months, when domestic heating and lighting fuel usage is lower. The largest fossil fuel emission estimated was located towards the north of the city, and corresponded to a crude oil refinery. Most point estimates were located on the outskirts of the city, with a few located within the central peninsula area. The road network is apparent



in the figure of the prior fossil fuel fluxes displaying the corresponding transport emissions, and clearly illustrates the large contribution that road transport contributes to the overall $CO_2$ budget of CT.





**Figure 3.** Prior estimates for day (top left) and night (top right) of the mean fossil fuel emissions ($\mathrm{kg\ CO_2\ m^{-2}\ week^{-1}}$) and the corresponding uncertainties, expressed as standard deviations ($\mathrm{kg\ CO_2\ m^{-2}\ week^{-1}}$) (bottom row), for the month of March 2012. These estimates were derived from an inventory analysis for CT based on vehicle, aviation and shipping vessel count data, population census data, and fuel usage at industrial point sources. These prior estimates are provided at a resolution of $1\,\mathrm{km} \times 1\,\mathrm{km}$ and the extent of the grid is between $34.5°$ and $33.5°$ south and between $18.2°$ and $19.2°$ east.





Since we are solving for weekly, rather than daily fluxes, we are using a strong assumption that fossil fuel fluxes within the same week are 100% correlated. To allow the inversion to react to local conditions within a given week, no correlation is assumed between weekly fluxes. Since fossil fuel emissions are expected to be localised in space, we also assumed no spatial correlation in the increments of the fluxes due to fossil fuel emissions. Relying on a correlation length approach to describe

spatial correlation would be inadequate, as it could lead to correlation between locations with and without fossil fuel sources.

### 2.9.2  Biogenic fluxes

The uncertainty in the estimate of biogenic prior estimates was set at the absolute value of the net primary productivity (NPP) as produced by CABLE. This is a very large error relative to the prior estimate itself, but there is a great deal of uncertainty in both the productivity and respiration budgets for NEE. This approach is similar to that used by Chevallier et al. (2010). Rather

than assigning a fixed proportional error, we preferred to base the uncertainty on the underlying processes driving NEE, as particularly in semi-arid regions, such as those conditions found throughout South Africa, very small NEE fluxes can occur as a result of both large productivity and respiration fluxes. This is different to the approach used by Bréon et al. (2015), where an uncertainty level of 70% was assigned to biogenic emissions. In the CT situation, this would lead to unrealistically low estimates of the uncertainty in biogenic emissions.

We consider two parameterisations of the covariances between biogenic fluxes at different locations in space. For the reference inversion we assumed an isotropic Balgovind correlation model as used in Wu et al. (2013). As sensitivity analyses in a companion paper we considered no correlation in space as an alternative parameterisation. For the Balgovind correlation model, the off-diagonal covariance elements between two spatial biogenic sources $\mathbf{s_i}$ and $\mathbf{s_j}$ are calculated as:

$$\mathbf{C_{s_o}}(\mathbf{s_i}, \mathbf{s_j}) = \sigma_b^2 (1 + \frac{h}{L}) exp(-\frac{h}{L}) \tag{10}$$

The characteristic correlation length $L$ is assumed to be $1\,\mathrm{km}$ and $h$ is the spatial distance between $\mathbf{s_i}$ and $\mathbf{s_j}$. As for the fossil fuel sources, no correlation is assumed between weekly biogenic fluxes, since the inversion setup is already assuming that biogenic fluxes within the same week are 100% correlated.

Figure 4 shows the spatial distribution of the biogenic emissions and their uncertainties for the month of March 2012. Day-

time biogenic emissions range between $-0.19\,\mathrm{kg}\ CO_2\ \mathrm{m}^{-2}\ \mathrm{week}^{-1}$ and $0.04\,\mathrm{kg}\ CO_2\ \mathrm{m}^{-2}\ \mathrm{week}^{-1}$, concentrated over areas such as the Cape Point Nature Reserve and Kogelberg Nature Reserve, located near the Hangklip lighthouse. At night the emissions are between $0.0$ and $0.06\,\mathrm{kg}\ CO_2\ \mathrm{m}^{-2}\ \mathrm{week}^{-1}$. The uncertainties in the biogenic daytime emissions have a median of 192% of the estimated emission and a mean of 881%, which translates to standard deviations ranging between $0.00001\,\mathrm{kg}$ $CO_2\ \mathrm{m}^{-2}\ \mathrm{week}^{-1}$ (over the ocean) and $0.30\,\mathrm{kg}\ CO_2\ \mathrm{m}^{-2}\ \mathrm{week}^{-1}$. Depending on the month, the median uncertainty can

range between 170 to 200% of the estimated biogenic emission, and estimates of biogenic emissions range between $-0.22$ and $0.004\,\mathrm{kg}\ CO_2\ \mathrm{m}^{-2}\ \mathrm{week}^{-1}$ during the summer to $-0.11$ to $0.007\,\mathrm{kg}\ CO_2\ \mathrm{m}^{-2}\ \mathrm{week}^{-1}$ at mid winter. At night, when only the biogenic respiration process is in operation, the median uncertainty drops to 23% of the estimated emission and the mean down to 75%, which translates to standard deviations ranging between $0.000001$ and $0.006\,\mathrm{kg}\ CO_2\ \mathrm{m}^{-2}\ \mathrm{week}^{-1}$.





**Figure 4.** Prior estimates for day (top left) and night (top right) of biogenic emissions ($kg \ CO_2 \ m^{-2} \ week^{-1}$) and the corresponding uncertainties, expressed as standard deviations ($kg \ CO_2 \ m^{-2} \ week^{-1}$) (bottom row), during the month of March 2012. The prior estimates were obtained from the CABLE land-atmosphere exchange model, which was coupled to the regional climate model CCAM and run at a spatial resolution of $1 \ km \times 1 \ km$. The extent of the grid is between $34.5°$ and $33.5°$ south and between $18.2°$ and $19.2°$ east.





## 2.10 Observation errors - $C_c$

The observation errors contain both the measurement error (which are known to be in the order of 0.3 ppm) (Bréon et al., 2015; Wu et al., 2016) and the error associate with modelling the concentrations. The modelling errors result from several sources, including errors within the atmospheric transport model and aggregation errors which are due to smoothing emission estimates from localised sources within the spatial grids (Kaminski et al., 2001).

Several sources of error were considered for the observation error covariance matrix. Similar to the approach adopted in the optimal network design for South Africa (Nickless et al., 2015b), to account for transport modelling errors, an error of 2 ppm during the day and 4 ppm at night was assigned to each observation, so that night-time observations carried less influence in the inversion. We did not account any further for aggregation or representation errors as we did in the network design, as we were running the inversion at the same spatial scale as the transport model. We acknowledge that there are still large sources of aggregation error due to smoothing of point source emissions across a relatively large grid, although we have kept the grid as small as could be managed in the inversion. The aggregation and representation errors are therefore contained within the assigned 2 ppm or 4 ppm. These errors may seem smaller relative to city scale inversion conducted in the Northern Hemisphere. We justify the use of these values in our application since we are dealing with a much smaller city compared to the megacity applications, such as Paris and Indianapolis. The sources in this application would each result in smaller deviations from background compared to larger cities. Measurements of background $CO_2$ have also shown that $CO_2$ concentrations in the Southern Hemisphere have smaller standard deviations. For example, for the years 2012 to 2013 the standard deviation between the monthly $CO_2$ means for Mauna Loa GAW station in the Northern Hemisphere was 2.3 ppm (Tans and Keeling , 2016), whereas for the same time period at Cape Point the standard deviation between the monthly means was 1.6 ppm.

As our reference approach, where we wanted to be as inclusive as possible of all sources of observation error, we took into consideration that transport errors in the modelled $CO_2$ concentrations would be larger when the wind speed was lower (Bréon et al., 2015), and this would be compounded at night when the planetary boundary layer height was lower and less stable (Feng et al., 2016). Additional error ranging between 0 and 1 ppm was added to the daytime uncertainty of 2 ppm, linearly scaled depending on the wind speed, with 0 ppm added when wind speeds were high ($20\,\mathrm{m\,s^{-1}}$) and 1 ppm added when the wind speed was close to zero. At night the additional uncertainty ranged between 0 and 4 ppm.

We also considered the standard deviation of the measured $CO_2$ concentrations during each hour. It would be expected that if there was a large amount of variability between the instantaneous measurements at the site, it is likely that the atmospheric transport model would have a greater chance of making errors during this period. The variability of the observed $CO_2$ concentrations, which contributed towards the mean estimate of the $CO_2$ concentration for that hour, was added to the overall error. Therefore in the reference approach each hour has a customised observation error dependant on the prevailing conditions at the measurement site. Therefore the total observation error, as a variance, is given as:

$$C_c(i,i)^2 = E_{trans}^2 + E_{wind}^2 + E_{obs}^2 \qquad (11)$$



where $E_{trans}$ is the transport modelling error of 2 ppm during the day and 4 ppm during the night, $E_{wind}$ is the additional error due to the wind speed conditions which ranged between 0 and 1, and $E_{obs}$ is variance of the observed concentrations within that hour. A time series of the customised observation errors is provided in Figure 5.




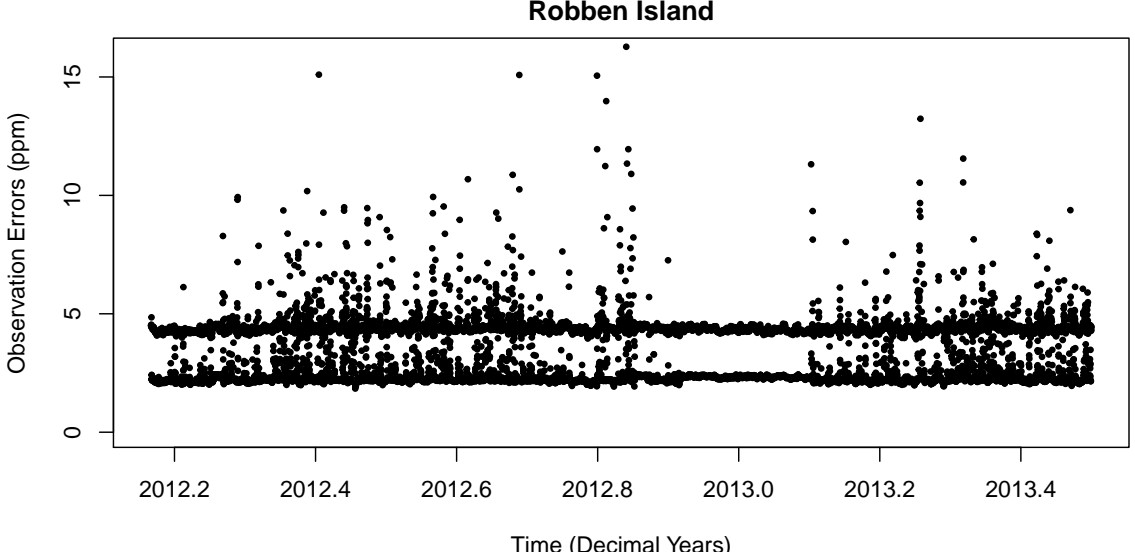

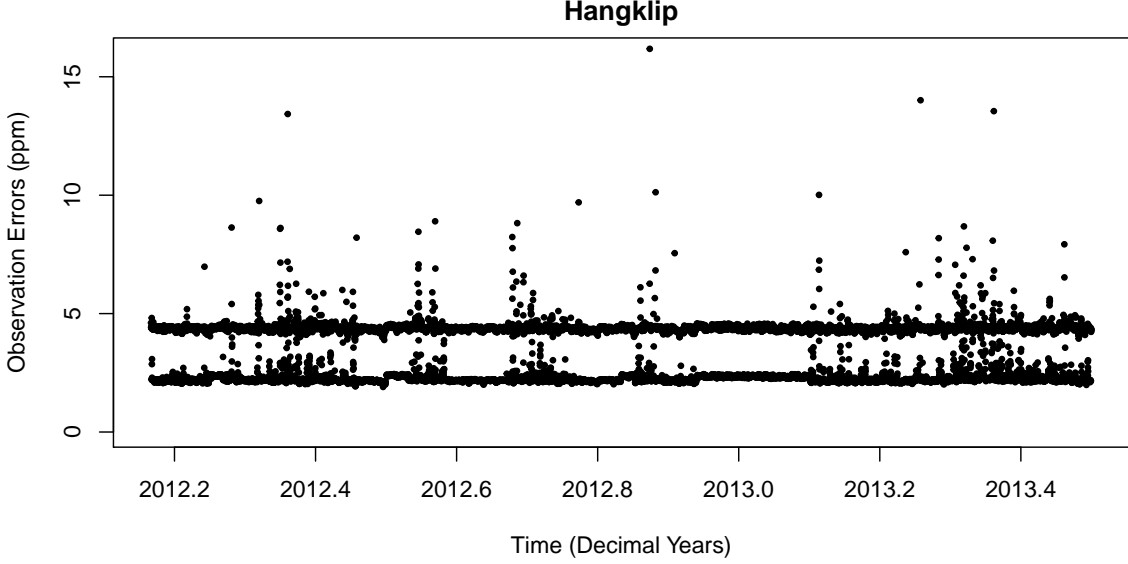

**Figure 5.** Time series of the customised observation errors (ppm) assigned to the $CO_2$ concentration measurement for each hour at the Robben Island and Hangklip measurement sites. The errors are made up of a modelling error component (set as 2 ppm during the day and 4 ppm at night), a wind speed component, and measurement variability component. The two distinct sets of points for each site arises due to the night-time modelling error component set to be always larger than the daytime component.





## 2.11 Model Assessment

In order to assess the appropriateness of the prior flux covariance parameters, the $\chi^2$ statistic, as described in Tarantola (2005) can be employed to determine the minimum value of the statistic:

$$\chi_1^2 = \frac{1}{\nu}(\mathbf{Hs_0} - \mathbf{c})^T (\mathbf{HC_{s_0}H}^T + \mathbf{C_c})^{-1}(\mathbf{Hs_0} - \mathbf{c}) \tag{12}$$

where $\nu$ is the dimension of the data space, in this case the length of observations in the inversion.

The squared residuals from the inversion (squared differences between observed and modelled concentrations) should follow the $\chi^2$ distribution with degrees of freedom equal to the number of observations (Michalak et al., 2005; Tarantola, 2005). Dividing this statistic by the degrees of freedom should yield a $\chi_1^2$ distribution. Values lower than one indicate that the variance is too large, and values greater than one indicate that the variance prescribed is lower than it should be and therefore the posterior estimates will be over-constrained by the prior fluxes. In order to ensure the suitability of the prior flux covariance matrix, the prior variances were scaled by a factor of two. This ensured that the $\chi_1^2$ statistic was close to a value of one for almost all months of the inversion. A single scaling factor was used to adjust all the prior flux variances. An alternative to a single value scaling factor will be considered in a subsequent paper.

Using the $\chi^2$ statistic to scale or estimate covariance parameters has been implemented by Lauvaux et al. (2016) and Michalak et al. (2005). Lauvaux et al. (2016) used the $\chi^2$ statistic to scale the elements of the observation error covariance matrix. As the estimation of the observation error covariance matrix is subject to a sensitivity analysis in the companion paper, the scaling was applied to the variance elements of the prior flux covariance matrix. Alternatively, a hierarchical Bayes approach can be used to estimate hyper-parameters for the covariance matrix, which are estimated based on the observed concentrations (Ganesan et al., 2014).

## 3 Results

The following section provides the results from the $CO_2$ concentration measurements obtained at the two sites over the sixteen month period, followed by the results of the reference inversion. The extent to which the atmospheric transport model was able to represent the wind conditions at the sites is described in the supplementary material (Supplementary material Section 1.1).

### 3.1 $CO_2$ Measurements

A single calibration standard was kept at each site, and run periodically in order to assess whether any drift was occurring in the $CO_2$ measurements over time, to determine if the calibration coefficients required any adjustment. At Robben Island the measurements of the calibration standard were (mean $\pm$ (standard deviation)) 386.89 ppm ($\pm$ 0.014) in November, 385.67 ppm ($\pm$ 0.012) in February 2013, and 385.73 ppm ($\pm$ 0.012) in June 2013. This indicates that the instrument was making stable measurements during the sixteen month campaign. The slightly higher reading in November 2012 occurred when the weather was wet, and there was more moisture contamination from the previous ambient measurements prior to calibration. At Hangklip





the instrument measured the calibration standard at 378.26 ppm ($\pm$ 0.009) in November 2012 and at 378.16 ppm ($\pm$ 0.022) in June 2013, indicating no evidence of drift over the measurement period.

In addition to each site's calibration standard, a travelling reference standard was also measured at close proximity in time at all three sites, including Cape Point. This standard consisted of clean air collected at the Cape Point site. The results of two

5  of these calibration measurements are presented in Table 1. By comparing calibration measurements with Cape Point inter-site differences are ensured to be negligible, and also links the Robben Island and Hangklip sites into the greater GAW network. The instruments at Cape Point are routinely calibrated with standards shared around the GAW network, which maintains high levels of quality control. Differences were found to be small between sites, and between calibration periods using the same standard in June 2012 and June 2013.



**Table 1.** Mean and standard deviation of $CO_2$ measurements (ppm) during calibration phase using a travelling $CO_2$ standard during June 2012 and June 2013. The Cape Point site is considered the gold standard as the instruments here are maintained to be within GAW limits.

| FA01830 | | Site | | |
| --- | --- | --- | --- | --- |
| Date | Cape Point | Robben Island | Hangklip | Maximum inter-site difference |
| June 2012 | 452.77 (0.03) | 452.88 (0.02) | 452.89 (0.03) | 0.12 |
| June 2013 | 452.28 (0.03) | 452.86 (0.02) | 452.60 (0.03) | 0.58 |
| June 2012 - June 2013 | 0.49 | 0.02 | 0.29 | |



The observed hourly $CO_2$ concentrations at the Robben Island and Hangklip sites are presented in Figure 6, together with the hourly measurements at Cape Point and the measured daily temperature at this site. There is no clear correlation between the peaks in $CO_2$ and daily temperature. From March 2012 until June 2013, the mean $CO_2$ concentration observed at Robben Island was 391.3 ppm ($\pm$ 5.02), usually ranging between 389.5 and 394.2 ppm, with a minimum of 382.4 ppm and a maximum of 445.0 ppm. The measurements at Hangklip had a similar mean of 390.6 ppm ($\pm$ 3.89), usually ranging between 389.5 and 391.4 ppm with a minimum of 380.4 ppm and a maximum of 430.6 ppm. The Cape Point measurements have a narrower range of 382.9 to 412.3 ppm, with a mean of 392.1 ppm, indicating less influence from local sources and sinks.



**Figure 6.** Observed hourly $CO_2$ concentrations (ppm) (left-side axis) at the Robben Island (top closed red circles) and Hangklip (bottom closed black circles) measurement sites. The blue line appearing at the bottom of each plot is the $CO_2$ concentration measurements at Cape Point station (ppm) and the green line at the top of each plot is the mean daily temperature (°C) as measured at the Cape Point station, which is represented by the right-side axis.





The mean diurnal cycle for each month is presented in Figure 7. Across all months, the diurnal cycle of $CO_2$ concentrations at Cape Point are relatively flat compared to Robben Island and Hangklip. In November 2012 and February 2013 the diurnal cycle for both measurement sites was the most flat. This is the summer period, when temperatures are high and the Western Cape experiences the lowest amount of rain. The amplitude of the diurnal cycle at both sites increased from April, reaching a

5    maximum amplitude in June and July. This is during the winter rainfall period in the Western Cape. Temperatures are mild and much of the vegetation growth occurs during this period. The diurnal cycle of the Hangklip site dipped below both Cape Point and Robben Island, indicating that this site is more affected by local sinks of $CO_2$. Robben Island consistently had the highest peaks in $CO_2$ concentrations across all months, indicating that this site was the most affected by local sources, which is what we expected.





**Figure 7.** Diurnal cycle of the observed $CO_2$ concentrations (ppm) for each month and at each site with standard error bars, where the standard error is calculated over all measurements available for that hour of the day during that particular month. Cape Point is the generally flat diurnal cycle in blue, Robben Island with the generally larger daytime $CO_2$ concentrations in red, and Hangklip with the generally lower afternoon $CO_2$ concentrations in black.



The mean diurnal cycle over the whole measurement period is presented in Figure 8, separated by site and by working week and weekend. The background site, Cape Point, shows no discernible difference between the mean concentrations over the week and weekend, whereas Robben Island and Hangklip sites measure concentrations during the working week which tend to be larger across most of the day. Both the early morning and afternoon means show a clear tendency for these sites to have

5  larger concentrations during the working week compared to the same time of day over the weekend, which can only be due to anthropogenic influences. This supports the separation of fossil fuel fluxes into working week and weekend contributions.





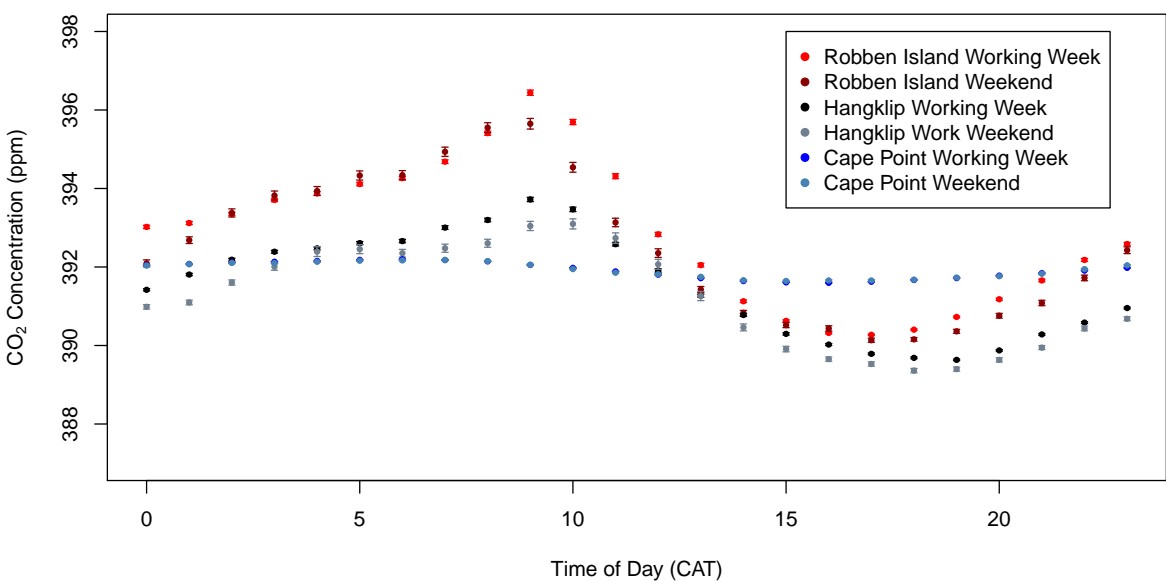

**Figure 8.** Diurnal cycle of the observed $CO_2$ concentrations (ppm) over the full measurement period from March 2012 until June 2013 at each site with standard error bars, where the standard error is calculated over all measurements available for that hour of the day during the entire measurement period, separated by site (Cape Point - blue and light blue, Robben Island - red and dark red, Hangklip - black and grey) and by working week (brighter colour) and weekend (duller colour).





Table 2 provides the measured $CO_2$ concentration availability at each site. Robben Island had slightly higher data availability compared to Hangklip, with a 65.6% data availability overall for the sixteen month measurement campaign. Each site was equipped with a Hauwei USB modem connected to a 3G network, and were set to submit data to an email address on a regular basis. Through these emails or through connecting remotely to the instruments, instrumentation problems could be detected.

5   Most of the down-time at the sites was attributed to either pump failure, or occasionally the instrument software had failed and there was no available person to restart the instrument due to limited access to these sites. Robben Island lighthouse is manned, and therefore it was possible to request the lighthouse keeper to restart the instrument when frozen, but more regular than expected pump problems required visits to the site to replace the offending device. This entailed making arrangements for unplanned voyages to Robben Island, which could take some time to arrange. Hangklip is unmanned, and the site has strict

10   access control, therefore problems at this site tended to take slightly longer to remedy. The final four months considered in this study had the best continuity of data availability.





**Table 2.** The number of days with available $CO_2$ measurement data for each month (out of a possible four week period considered) and overall percentage available data out of 16 four-week periods for each site.

| Site | \multicolumn{10}{c}{Year / Month} | | | | | | | | Overall Percentage |
|---|---|---|---|---|---|---|---|---|---|---|---|---|---|---|---|---|
| | \multicolumn{10}{c}{2012} | | | | | | | \multicolumn{6}{c}{2013} | | | | | |
| | 3 | 4 | 5 | 6 | 7 | 8 | 9 | 10 | 11 | 12 | 1 | 2 | 3 | 4 | 5 | 6 | |
| Robben Island | 28 | 21 | 28 | 14 | 14 | 28 | 28 | 0 | 14 | 0 | 0 | 21 | 28 | 28 | 28 | 28 | 68.75 |
| Hangklip | 28 | 21 | 28 | 14 | 14 | 0 | 28 | 0 | 14 | 0 | 0 | 21 | 28 | 28 | 28 | 28 | 62.50 |
| Combined | 56 | 42 | 56 | 28 | 28 | 28 | 56 | 0 | 28 | 0 | 0 | 42 | 56 | 56 | 56 | 56 | 65.63 |





## 3.2 Reference Inversion

The inversion configuration considered as the reference inversion in this study consisted of the most comprehensive inclusion of information available. Here full monthly inversions were performed solving for four separate weekly fluxes, decomposed into working week and weekend fossil fuel contributions, full week NEE contributions, each separated into day and night

contributions. The observation error covariance matrix consisted of variance elements which have been estimated based on wind speed, whether day or night, and on the concentration measurement variance, and temporal correlation, with a one hour correlation length, was included between measurements from the same site. The variance elements of the prior flux covariance matrix were estimated from the fossil fuel inventory analysis and net ecosystem productivity (NEP) estimates from CABLE, with correlation between NEE estimates from adjacent pixels.

### 3.2.1 Modelled Concentrations

The time series of the posterior modelled concentrations at Robben Island were compared to the observed concentrations and the prior modelled concentrations (Figures 9 a and b). The prior estimates tended to be in the correct range for $CO_2$ concentration measurements, but could be higher or lower compared to the observations by as much as 100 ppm. The agreement between the posterior and observed concentrations was much stronger compared to the prior estimates, with posterior estimates

showing the inversion's ability to track observed concentrations during localised events. The agreement can be assessed by means of the intraclass correlation coefficient (ICC) (Shrout and Fleiss, 1979). Values close to zero indicate poor agreement while values close to one indicate strong agreement. The ICC is low at 0.03 (95% CI: 0.01 to 0.06), but still significant, between the observed and prior modelled concentrations, but goes up to 0.59 (95% CI: 0.57 to 0.61) between the observed and posterior modelled concentrations.

A time series plot of the prior and posterior residuals is given in Figures 9 c and d. The prior residuals could be very large in either the positive or negative direction, up to 100 ppm and occasionally out by as much as 200 ppm. The posterior residuals were much closer to the zero line, with the highest deviation equal to 33 ppm. Summary statistics of the residuals indicate that the mean bias in the prior estimates was -2.9 ppm with standard deviation equal to 21.4 ppm and interquartile range between -9.1 and 3.7 ppm. From the residuals of the posterior estimates, the bias was reduced to 0.5 ppm, the standard

deviation of residuals to 3.9 ppm, and interquartile range to between -1.5 and 1.5 ppm, showing a significant improvement over the prior estimates. The standard deviation of the posterior residuals was lower by 1.1 ppm compared with that of observed concentrations, which was 5.02 ppm.

The time series of the observed, prior and posterior concentrations at Hangklip reveal a similar result compared to Robben Island (Figures 10 a and b). The prior estimates can be much larger or smaller compared to the observed concentrations, but

the posterior concentration estimates match much more closely with the observed concentrations, also showing ability to track local events at the site, even more closely compared to the Robben Island posterior concentrations. The ICC between the observed and prior modelled concentrations was similar to Robben Island at 0.03 (95% CI: 0.003 to 0.05), but shows much better agreement between the observed and posterior modelled concentrations with an ICC of 0.76 (95% CI: 0.75 to 0.77).



The prior residuals at the Hangklip site tended to be less extreme compared to Robben Island, with a maximum deviation of 117 ppm in either direction (Figures 10 c and d). The summary statistics of the residuals indicate that the mean bias in the prior estimates was 2.4 ppm with standard deviation equal to 17.6 and interquartile range between -2.3 and 6.5. For the posterior residuals, the bias was reduced to 0.04 ppm with standard deviation equal to 2.46 (interquartile range -1.1 to 0.8), the standard deviation lower by 1.4 ppm compared with the standard deviation of the observed concentrations, which was 3.89 ppm.

The large improvement in the representativeness of the posterior concentrations in relation to the observed concentrations at both sites lends confidence to the reference inversion's ability to improve estimates of the sources to better match observed conditions in the region.





**Figure 9.** A time series over the sixteen month measurement campaign of (a) the observed and prior modelled $CO_2$ concentrations (ppm) from the reference inversion at Robben Island, (b) the observed and posterior modelled $CO_2$ concentrations at Robben Island, (c) the prior residuals (ppm), defined as the difference between the observed and prior modelled concentrations, and (d) the posterior residuals, defined as the difference between the observed and posterior modelled concentrations. Closed black circles - observed concentration; open red triangles - prior concentrations; purple crosses - posterior concentrations; open red inverted triangles - prior residuals; open purple circles - posterior residuals.







**Figure 10.** A time series over the sixteen month measurement campaign of (a) the observed and prior modelled $CO_2$ concentrations (ppm) from the reference inversion at Hangklip, (b) the observed and posterior modelled $CO_2$ concentrations at Hangklip, (c) the prior residuals (ppm), defined as the difference between the observed and prior modelled concentrations, and (d) the posterior residuals, defined as the difference between the observed and posterior modelled concentrations. Closed black circles - observed concentration; open red triangles - prior concentrations; purple crosses - posterior concentrations; open red inverted triangles - prior residuals; open purple circles - posterior residuals.





The mean working week and weekend diurnal cycles in the observed, prior and posterior modelled concentrations are shown for each site and for each month in the supplementary material (Supplementary Material Section 1.2). Figure 11 provides the mean working week and weekend diurnal cycle over the full measurement period. For Robben Island, the mean concentrations for each hour indicate that the emissions are overestimated by the prior estimates, therefore producing larger concentrations

compared to the observed. The posterior modelled concentrations are all much closer to the observed concentrations, reproducing the peak in concentrations to be between 8:00 and 9:00 in the morning and the trough in concentrations to occur between 15:00 and 18:00. Overall the cycle in the posterior concentrations is more flat compared to that of the observed concentrations. The mean observed concentrations for each hour indicates that for most hours of the day, the concentrations during the week are usually slightly higher compared to those over the weekend. The posterior estimates show a smaller deviation between the

week and weekend concentrations at each hour of the day, particularly around mid-morning, compared to the observed week and weekend concentrations.

    The prior estimates for the Hangklip measurement show the opposite bias compared with Robben Island, with prior modelled concentrations lower at each hour compared to the observed concentrations. The posterior modelled concentrations for Hangklip overlap closely with the observed concentrations. Compared with Robben Island, there is slightly less separation

between the working week and weekend concentrations at each hour. This should be expected as the concentrations observed at the Hangklip site are more dominated by biogenic sources compared to Robben Island. The closest fossil fuel sources are those from transport and domestic emissions. The main road through this area carries a large amount of commercial traffic during the week, and over the weekend the area is frequented by weekend residents and tourists, and therefore anthropogenic activity is not expected to be much lower over the weekend. The posterior concentrations show that the inversion has managed

to replicate the separation between the mean hourly working week and weekend concentrations shown by the observed concentrations for most hours of the day, particularly the difference between these concentrations occurring between the hours of 7:00 and 9:00.



**Figure 11.** The hourly diurnal cycle (mean estimates for each hour with standard error bar) in the observed, prior and posterior modelled $CO_2$ concentrations (ppm) over the full measurement period from March 2012 until June 2013, separated by working week and weekend, and plotted separately for Robben Island (top) and Hangklip (bottom) measurement sites. The diurnal plots are separated into working week and weekend observed concentrations (black and grey), working week and weekend prior modelled concentrations (red and dark red), and working week and weekend posterior modelled concentrations (dark and light purple).





The inversion-corrected background $CO_2$ concentrations are obtained from the posterior source estimates. Figure 12 provides a time series of the prior and posterior estimates. The north and east border are terrestrial, whereas the ocean borders the south and west. For all borders across the total measurement period, the inversion has generally had very small innovations, with the posterior estimates remaining within the limits of the prior concentrations. Only the northern and eastern terrestrial

5   borders show some deviations from the priors between May and June 2012, and between March and April 2013. As these are the terrestrial borders, it would be expected that if the inversion needed to correct for unknown sources outside of the domain, it would be at these border locations, whereas we expect the ocean borders to have background concentrations very close to those provided by the Cape Point background measurement site.





**Figure 12.** Time series with 95% confidence interval (represented by the shaded area) of the prior (black line) and posterior estimates of the background $CO_2$ concentrations (ppm) (north - green, east - yellow, south - red, west - blue). The prior estimates are the same for each cardinal direction, and are obtained from the Cape Point percentile-filtered observations. The posterior estimates for the concentrations are solved for as additional unknowns in the reference inversion.





### 3.2.2 Weekly flux Estimates

The innovation of the inversion can be observed through the differences between the prior and posterior flux estimates, which we refer to as the innovations, and through the change in the uncertainty estimates. Positive innovations indicate that the prior estimates were too large, and that the estimate should be more in the negative direction, whereas negative innovations indicate

that prior estimates were set too small and should be more in the positive direction. Figure 13 shows the changes in total flux estimates for each pixel in $kg\ CO_2 m^{-2}\ week^{-1}$ for the month of May 2012, as well as the percentage change in the flux uncertainty, the percentage change in the fossil fuel emissions and change in NEE fluxes. The mean total weekly flux of a pixel was obtained by summing together all of the flux components of the state vector and dividing by the number of weeks, and the innovations calculated from these mean weekly fluxes. The uncertainties for each pixel are represented by standard deviations

which have been derived from the variance estimates for the weekly fluxes for each pixel over the month of May 2012.

   May falls within the winter rainfall season of the Western Cape region. The innovations indicate that the total flux for the pixel over the petrol refinery, which had the largest prior flux estimate, was overestimated by the prior ($9.43\ kg\ CO_2 m^{-2}\ week^{-1}$) relative to the posterior estimate ($6.62\ kg\ CO_2 m^{-2}\ week^{-1}$) by an amount of $2.81\ kg\ CO_2 m^{-2}\ week^{-1}$ (Fig. 13a). Adjustments were generally small ranging between -0.001 and $0.003\ kg\ CO_2 m^{-2}\ week^{-1}$. The area around the natural reserves, such

as Cape Point and Kogelberg Nature Reserves, had adjustments by the inversion that were close to zero or slightly negative, indicating that the amount of carbon uptake in these regions was overestimated by the CABLE model. The largest negative adjustment by the inversion was $-0.08\ kg\ CO_2 m^{-2}\ week^{-1}$, from -0.03 up to $-0.04\ kg\ CO_2 m^{-2}\ week^{-1}$, over a pixel in the Cape Point Nature Reserve. With respect to the rest of the domain, excluding the crude oil refinery, the largest adjustments to the prior fluxes were made over the CBD area to the south east of Robben Island.

In terms of the percentage standard deviation reduction (Fig. 13b), the largest reductions occurred over the natural areas, particularly Cape Point to the south of Robben Island, where the posterior uncertainty over the area was significantly lower, by over 50%, compared to the prior uncertainties. Significant reductions are also shown over largely agricultural areas to the north of the CBD region. Over the CBD area itself, the reductions were present, reaching values of close to 60% over a few central CBD pixels, but generally smaller compared to the uncertainty reductions over natural areas in the domain which reached

levels as high as 92%.

   The sources were split into those which originate from fossil fuels and those from biogenic sources. The percentage change in the fossil fuel estimates, from prior to posterior, indicate the changes tended to be small, except on Robben Island itself, where percentage changes were up to 75% indicating that the emissions on the island were significantly reduced by the inversion (Fig. 13c). There were a few pixels which had negative change, indicating that the inversion increased fossil fuel emissions,

just north of the island. Located on the north eastern shore of Robben Island is a diesel-fuelled power generation plant, as well as desalination plant which is powered by this station. Increases in the fossil fuel fluxes may be due to emissions arising from these activities which have not been accounted for in the inventory analysis. In the inventory analysis there was no fuel information available for any industrial sources on Robben Island, but fossil fuel emissions were included due to domestic and transport sources, therefore these could have been adjusted by the inversion. The inventory analysis does not take into



account explicitly the shipping routes going into CT harbour, or into Robben Island harbour, but rather all the emissions are concentrated within CT harbour, where the shipping information is available. This could also result in the inversion adjusting emissions on the island to deal with near shipping related emissions. There was a region in the Western Cape interior to the east of Robben Island which had slightly increased fossil fuel emissions. The inversion had the effect of mainly reducing fossil fuel

emissions along the south east transect extending from Robben Island over the CBD towards Hangklip, or leaving the emission unchanged.

The inversion had a much larger impact on the terrestrial biogenic fluxes (Fig. 13d). This is unsurprising, as the relative uncertainties of the prior biogenic fluxes were much larger compared with those of the prior fossil fuel estimates. The area of the domain experiencing innovation from the inversion was also much more widespread compared to the fossil fuel emissions.

This is in part due to the correlation specified between the biogenic fluxes of adjacent pixels, but not between their fossil fuel fluxes. The majority of the innovation over the domain was close to zero, between -0.02 and $0.02\,\mathrm{kg}\,\mathrm{CO_2}\mathrm{m}^{-2}\,\mathrm{week}^{-1}$, indicating that the inversion was making small absolute changes to the biogenic flux estimates. Over the CBD region, the changes were the largest, up to $0.32\,\mathrm{kg}\,\mathrm{CO_2}\mathrm{m}^{-2}\,\mathrm{week}^{-1}$ and these differences were positive indicating that the posterior estimates were smaller and therefore the inversion was acting to reduce emissions over the CBD region relative to the prior

estimates, through changes to the biogenic fluxes. The natural region around Cape Point and within the Kogelberg Nature Reserve north of Hangklip showed slightly negative changes in the biogenic fluxes, indicating that the drawdown of $CO_2$ was reduced by the inversion (i.e. the carbon uptake was less than predicted by CABLE) making the overall flux in these regions more positive towards the atmosphere.



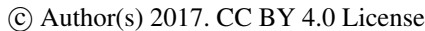


**Figure 13.** (a) Differences between the prior and posterior total flux estimates (kg $CO_2 m^{-2}$ week$^{-1}$) for May 2012 (prior - posterior). (b) Percentage reduction in the standard deviation of the flux estimate from prior to posterior (calculated as (1 - posterior uncertainty / prior uncertainty)×100). (c) Percentage reduction in the fossil fuel emission estimates from prior to posterior (calculated as (1 - posterior estimate / prior estimate)×100) where estimates were in kg $CO_2 m^{-2}$ week$^{-1}$. (d) Differences in the biogenic flux estimates between prior and posterior estimates (prior - posterior) (kg $CO_2 m^{-2}$ week$^{-1}$), with negative values indicating larger posterior estimates of $CO_2$ fluxes relative to the prior estimates. Extent: between $34.5°$ and $33.5°$ south and between $18.2°$ and $19.2°$ east.



September is the beginning of the summer months in the Western Cape region, when temperatures start to rise, and the mean monthly rainfall reduces. The innovations across the region during this month were dominated by negative values with the majority ranging between -0.03 and -0.01 $\mathrm{kg\,CO_2\,m^{-2}\,week^{-1}}$, indicating that the inversion increased emissions of $CO_2$ over the majority of the domain, with a maximum reduction of 2.1 $\mathrm{kg\,CO_2\,m^{-2}\,week^{-1}}$ over the crude oil refinery site, from

9.4 down to 7.2 $\mathrm{kg\,CO_2\,m^{-2}\,week^{-1}}$, and a maximum increase in $CO_2$ fluxes of 0.2 $\mathrm{kg\,CO_2\,m^{-2}\,week^{-1}}$ over an agricultural region north east of the CBD region, from -0.10 up to 0.08 $\mathrm{kg\,CO_2\,m^{-2}\,week^{-1}}$, where mainly vineyards are located (Fig. 14a). There are a further five areas where the inversion had similar sized increases in the total emissions. One lies further inland to the east of the area with largest increase in emissions. Three more regions lie to the north, north east and east of the Hangklip site. These regions are all within the Hottentots-Holland Mountain catchment area which is largely dominated by

vineyard agriculture. The fifth area is located within the Kogelberg Nature Reserve near the Hangklip site. The CBD area of CT had very small reductions in emissions or the posterior emissions remained similar to the priors.

Reductions in the total pixel standard deviation ranged mainly between 2.3 and 18.6%, with a maximum reduction of 88.5% (Fig. 14b). The largest uncertainty reductions induced by the inversion occurred over the natural areas bordering on the CBD, particularly over the Table Mountain National Park, as well as the natural areas surrounding the Hangklip site. The areas to the

east of Robben Island over the Durbanville and Bellville townships, which consists of a mix of residential suburbs, vineyard agricultural areas and industrial areas also showed reductions in the uncertainties of the fluxes.

Adjustments made to the fossil fuel fluxes were mainly made on the transect of the city running between Robben Island and Hangklip, as well as to fossil fuel emissions on Robben Island itself, similar to the month of May. The maximum percentage adjustment to the fossil fuel fluxes was 51.1%, and the mean adjustment close to zero, with almost all adjustments positive,

indicating that the posterior estimates were smaller relative to the priors. Robben Island itself showed a mixed of positive and negative adjustments, with posterior fluxes larger than the priors on the west of the island but smaller than the priors on the east of the island.

The biogenic flux adjustments by the inversion consisted of reductions in emissions over the CBD region, and increases in $CO_2$ emissions relative to the prior estimates over the Table Mountain region, over the agricultural area to the north of

the CBD and over the natural regions near Hangklip and generally to the eastern region of the domain. The fluxes over the CBD were decreased by up to 0.21 $\mathrm{kg\,CO_2\,m^{-2}\,week^{-1}}$ through adjustments to the biogenic fluxes. The areas to the north of the domain which had prior NEE estimates between -0.20 and -0.10 $\mathrm{kg\,CO_2\,m^{-2}\,week^{-1}}$ were made more positive, with posterior estimates ranging between -0.10 and 0.10 $\mathrm{kg\,CO_2\,m^{-2}\,week^{-1}}$, where the largest changes were over the vineyard agricultural areas. The natural area surrounding Hangklip site showed negative differences between the prior and posterior

estimates, indicating that the inversion corrected the negative prior biogenic fluxes by making these more positive, and so the CABLE model appears to have overestimated the amount of $CO_2$ drawdown during this period.





**Figure 14.** (a) Differences between the prior and posterior total flux estimates (kg $CO_2$ m$^{-2}$ week$^{-1}$) for September 2012 (prior - posterior). (b) Percentage reduction in the standard deviation of the flux estimate from prior to posterior (calculated as (1 - posterior uncertainty / prior uncertainty)×100). (c) Percentage reduction in the fossil fuel emission estimates from prior to posterior (calculated as (1 - posterior estimate / prior estimate)×100) where estimates were in kg $CO_2$ m$^{-2}$ week$^{-1}$. (d) Differences in the biogenic flux estimates between prior and posterior estimates (prior - posterior) (kg $CO_2$ m$^{-2}$ week$^{-1}$), with negative values indicating larger posterior estimates of $CO_2$ fluxes relative to the prior estimates. Extent: between 34.5° and 33.5° south and between 18.2° and 19.2° east.



Table 3 presents summary statistics of the pixel-level weekly fluxes over the domain for each month, and the supplementary material (Supplementary Material Section 1.3) displays the spatial extent of the adjustments made to prior flux estimates by the inversion, as well as the uncertainty reductions, for each month. The limits of the range for the posterior pixel-level weekly fluxes are more negative compared with those of the prior estimates, indicating that for all the months, the inversion tended to

reduce the emission of $CO_2$, and is also evident from the maps of change in total flux estimates between the prior and posterior estimates. Specifically, the inversion tended to reduce the fossil fuel emissions, evident from the smaller mean flux value over the domain. The variability between pixels was also reduced. The pixel-level natural fluxes, although generally smaller in magnitude and range, were the most altered by the inversion, where these changes were related to the season in which the month fell. For most months, the biogenic fluxes became more positive, indicating less uptake of $CO_2$ than predicted by the CABLE

model. But for the months of June and July, which occur mid-winter, the mean NEE was made more negative. The minimum values were also at least twice the minimum value of the prior estimates for all months, and this is also evident from the maps of change in NEE from prior to posterior, which show that the inversion is reducing the $CO_2$ flux towards the atmosphere over the CBD region by altering the biogenic fluxes within those pixels. For most months the percentage uncertainty reduction for the total flux reached over 90% for at least some of the pixels, with a maximum uncertainty reduction over a pixel of 97.7% in

March 2012. The lowest maximum reduction in the uncertainty of the total flux in a pixel in a month was 68.2% in June 2012.





**Table 3.** Summary statistics of the pixel-level weekly prior and posterior flux estimates (kg $CO_2 m^{-2}$ week$^{-1}$) for each month. These are summary statistics calculated over all pixels in the domain of the average total flux within a pixel.

| Month | Total Weekly Flux kg $CO_2 m^{-2}$ week$^{-1}$ | | Total Weekly Fossil Fuel Flux kg $CO_2 m^{-2}$ week$^{-1}$ | | Total Weekly NEE Flux kg $CO_2 m^{-2}$ week$^{-1}$ | |
|---|---|---|---|---|---|---|
| | Prior mean (sd) (min; max) | Posterior mean (sd) (min; max) | Prior mean (sd) (min; max) | Posterior mean (sd) (min; max) | Prior mean (sd) (min; max) | Posterior mean (sd) (min; max) |
| Mar2012 | 0.0077 (0.1123) (-0.1755; 9.4523) | 0.0102 (0.1059) (-0.2089; 9.2106) | 0.0144 (0.1083) (0; 9.4510) | 0.0139 (0.1053) (0; 9.2408) | -0.0066 (0.0248) (-0.1762; 0.0958) | -0.0037 (0.0342) (-0.4411; 0.2984) |
| Apr2012 | -0.0004 (0.1114) (-0.1283; 9.4345) | 0.0015 (0.0879) (-0.4673; 6.3828) | 0.0143 (0.1081) (0; 9.4510) | 0.0137 (0.0829) (0; 6.5232) | -0.0147 (0.0249) (-0.1283; 0.056) | -0.0122 (0.0526) (-0.4673; 0.2463) |
| May2012 | 0.0021 (0.1094) (-0.0932; 9.4318) | 0.0011 (0.0536) (-0.3663; 2.0896) | 0.0143 (0.108) (0; 9.4510) | 0.0132 (0.0557) (0; 2.1800) | -0.0123 (0.0174) (-0.0932; 0.0479) | -0.0121 (0.0435) (-0.5605; 0.1665) |
| Jun2012 | 0.0073 (0.1092) (-0.0694; 9.4413) | 0.0024 (0.0544) (-0.2792; 2.9713) | 0.0143 (0.1079) (0; 9.4510) | 0.0134 (0.0601) (0; 3.0194) | -0.0071 (0.0142) (-0.0694; 0.0581) | -0.011 (0.0332) (-0.5741; 0.1437) |
| Jul2012 | 0.0023 (0.1100) (-0.0792; 9.4278) | 0 (0.0665) (-0.2933; 4.5750) | 0.0143 (0.1079) (0; 9.4510) | 0.0136 (0.0702) (0; 4.6980) | -0.012 (0.0192) (-0.0792; 0.0634) | -0.0136 (0.0420) (-0.5992; 0.3387) |
| Aug2012 | -0.0091 (0.1116) (-0.1200; 9.4124) | -0.0013 (0.1021) (-0.2796; 8.4318) | 0.0143 (0.1079) (0; 9.4510) | 0.0141 (0.1000) (0; 8.5457) | -0.0234 (0.0281) (-0.1200; 0.0480) | -0.0154 (0.0398) (-0.4306; 0.4391) |
| Sep2012 | -0.0235 (0.1094) (-0.1518; 9.3803) | -0.0016 (0.0575) (-0.3355; 2.6130) | 0.0103 (0.1030) (0; 9.4364) | 0.0096 (0.0517) (0; 2.9473) | -0.0338 (0.0369) (-0.1518; 0.0409) | -0.0112 (0.0469) (-0.3532; 0.2815) |
| Nov2012 | -0.0267 (0.1104) (-0.2165; 9.3833) | -0.0146 (0.0572) (-0.2834; 2.7560) | 0.0104 (0.1032) (0; 9.4364) | 0.0096 (0.0522) (0; 2.9546) | -0.037 (0.0402) (-0.2231; 0.0229) | -0.0242 (0.0408) (-0.3178; 0.1280) |
| Feb2013 | 0.0029 (0.1063) (-0.1566; 9.4313) | 0.0065 (0.0988) (-0.2195; 8.7948) | 0.0104 (0.1034) (0; 9.4364) | 0.0101 (0.0971) (0; 8.7996) | -0.0076 (0.0226) (-0.1569; 0.1050) | -0.0036 (0.0317) (-0.2436; 0.1736) |
| Mar2013 | 0.0088 (0.1109) (-0.1404; 9.4452) | 0.0057 (0.0513) (-0.2728; 2.1595) | 0.0143 (0.1081) (0; 9.451) | 0.0132 (0.0538) (0; 2.1832) | -0.0055 (0.0213) (-0.1412; 0.0919) | -0.0075 (0.0400) (-0.3941; 0.1846) |
| Apr2013 | -0.0002 (0.1112) (-0.1512; 9.4329) | 0.0018 (0.0579) (-0.2732; 2.8216) | 0.0143 (0.108) (0; 9.451) | 0.0133 (0.0593) (-0.0003; 3.0344) | -0.0146 (0.0256) (-0.1512; 0.057) | -0.0115 (0.0493) (-0.4285; 0.2372) |
| May2013 | 0 (0.1094) (-0.0939; 9.4269) | 0.0017 (0.0636) (-0.3835; 4.0001) | 0.0143 (0.1079) (0; 9.451) | 0.0134 (0.0662) (0; 4.1977) | -0.0143 (0.0194) (-0.0979; 0.043) | -0.0116 (0.0511) (-0.4977; 0.2226) |
| Jun2013 | -0.0032 (0.1099) (-0.0993; 9.4218) | 0.0023 (0.0552) (-0.3192; 2.0915) | 0.0143 (0.1078) (0; 9.451) | 0.0131 (0.0542) (0; 2.1748) | -0.0175 (0.0217) (-0.0993; 0.0515) | -0.0108 (0.0502) (-0.4197; 0.2218) |





An example pixel, located near the CBD sources, was selected in order to investigate posterior flux covariances estimated by the inversion. For a given week, each pixel is composed of potentially six sources: working week and weekend fossil fuel sources, both day and night, and the day and night biogenic sources; and each of these sources could have a non-zero covariance term between itself and the same source but from surrounding pixels, or with one of the other five sources from the same pixel or from surrounding pixels. The posterior covariance terms are determined by the product of the sensitivity matrix and observation error covariance matrix, and by the prior flux covariance matrix. Once the covariances are available, the correlations can be calculated between the reference source and all other sources. Strong correlations on uncertainties indicate that the two parameters have only been resolved as some linear combination of the two (Tarantola, 2005).

The posterior covariances between the day working week flux of the reference pixel and other sources are only notably different from zero for working week daytime fossil fuel fluxes, working week night-time fossil fuel fluxes, and the daytime biogenic fluxes. The covariances between the daytime working week fluxes reveal that non-zero covariances do not necessarily have to be close in proximity to the reference pixel, and negative and positive covariance pixels can cluster around each other. The covariances with the night-time fossil fuel fluxes were larger than those during the day, but were limited to a few pixels around the reference pixel. These covariances ranged between -0.15 and $0.09\,\mathrm{g^2\,CO_2\,m^{-4}\,week^{-2}}$. The non-zero covariances with biogenic sources were larger (between -1.50 and $0.88\,\mathrm{g^2\,CO_2\,m^{-4}\,week^{-2}}$) and fluctuated between patches of positive and negative values. Close to the CBD area there was a distinctive region of positive covariance between the fossil fuel source of the reference pixel and the biogenic fluxes from a region over the Table Mountain area and a negative covariance patch south of the CBD. The eastern terrestrial part of the domain had patches of positive and negative covariances between the fossil fuel source of the reference CBD pixel and the daytime biogenic fluxes. Considering the correlations, these are very small when comparing pixel to pixel correlations; no bigger than 0.001 in either direction. The sum of the covariances between the reference and all other sources equals $-25.8\,\mathrm{g^2\,CO_2\,m^{-4}\,week^{-2}}$. Therefore the covariances for this pixel would reduce the total variance for the total daytime fossil fuel source by 51.7, where the total variance for the fossil fuel source of this pixel was $233.7\,\mathrm{g^2\,CO_2\,m^{-4}\,week^{-2}}$. As the prior covariance matrix had no non-zero covariance terms between any of the fossil fuel sources, these covariances in the posterior covariance matrix for this source are all as a consequence of the projection of the observation error covariance matrix into the flux uncertainty space by the sensitivity matrix in the term $\mathbf{H}^T\mathbf{C_c}^{-1}\mathbf{H}$ of the solution for $\mathbf{C_s}$. The non-zero covariance terms indicate which of the sources could not be solved independently by the inversion, but where the inversion solved for a linear combination of these sources.



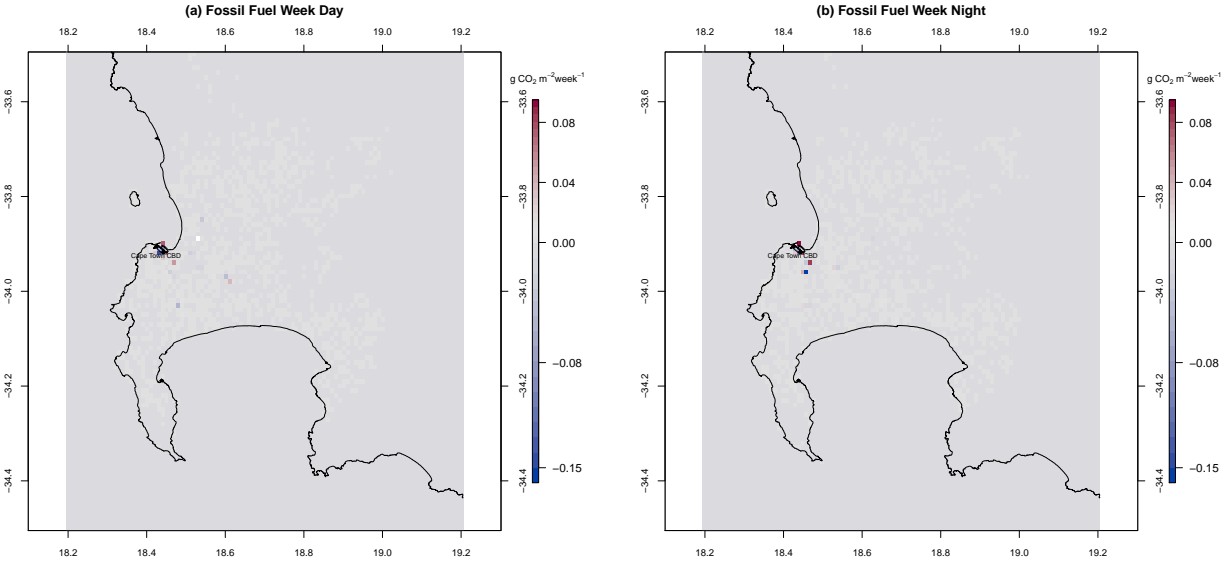

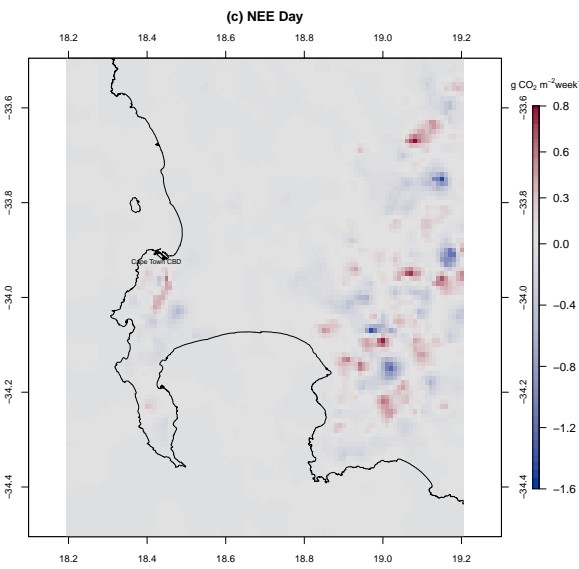

**Figure 15.** A single pixel was selected near the Cape Town CBD during the first week in March 2012. The posterior covariances were extracted for this pixel, and the covariances considered between the daytime working week fossil fuel flux within this pixel and all other sources (i.e. the remaining sources in this pixel, and all sources within all other pixels). The three plots are (a) covariances ($\mathrm{g^{-2}\,CO_2\,m^{-4}\,week^{-2}}$) between fossil fuel working week daytime source during the first week of March 2012 in the selected pixel and surrounding fossil fuel working week daytime sources from other pixels; (b) covariances between fossil fuel working week daytime source in the selected pixel and fossil fuel working week night-time sources within this pixel and all other pixels; (c) covariances between fossil fuel working week daytime source in the selected pixel and biogenic flux daytime sources within this pixel and all other pixels. Extent: between $34.5°$ and $33.5°$ south and between $18.2°$ and $19.2°$ east.



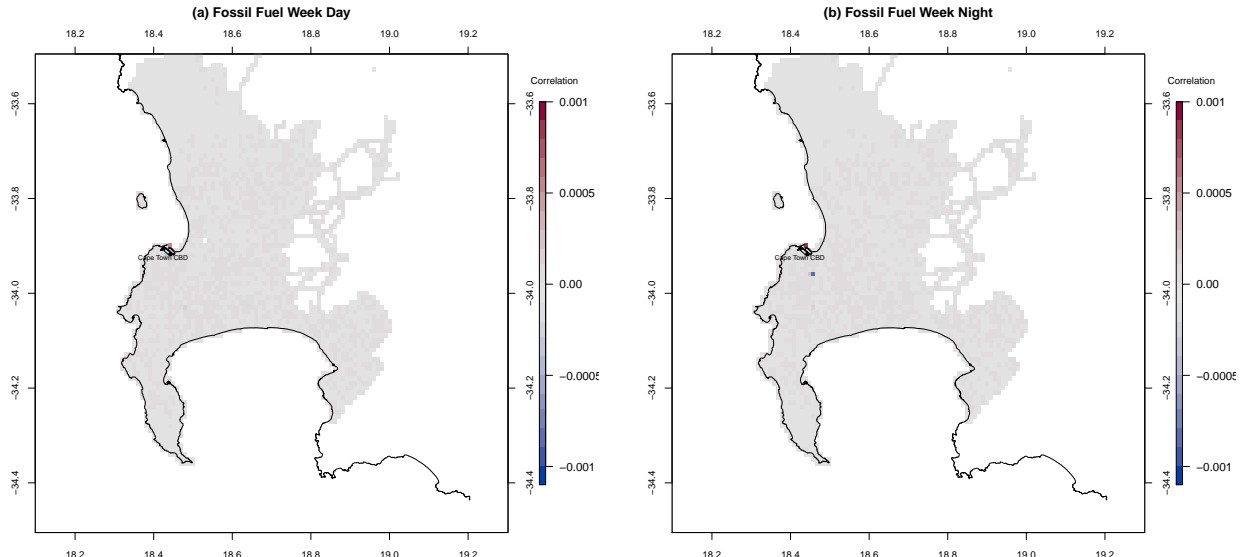

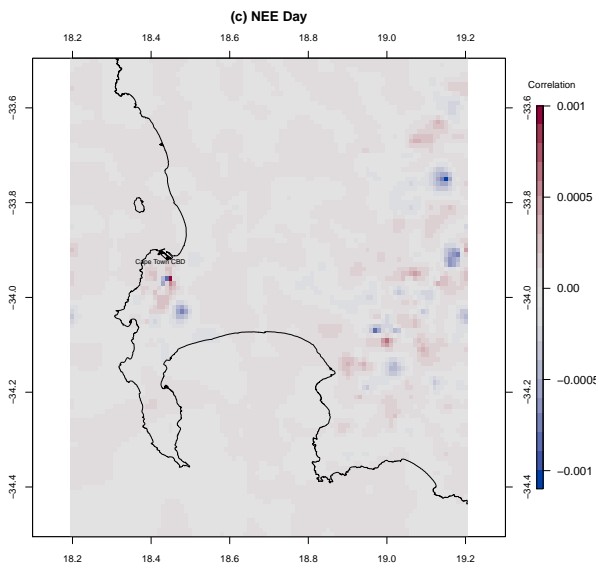

**Figure 16.** A single pixel was selected near the Cape Town CBD during the first week in March 2012. The posterior correlations were extracted for this pixel, and the correlations considered between the daytime working week fossil fuel flux within this pixel and all other sources (i.e. the remaining sources in this pixel, and all sources within all other pixels). The three plots are (a) correlations between fossil fuel working week daytime source during the first week of March 2012 in the selected pixel and surrounding fossil fuel working week daytime sources from other pixels; (b) correlations between fossil fuel working week daytime source in the selected pixel and fossil fuel working week night-time sources within this pixel and all other pixels; (c) correlations between fossil fuel working week daytime source in the selected pixel and biogenic flux daytime sources within this pixel and all other pixels. Extent: between $34.5°$ and $33.5°$ south and between $18.2°$ and $19.2°$ east.





### 3.2.3  Weekly Totals

Three time series plots of the total weekly estimates of $CO_2$ flux over the full domain are presented in Figure 17. The total estimate for a week represents the sum of the fossil fuel and biogenic fluxes for all periods during that week. The uncertainties for these total estimates accounts for correlations between fossil fuel and biogenic fluxes. For the prior estimates the only

correlations are those imposed between biogenic fluxes of neighbouring pixels. The variance for each total estimate accounts for those non-zero covariance terms in the posterior covariance matrix between the different flux components contributing to the total estimate.

The posterior total estimate for the emission of $CO_2$ from the domain is within the confidence bounds of the prior total estimate for the majority of the period from March 2012 until June 2013. The confidence bounds of the posterior total estimates

are narrower compared to those of the prior total estimates. Total prior estimates range between 139.5 and -386.8 kt $CO_2$, with the maximum total during March 2013 and the minimum estimate falling in November 2012. The posterior totals range between 149.5 and -375.1 kt $CO_2$, with the maximum occurring in March 2013 and the minimum in October 2012. During the winter months, from March to July, the posterior overlaps with the prior. The posterior total estimate moves outside of the prior's confidence limits during August and September 2012, which is during the South African spring period. Posterior estimates

are larger compared to the prior estimates. Data are missing during October and from December 2012 to January 2013, and therefore the estimates are completely overlapping during these periods. When observations are available during the summer months in October 2012 and February 2013 the posterior estimates are larger compared to the prior estimates.

The total prior fossil fuel estimate was flat and close to 150 kt $CO_2$ during the winter months, and close to 110 kt $CO_2$ during the summer months. This stepped effect in the fossil fuel fluxes is due to the simple representation of the domestic

emissions in the fossil fuel inventory. It is unlikely that fossil fuel emissions would have a sharp change between summer and winter, and therefore as a separate sensitivity analysis we adjusted the assumption of domestic emissions such that domestic emissions were distributed uniformly during the year. The inversion had the effect of reducing the total estimate, particularly during May 2012 and between March 2013 and June 2013, to a value of as low as 138 kt $CO_2$ during the winter months and to a value of 107 kt $CO_2$ during the summer months and early winter 2013. The total estimates and confidence bounds for June

2013 are outside of those for the prior estimates for the full month. Compared to the total flux, the range of the total fossil fuel fluxes is much narrower (between 100 and 160 kt $CO_2$), and the confidence bounds around the estimates are also narrower. This is not immediately apparent from the plot, but the range of the y-axis needed to be adjusted for the fossil fuel fluxes, otherwise it would appear as a thin line if plotted on the same range as the total fluxes.

Biogenic prior total fluxes range between values close to zero and -494.9 kt $CO_2$ and between zero and -483.1 kt $CO_2$

for the posterior estimates. During the winter months posterior estimates were generally contained within the limits of the prior estimates, except for May 2012, where the total estimate was slightly lower compared to the prior. From August 2012 to September 2012, the posterior total estimate was well above the total prior estimate, indicating that the total uptake of $CO_2$ by the domain was reduced by the inversion during this period.





Comparisons of the biogenic and fossil fuel estimates to the total estimates show that the variability in the total estimates was driven by variability in the biogenic fluxes, and differences between the posterior and prior total estimates are mainly driven by changes to the biogenic fluxes induced by inversion. The fossil fuel flux estimates are limited to a very narrow range compared to the total and biogenic fluxes, with much narrower confidence bands compared with biogenic fluxes. Since the inversion

tended to increase biogenic fluxes when it was decreasing fossil fuel fluxes, it is evident that the inversion is most likely unable to correctly differentiate between these two fluxes when they co-occur in the same pixel. Therefore the inversion is correcting the prior biogenic fluxes to force the posterior modelled concentrations to better match with the observed concentrations, regardless of whether the correction is due to incorrectly specified biogenic or fossil fuel fluxes. This is unsurprising as we provided no information regarding what proportion of the observed $CO_2$ concentration was from fossil fuel contributions

and what proportion was from biogenic sources. The posterior total flux estimate for a pixel and across the whole domain is more reliable compared to the individual fossil fuel and biogenic flux contributions. The cycle in the biogenic fluxes tracks that of temperature well, with emissions more negative during the summer months. There is a lag between when the biogenic emissions begin to decrease and when temperatures begin to decrease, where temperatures start to drop in May but the biogenic emissions start to drop around August. This is due to the rainfall peaking in the winter months so that the period of greatest

productivity occurs in spring and early summer when temperatures are rising but still cool. The opposite trend throughout the rest of South Africa which is dominated by summer rainfall, where the driest period is usually around spring when temperatures start to rise quickly but water availability is at its lowest.





**Figure 17.** Prior (grey) and posterior (green) total weekly $CO_2$ flux estimates ($\text{kt } CO_2$) (right-side axis) and uncertainty limits (shaded area), represented as a 95 % confidence interval, across the full domain of Cape Town, where (a) is the total flux, (b) is the fossil fuel flux, and (c) is the biogenic flux. The daily temperature (°C) as recorded at the Cape Point GAW station is provided by the lower red line (right-side axis). The total weekly flux is obtained by summing over the working week, weekend and day and night-time sources from all pixels for each week.



### 3.2.4 Monthly Totals

Table 4 provides the total prior and posterior flux estimates $(kt\ CO_2\ month^{-1})$ for the domain over a one month period (four weeks) and their uncertainties expressed as standard deviations. The prior total $CO_2$ flux estimates ranged between -1112.1 and $368.7\,kt\ CO_2$, whereas this range was narrowed for the posterior estimates to be between -896.8 and $433.9\,kt\ CO_2$. The

estimates of the total fluxes demonstrate that the inversion has the ability to either decrease or increase the overall flux for the domain, allowing the domain to go from a prior net emitter of $CO_2$ to a posterior net uptaker of $CO_2$ and vice versa. For the months when the prior net flux for the domain was negative, the inversion had the impact of making the posterior net flux more positive. This occurred in August, September and November 2012 during the early summer months. The prior uncertainty in the total flux for the month ranged between 51.1 and $96.4\,kt\ CO_2$, with larger uncertainties during the summer months. The

uncertainties of the posterior estimates were reduced by between 8.7% and 39.9% to range between 35.4 and $81.4\,kt\ CO_2$.

    The inversion acted to reduce the total fossil fuel flux for all months. The prior total fossil fuel flux was approximately $597\,kt$ $CO_2$ during the winter months and $432\,kt\ CO_2$ during the summer months. The posterior estimates were reduced by at most $24\,kt\ CO_2$, with the largest reductions in May to June 2012 and March to June 2013, which was during the winter periods. The prior and posterior uncertainty in the total estimates were very similar for all months, at approximately $2.8\,kt\ CO_2$ for prior

estimates and $2.7\,kt\ CO_2$ for posterior estimates. The largest uncertainty reduction was 16.4% for June 2013. Both June 2012 and November 2012 had uncertainty reductions over 10%.

    In general, the inversion had a dampening effect on the total biogenic flux, resulting in posterior estimates that were closer to zero. The prior total NEE ranged between -1544.4 and $-229.6\,kt\ CO_2$, with the largest uptake predicted for early summer (September and November 2012). The lowest uptake was predicted for March 2012 and 2013. The posterior estimates ranged

between -1274.4 and $-175.2\,kt\ CO_2$, with the lowest amount of uptake also taking place in March 2012 and 2013. In May 2012, June 2012 and April 2013 the inversion led to a more negative total biogenic flux, indicating larger uptake of $CO_2$ predicted by the posterior estimates compared with the prior estimates. The uncertainties for the biogenic fluxes were similar in magnitude compared with those of the total flux estimates, with uncertainty reductions of between 8.6 and 40.0%.

    The total estimates were obtained for only those pixels which had an assigned prior fossil fuel flux. As the inversion was

unlikely to distinguish between fossil fuel and biogenic fluxes from the same pixel, this total was obtained in order to assess how the innovations applied to the biogenic emissions by the inversion altered the total flux for the pixels contributing towards the fossil fuel emissions. The total flux from the fossil fuel pixels was made larger in August, September and November 2012 and February 2013, which is during the spring and summer seasons. Between March and July 2012 and between March and June 2013 the total estimates were made smaller by the inversion, which occur during the winter months. The largest decrease

in the total flux occurred in May 2012, where the prior estimate was reduced from 387.4 to a posterior estimate of $198.5\,kt$ $CO_2$. Similarly large reductions occurred between March 2013 and June 2013. The largest increase in total flux over the fossil fuel emitting pixels occurred in September 2012, with an increase of $183.8\,kt\ CO_2$. Here the inversion changed a large negative prior net total flux over the fossil fuel pixels into a near-zero posterior net flux.





**Table 4.** Estimates of the prior and posterior monthly flux totals ($kt\ CO_2\ month^{-1}$) for the full domain, separated into total flux, fossil fuel flux, biogenic flux, and sum of the total flux from pixels prescribed fossil fuel fluxes, together with the uncertainty estimates expressed as a standard deviation (sd) and the percentage uncertainty reduction from prior to posterior (UR).

| Month | Total Flux $kt\ CO_2\ month^{-1}$ | | | Total Fossil Fuel Flux $kt\ CO_2\ month^{-1}$ | | |
|---|---|---|---|---|---|---|
| | Prior mean (sd) | Posterior mean (sd) | UR % | Prior mean (sd) | Posterior mean (sd) | UR % |
| Mar2012 | 323.4 (60.7) | 433.9 (36.4) | 39.9 | 599.0 (3.0) | 592.9 (3.0) | 0.4 |
| Apr2012 | -15.7 (76.2) | 47.2 (51.4) | 32.5 | 598.1 (2.9) | 592.1 (2.9) | 1.8 |
| May2012 | 85.8 (58.4) | -44.3 (48.6) | 16.7 | 597.8 (2.9) | 580.6 (2.7) | 9.7 |
| Jun2012 | 302.3 (51.1) | 203.6 (46.7) | 8.7 | 596.9 (2.9) | 583 (2.6) | 10.9 |
| Jul2012 | 93.9 (64.9) | 24.0 (54.5) | 16.1 | 596.4 (2.9) | 583.6 (2.7) | 6.5 |
| Aug2012 | -381.3 (80.4) | -244.7 (70.8) | 11.9 | 597.1 (2.9) | 595.7 (2.9) | 0.3 |
| Sep2012 | -980.4 (96.1) | -363.5 (58.6) | 39.0 | 431.4 (2.6) | 421.6 (2.4) | 8.8 |
| Nov2012 | -1112.1 (96.4) | -896.8 (81.4) | 15.6 | 432.3 (2.7) | 419.8 (2.3) | 13.7 |
| Feb2013 | 119.4 (56.5) | 263.7 (36.2) | 36.0 | 434.0 (2.7) | 431.0 (2.7) | 0.6 |
| Mar2013 | 368.7 (57.2) | 354.3 (35.4) | 38.2 | 598.2 (2.9) | 585.1 (2.7) | 6.9 |
| Apr2013 | -9.3 (71.6) | 6.8 (52.1) | 27.2 | 597.7 (2.9) | 580.7 (2.7) | 8.7 |
| May2013 | 1.5 (60.4) | 29.6 (39.1) | 35.3 | 596.5 (2.9) | 581.6 (2.7) | 7.7 |
| Jun2013 | -132 (69) | -130.5 (49.4) | 28.4 | 596.8 (2.9) | 572.5 (2.4) | 16.4 |

| Month | Total NEE Flux $kt\ CO_2\ month^{-1}$ | | | Total Flux Fossil Fuel Pixels $kt\ CO_2\ month^{-1}$ | | |
|---|---|---|---|---|---|---|
| | Prior mean (sd) | Posterior mean (sd) | UR % | Prior mean (sd) | Posterior mean (sd) | UR % |
| Mar2012 | -275.6 (60.6) | -175.2 (36.3) | 40.0 | 558.3 (8.9) | 478.4 (7.9) | 11.1 |
| Apr2012 | -613.8 (76.2) | -613.8 (51.4) | 32.5 | 378.2 (11.7) | 307.5 (11.0) | 6.2 |
| May2012 | -512.0 (58.3) | -681.2 (48.6) | 16.6 | 387.4 (9.6) | 198.5 (9.1) | 5.5 |
| Jun2012 | -294.6 (51.0) | -350.1 (46.6) | 8.6 | 475.3 (8.7) | 363.4 (8.4) | 3.5 |
| Jul2012 | -502.5 (64.9) | -613.3 (54.5) | 16.0 | 378.1 (11.1) | 271.5 (10.7) | 4.0 |
| Aug2012 | -978.4 (80.3) | -720.4 (70.8) | 11.9 | 160.8 (13.9) | 243.9 (13.5) | 3.0 |
| Sep2012 | -1411.7 (96.0) | -821.7 (58.6) | 38.9 | -184.0 (16.1) | -0.2 (14.8) | 8.1 |
| Nov2012 | -1544.4 (96.4) | -1274.4 (81.4) | 15.6 | -184.4 (15) | -150.3 (14.1) | 6.4 |
| Feb2013 | -314.6 (56.4) | -143.5 (36.1) | 36.0 | 339.8 (8.9) | 345.9 (8.0) | 10.3 |
| Mar2013 | -229.6 (57.2) | -177.7 (35.3) | 38.2 | 529.6 (9.3) | 394.5 (8.2) | 11.6 |
| Apr2013 | -606.9 (71.5) | -622.8 (52.1) | 27.1 | 367.2 (11.2) | 204.3 (10.3) | 8.3 |
| May2013 | -595.1 (60.3) | -496.1 (39.1) | 35.2 | 342.2 (10.0) | 169.9 (9.2) | 8.5 |
| Jun2013 | -728.9 (68.9) | -672.9 (49.4) | 28.3 | 274.1 (11.6) | 118.6 (10.6) | 8.2 |





### 3.2.5 Model Assessment

To determine if the prescribed prior covariance parameters were consistent with the model assumptions, the sum of the squared normalised residuals was compared against the $\chi^2$ distribution. For most months the standardised statistic was close to one, but in the case of June and July 2012, this statistic was above 2. We did not scale the variances independently for each month, and

5    therefore the single scaling factor of 2 for the prior flux variances was not large enough for all months. The statistic remained below 2.5 for all months, and had a minimum of 0.68 for the month of November 2012. The mean of the statistic over all months was 1.48. A subsequent study will assess an alternative approach to determine prior flux uncertainties, which would guarantee compliance of the sum of the squared normalised residuals to follow the $\chi^2$ distribution.



**Table 5.** Sum of the squared normalised residuals for each month, which should approximate a $\chi^2$ distribution with one degree of freedom.

| Month | $\chi^2$ statistic | Percentage available data |
|---|---|---|
| Mar2012 | 1.006 | 100% |
| Apr2012 | 1.089 | 75% |
| May2012 | 1.959 | 100% |
| Jun2012 | 2.473 | 50% |
| Jul2012 | 2.170 | 50% |
| Aug2012 | 1.586 | 50% |
| Sep2012 | 1.028 | 100% |
| Nov2012 | 0.678 | 50% |
| Feb2012 | 0.938 | 75% |
| Mar2013 | 1.048 | 100% |
| Apr2013 | 1.776 | 100% |
| May2013 | 1.829 | 100% |
| Jun2013 | 1.706 | 100% |





## 4 Discussion and Conclusions

### 4.1 Inversion Innovation

When comparing the differences between the observed and modelled concentrations for the reference inversion, these residuals are much smaller when using the posterior flux estimates compared with the prior estimates. Moreover, the posterior modelled concentrations were able to track local events observed in the measurements. The agreement between the observed and modelled concentrations was also greatly improved, with standard deviations of the posterior residuals smaller by more the $1\,\mathrm{ppm}$ compared with those of the observed concentrations. Corrections to the sources by the inversion were mainly made to the pixel with the largest point industrial source, to pixels located on Robben Island where known, unaccounted for emission activities were taking place, and to the areas dominated by biogenic sources, which had the greatest amount of uncertainty associated with the prior estimates relative to other sources included in the inversion. This evidence suggests that the inversion setup used here has had some success in capturing information regarding the $CO_2$ emissions in the CT domain, and has applied reasonable corrections to the sources considered.

The inversion was able to reduce uncertainty of the total flux within a pixel by up to 97.7%, and was able to reduce the uncertainty in the total monthly flux over the whole domain by up to 39.9%. In particular, the uncertainty of the total flux attributed to pixels expected to contain fossil fuel sources was reduced to be between 3.0 and 11.6%.

The largest innovation to a fossil fuel source was applied to a pixel which contained an important point source in the domain, which was a crude oil refinery. This facility can process up to 100,000 barrels of crude oil per day. Unlike most industrial sources in the area which would be expected to have fairly consistent activity from day to day, activity at the crude oil refinery would depend on oil supply and on the global oil prices. During the period of March 2012 to June 2013, the global monthly oil price deviated between $117.29 in March 2012 and $90.73 in June 2012, ending on $99.74 in June 2013 (World Bank commodity prices). In addition, the consumption of liquid petroleum gas and heavy furnace oils may have decreased during this period (City of Cape Town, 2015). As this is a source with a large amount of expected variability, it's unlikely that the inversion method with distant measurement sites will be able to adequately estimate the flux in this pixel. In order for this to occur, the measurement site would have to consistently view the source during periods of both high and low activity in order to provide an unbiased estimate of the mean flux from the site which is representative of the average activity occurring at the site. Inverse modelling could be used to estimate this particular source if a ring of instruments were placed around the site in order to capture information from the site at all times, regardless of the prevailing wind direction, such as suggested for the Otway $CO_2$CRC carbon capture project (Cook, 2012).

Compared to the fossil fuel emissions, relative innovations to the biogenic fluxes were much larger, due to the large uncertainty prescribed to these fluxes, particularly to the natural areas near the CBD of CT, as well as to agricultural regions within the domain. The prior estimates are dependent on the CABLE land-atmosphere interaction model, and despite being dynamically driven by the CCAM regional climate model, which relies on reanalysed observations of the climate from NCEP, is still under ongoing development for use over South Africa. In general, the inversion tended to increase the biogenic fluxes so that the total flux was less negative compared to the priors, except in the case of May to July 2012, where the overall flux



was made more negative. This occurs during the winter period, which is the dominant growing season in the Western Cape. Therefore the CABLE model appears to overestimate the drawdown of $CO_2$ in the warmer, drier period, but underestimates the drawdown in the cooler, wetter season.

## 4.2 Distinguishing Fossil Fuel and Biogenic Emissions

The spatial distribution and magnitude of the posterior fluxes and their uncertainties is strongly dependent on the prior spatial assignment of sources. In a city like CT, these two sources are usually overlapping, with vegetation within the city representing a significant and large sink of $CO_2$, particularly during the growing season. If no fossil fuel emission source is thought to occur in a particular pixel, the inversion under the current setup will only be able to adjust the biogenic emissions if in reality there is an unknown fossil fuel source in the pixel. Therefore the success of the inversion is largely dependent on how well the spatial extent of fossil fuel and biogenic sources are prescribed in the prior information. The fossil fuel and biogenic sources could be better differentiated if there were additional measurements of $\Delta^{14}C$ at each of the sites (Turnbull et al., 2015).

In Bréon et al. (2015) biogenic fluxes were applied at a larger grid size compared to the fossil fuel emissions. Effectively this means that perfect correlation was applied between the biogenic fluxes for all pixels which fall within the same larger grid cell. By distinguishing the biogenic and fossil fuel sources in this way, it may allow the inversion to correctly allocate corrections between the fossil fuel and biogenic sources. We attempted to implement a similar idea by allowing correlation between biogenic emissions of neighbouring pixels and not prescribing correlations with fossil fuel sources. As the model tended to reduce fossil fuel emissions and increase biogenic fluxes in the same pixel, it appears that the inversion is likely not correctly adjusting the individual sources. With the large coverage of vegetation within the domain, it is unlikely that a measurement network with only two sites could accurately estimate a given industrial point source, but there is still potential to monitor the overall city emissions, and assess the feasibility of inventory information.

## 4.3 Strengths and Limitations

This paper represents a first attempt at estimating $CO_2$ fluxes at the high resolution of $1\,km$ by $1\,km$ over CT, distinguishing fossil fuel, terrestrial and oceanic biogenic sources. A previous network design study for South Africa (Nickless et al., 2015b) showed the aggregation errors could be very high at the regional level. By maintaining the $1\,km$ by $1\,km$ through the inversion process, aggregation errors could be minimised. We also used the CABLE model to provide estimates of the biogenic fluxes which was dynamically coupled to the CCAM regional climate model, and therefore prior emissions would be determined by the estimated climatic variables at this high resolution scale. This is advantageous over using modelled fluxes which are derived at a different scale to the atmospheric transport, and which then have to be scaled artificially to the resolution of the inversion.

A limitation of this study is that human respiration was not accounted for. With a population of over three million, this flux could represent up to $104.7\,kt\ CO_2\ month^{-1}$, if we attribute $1\,kilogram\ CO_2$ per day to each person (Bréon et al., 2015). This represents between a fifth to a quarter of the total fossil fuel flux estimated for the domain, and therefore is by no means a negligible quantity. Including this fossil fuel flux in the inventory information would most likely lead to confounding between the domestic emissions and the human respiration, as this source calculated based on population data. A sensitivity analysis



could be performed where the domestic emissions include an additional contribution from human respiration. Domestic emissions were also heavily dependent on the assumption regarding how heating emissions were distributed during the year. This had a large impact on the temporal profile of fossil fuel emissions, resulting in a lower average emission in summer compared to winter, which persisted in the posterior estimates of the fossil fuel fluxes. An additional sensitivity analysis in the companion paper considers assuming homogeneous heating emissions through the year, and investigates the impact on the total posterior estimates.

This inversion was performed by solving for the actual concentrations, rather than using the gradient method of Bréon et al. (2015). There are several reasons why this is suitable in these circumstances. Firstly, we had the advantage of a background site which viewed background levels over 70% of the time. As our city is located within the Southern Hemisphere, far less variability is expected between the hourly measurements or from week to week compared to the Northern Hemisphere, and we would expect the oceanic boundaries to have very similar background levels of $CO_2$ concentrations. This was confirmed by the results of the inversion, which made almost no adjustments to the oceanic boundary concentrations, but made slightly larger adjustments to the boundary concentrations of the north and east borders. This implies that the adjustments made by the inversion were largely due to the sources within the domain, which then allowed for the stronger agreement between the posterior modelled and observed concentrations. In addition, there are no large expected sources located anywhere near the boundary of the domain. The next major city in the proximity of CT is Port Elizabeth, which is over 600 km away. For this reason, it is unnecessary to solve for the gradients between the two sites, as in these circumstances we do not require the removal of unknown sources from the observation dataset. This allowed us to use the entire measurement record, which is an advantage as we only had two sites available.

Secondly, the gradient method would likely have performed poorly here, as the direction of travel of an air parcel between the two sites would not necessarily be in a straight line due to the topography of the site and demonstrated by the plot of the domain sensitivity at the two sites (Figure 18). Therefore extracting observations based on the prevailing wind speed at the site would have not represented true gradients in the $CO_2$ concentrations between the two measurements sites.

The sensitivity to the domain also reveals that the sites are often viewing oceanic sources (Figure 18). A limitation of this study is that a single time series of ocean biogenic emissions was used as the prior estimate for all oceanic pixels in the domain. The fluxes from the near-coastal oceanic pixels are likely to have significant spatial heterogeneity, although smaller compared to the terrestrial biogenic fluxes. A way of improving this would be to use the output of a model representing atmosphere-land-ocean biogeochemical exchanges to provide prior fluxes over both the land and ocean. The CSIR's Variable Resolution Earth System Model (VRESM) is such a model currently under development, which aims to couple CCAM, CABLE, and CSIR's Variable-Cubic Ocean Model (VCOM), and Pelagic Interactions Scheme for Carbon and Ecosystem Studies (PISCES) to model ocean transport and biogeochemistry (Engelbrecht et al., 2016). Due to the amount of shipping activity around the CT harbour and within the ocean domain viewed by the Robben Island site, the inventory analysis could also include information on shipping routes so that the inversion can adjust fossil fuel sources in these ocean pixels as well.

The uncertainty in the biogenic sources played an important role in the outcome of the inversion. If tighter uncertainty limits could be placed on the biogenic flux estimates from the land-atmosphere exchange model, made possible by validation work





through, for example, eddy-covariance measurements over dominant vegetation types within the domain, it would allow the inversion to better constrain the emissions from the fossil fuel sources. The use of a land-atmosphere exchange model well suited to the vegetation within the domain of a city scale inversion is essential for reducing the tendency of the inversion to adjust the most uncertain sources to suit the observations, rather than fossil fuel sources which are of foremost interest.







**Figure 18.** Mean weakly sensitivities ($\mathrm{ppm\,kg^{-1}\,CO_2\,m^{-2}\,week^{-1}}$) of the measurement sites at Robben Island and Hangklip to the surface sources, which are derived from the mean sensitivities over all sensitivity matrices for the full duration of the study.



*Code and data availability.* Data and code related to the Bayesian inversion procedure can be made available upon request





*Author contributions.* AN installed and maintained all the instrumentation at Robben Island and Hangklip, obtained the measurements and processed these into hourly concentrations, ran and processed the result of the LPDM in Fortran, produced all code and ran the inversion in Python, processed all the inversion results using R Statistical Software, produced all graphics and tables, and was responsible for the development of the manuscript which forms part of her PhD. PJR was the main scientific supervisor, oversaw all implementation of the
inversion, and provided guidance on the presentation of results. E-GB provided the Cape Point concentration and meterological data, and provided assistance and guidance in the installation and maintenance of the Picarro instrumentation. FE provided the CCAM and CABLE data. BE provided guidance on statistical issues. RJS provided guidance on the location of the sites and provided advice on the interpretation of the biogenic fluxes. All authors had the opportunity to comment on the manuscript.

*Competing interests.* The authors declare that they have no conflict of interest

*Acknowledgements.* We would like to acknowledge and thank Dr. Casper Labuschagne and Danie van der Spuy of the South African Weather Service for their assistance in maintaining the instruments at Robben Island and Hangklip, and Dr. Casper Labuschagne for his guidance on processing the instantaneous $CO_2$ concentration data; Martin Steinbacher for providing guidance and schematics on the calibration system used on the Picarro instruments; Robin Poggenpoel and Jacobus Smith of Transnet for allowing us access to the lighthouses; Peter Saaise of Transnet, the Robben Island lighthouse keeper, (and his daughter) for assisting when the instrument was not responding; Marek Uliasz
for providing us access to the code for his LPDM model; Thomas Lauvaux for providing guidance on processing the LPDM results and useful discussion on the boundary contribution in the inversion; Keith Moir, Rooi Els, for providing wind data near Hangklip. Use was made of the University of Cape Town ICTS-HPC cluster. Please see http://hpc.uct.ac.za/ for details. We would like to thank Andrew Lewis of the University of Cape Town HPC facility for providing useful advice on improving the efficiency of the Python runs. This research was funded by competitive parliamentary grant funding from the Council of Scientific and Industrial Research awarded to the Global Change
Competency Area towards the development of the Variable-resolution Earth System Model (VRESM; Grants EEGC030 and EECM066). Additional funding was obtained from the South African National Research Foundation for the Picarro instrumentation.



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
