# Peer review of "Estimates of CO2 fluxes over the City of Cape Town, South Africa, through Bayesian inverse modelling"

_Atmospheric Chemistry and Physics, 2017_

## Referee Comment (RC1) · Anonymous Referee #1 · 19 Sep 2017

The manuscript "Estimates of CO2 fluxes over the City of Cape Town, South Africa, through Bayesian inverse modelling" presents an atmospheric observation and modeling system dedicated to the monitoring of the CO2 emissions from Cape Town. In particular, it presents series of CO2 measurements at two new stations in the vicinity of the city, a high resolution atmospheric inverse modeling system and results of emission inversions over more than 1 year. One the main conclusion of the manuscript is that the inversions hardly manages to distinguish between the biogenic fluxes from the ecosystems in the region and the city emissions.

I appreciate that the authors installed and maintained the two new measurement sta-

tions, made an inventory analysis and developed and applied a dedicated inverse modeling framework. To this date, the number of attempts at developing city scale inverse modeling systems is still small. The design of the measurement network in view to monitor Cape Town emissions seem highly relevant and promising. In such a context, I would like to support the publication of a good presentation and analysis of the authors' work.

However, as detailed below, I think that there are serious issues in this manuscript, which should undergo a major revision.

First, while the text could appear to be well structured at first sight, it is actually strongly hampered by a critical lack of rigor in the details. The pieces of text written too fast, the shortcuts, the theoretical mistakes and the approximations narrow the scope of this study to a community of experts well used to the theoretical material of this paper. They also often make the text really confusing and difficult to read in details. Redundancies and the detailed account of diagnostics that are sometimes uselessly complicated also participate to the difficulty to read this manuscript. I provide a list of examples below to demonstrate it. These examples represent only a small part of the issues I have met when reading this manuscript.

Second, I have concerns regarding some of the main results of this manuscript and regarding the lack of material that could help evaluate them.

A) Figures 9a,c and 10a,c indicate very large misfits between the model and the measurements when using the prior fluxes while, as highlighted by the manuscript itself, one does not expect a very large variability of the CO2 in the area. How is it possible to get such an amount of misfits larger than 30 ppm (and in the range 100 to 200 ppm) while the data nearly never reach an excess of 30 ppm over their baseline, and while the measurement stations seem to be quite distant from the city major point sources ? The manuscript ignores that such prior misfits strongly question the reliability of the atmospheric transport modeling framework, and thus of the inversion system.

The authors say that the model "shows ability to track local events at the sites" but it is impossible to assess on Figures 9 and 10. Furthermore, given the very large size of the control vector (and thus the very large number of degrees of freedom in the inversion system), it is not really surprising to see that the inversion manages to fit the data to a far better extent. I find it difficult to take it as a demonstration that the atmospheric transport model is reliable. In particular, opposed to what is said on page 1 of the supplementary material ("some evidence to provide confidence in the modelled meteorology is provided in this section"), the figures 1 and 2 of this supplementary material strongly question this reliability, displaying very large misfits between the modeled and measured timeseries of wind speed, with a weak correlation between them. The difference between the height of the measurement sites and the model vertical representativeness can hardly explain such misfits and even less such a low correlation. This requires a deeper analysis or better insights regarding the skill of the 1 km resolution CCAM model at city scale from studies like Engelbrecht et al. (2009, 2011).

By gathering the night-time and daytime data both in their analysis and in the inversion system, the authors do not help investigating this issue. We can assume that the largest misfits are obtained at night. However, opposed to what is said on page 1 line 12 ("Night-time observations were included, but allocated much larger errors compared to the daytime observations"), figure 5 indicates that the increase of the observation error at night is likely far from sufficient to cope with the increase of model errors at night. I think that the authors have overestimated their ability to assimilate night-time data. Analysis of the misfits between the modeled and measured $CO_2$ and of the corrections to the fluxes applied by the inversion at night vs day could help investigate this topic.

B) One of the main discussion of the manuscript is related to the lack of distinction between the anthropogenic and natural fluxes in the inversion results. What puzzles me is that the analysis of the posterior error covariance matrices should be a very helpful

tool to feed such a discussion. The authors display correlations between uncertainties in the emissions at a given pixel and the NEE field in Figure 16 but ignore them when conducting this discussion.

Actually, Figure 16 shows correlations that are very close to 0, which undermines the assumption of the lack of distinction. When looking at the station locations and at the situation of the city vs. the areas of high NEE (which are also the areas of high prior uncertainties in the NEE and of high corrections to the NEE in the inversion), it is difficult to understand why the separation between the NEE and the anthropogenic emissions should be so problematic for the inversion. The authors have used very large prior uncertainties in the NEE, and the NEE dominates the mean diurnal cycles of the stations. This explains why, on the first order, the inversion focuses on the NEE rather than on the anthropogenic emissions (from that point of view, it would not be a problem of distinction between NEE and anthropogenic emissions, but rather a problem of detection of the emissions despite the dominating signal from NEE). But, according to figures 3 and 4, this should not prevent the inversion from getting a signal that is dominated by the anthropogenic emissions when the wind blows roughly from one station to the other one through Cape town. Paradoxically, the type of "gradient approach" that the author assume to be useless for their study case (p13 line18 p61lines7-9) may help them to cope with the NEE signal. All of this needs to be better analyzed. The analysis of the variations of the modeled contribution from the NEE vs that from Cape Town at the measurement sites could be very useful.

C) Regarding the prior uncertainties in the NEE, the relative values discussed in section 2.9.2 can be very large. They deserve some justification based on the CABLE validation studies, especially since they will be amplified by the multiplication of the prior uncertainties by a factor of 2 in section 2.11 (Figure 4 being misleading). I understand that when aggregating them over the modeling domain, we get much smaller relative values due to using a small spatial correlation length scale. However, could not it be an issue for the results at the pixel scale and, implicitly, for the control of the

highly localized anthropogenic emissions (see the strange correction patterns in figure 13 and 14) ?

D) I am not sure to understand the distribution of the emissions from Cape Town according to the author's inventory. p11 says "But of the carbon emissions due to energy usage, only 27% were attributed to the transport sector as a result of the carbon intensive usage of coal for electricity generation to provide almost all of the energy to the residential and commercial sectors in South Africa, which emit approximately 29% and 28%, respectively, of the total carbon emissions of CT (City of Cape Town, 2011)." Paying much attention to the terms "electricity generation" and "almost all of the energy", my understanding of this sentence is that there is a large number of coal power plants within the city bounds (otherwise the part of the emissions within the city due to the transport would be very high), which represent almost 60% of the city CO2 emissions, while the direct emissions from the residential and commercial areas should be very low. However, this seems strongly at odd with the figures and discussions of this manuscript and this would be highly problematic for the atmospheric inversion. Could the authors clarify this point ?

It would be difficult and useless to discuss the secondary scientific issues at this stage. The authors should first improve the presentation of this study. The following illustrations of the lack of rigor and clarity of the text are mainly picked up from the abstract and sections 1 and 2. Actually, the text of these following sections is often more problematic than that of these first ones. Since I could have listed a far larger amount of issues, the author should not limit themselves to correct for the ones given below. They should rather consider a full and deep rewriting of the manuscript.

- Section 2.1 makes a rough account of the traditional theoretical framework of the inversions (e.g. sentences like "If we assume a Gaussian error distribution for the surface fluxes and concentrations we obtain the following cost function for our least squares problem" on page 6). Throughout the manuscript, the covariances between uncertainties in fluxes are often called covariances between fluxes.

- in the abstract and first sections, the text introduces the concept of "boundary concentrations" without specifying that the boundaries relate to the modeling framework. This becomes problematic when explicitly speaking about the sensitivities of the measurements to the boundary concentrations (e.g. l23-24 p5)

- the text often uses the terms "sources" and "emissions" while speaking about (natural) fluxes that can be negative

- p10 line17-19 the sensitivity is not the influence, H is not HTtranspose

- p11 "and allowed for small scale transport features to be maintained in H": using a coarse resolution control vector does not remove the small scale transport features in H

- Section 2.11 states that the error covariance matrix that is underestimated in the first configuration according to the chi test is necessarily B ("and values greater than one indicate that the variance prescribed is lower than it should be and therefore the posterior estimates will be over-constrained by the prior fluxes") while it could be R (and actually some of the results favor the assumption that it is R).

- sample of awkward, meaningless or confusing sentences:

p1: "The inversion solved for the actual concentration measurements at each site, which was made possible by the use of the Cape Point background site to provide information on the boundaries, and was necessary due to the effect of topography on the atmospheric transport, affecting particularly the sensitivity of the Robben Island site to the surface fluxes."

p1: "The mean bias in the modelled concentrations was reduced from -2.9 ppm, with interquartile range -9.1 to 3.7. . ."

p2 "The inversion was also allowed to solve for each of the four boundary concentrations (north, east, south and west), but these were provided with tight constraints provided by the background site."

p2: "Model assessment by means of the chi2 statistic indicated that the mean statistic was 1.48 over all months"

p2: "prior values for the model errors or the uncertainty in the fluxes"

p2: "By mitigating the CO2 impact of cities, cities play a pivotal role in decreasing their own climate vulnerability."

p3: "Originally implemented to determine global, large scale sources and sinks of CO2, regional or mesoscale scale atmospheric inversions are becoming more common."

p3: "All emissions are observed as an aggregated total, therefore all emission sources are accounted for, but it is challenging to separate out these CO2 emissions into different components of the total CO2 budget without additional measurements or confidence about spatial and temporal patterns of emission"

p3: "At the moment background conditions are not sufficiently characterised in order to use isotope tracers to differentiate between fossil fuel and biogenic sources, as these measurements are far rarer than atmospheric measurements of CO2 mole fractions" (all the last paragraph of page 3 follows a very loose reasoning which is a bit difficult to follow)

p4: "This analysis underpins the assumptions of human behaviour driving the anthropogenic emissions"

p4: "Previous studies on estimating CO2 emission for cities have found that errors in atmospheric transport modelling are a significant contributor to the overall uncertainty of emission estimates and therefore more work is required to refine these models so that they can perform more reliably during these periods of high uncertainty before they can be used to infer emission estimates at all times of the day."

p4: "To be able to verify emissions from underlying processes, higher resolution inverse modelling systems are needed to better understand and quantify emissions from different sectors."

p4: "This model at a slightly lower resolution was previously used for a regional network design study, making use of he same Bayesian inverse methodology and has been verified over southern Africa at relatively low resolutions through to ultra high resolution (1 kmâĞě1 km)"

p5 "A linear relationship can be used to describe the relationship"

p5 "The vector of the modelled concentrations cmod is a result of the contribution from the sources s, described by the transport or sensitivity matrix H ."

p11 "In the case of the boundary sources which are given as concentrations, their contributions to the concentration at the measurement site are expressed as a proportion of their concentration, dependent on their influence at the receptor site."

- Redundancies: p10: lines 6, 10-11 and 13 p10 lines 29-30 with all other parts of the text explaining it before and with p11 line1-2 + many details of the analysis in section 3

- The design of the figures should be improved. The location and name of the sites are hardly visible in figure 1. Labels are too small and the fields are fuzzy (mainly due to the choice of the colorbars) in figures like Figure 3. The choice of the colors in figures like Figure 8 is poor: on my screen, it is really hard to distinguish between the different curves. In figures like Figure 9, it is impossible to analyze the different timeseries since they are compressed along the x axis (with nighttime and daytime data mixed together) and most of the measurements are hidden behind the model patches. In most of the figures, there is a lack of subtitle and legends to help the reader while the captions are sometimes quite complex (e.g. the legend of figure 15). Therefore, in general, the figures are very difficult to read. In Figure 15: it is difficult to see the pixel against which covariances are computed.

- The notations used in several equations are not really optimal. At least, they do not help understand the meaning of the variables, e.g. Etrans in equation 11 which refers to a subcomponent of the transport error, while Eobs refers to another part of

the transport error, and not to the observation error (which is the sum of the transport and measurement errors). Eq 8 is not really adapted to equations 1 and 6. Eq 9 and 11 are informal.

- There are too many significant digits in tables 3 and 4 which makes these tables difficult to read. Would not it be better to show the content of these tables using plots ? I do not understand why the authors produce distinct sections (3.2.3 and 3.2.4) and tables (3 and 4) for the variations of the 1-week mean and 1-month mean flux budgets. This is a source of redundancies and I do not think that they manage to bring specific insights for each of the two timescales.

- The acronyms CT and CBD are not defined explicitely.

- Section 2.2. and Equation (6) are confusing regarding the composition of the control vector (regarding the fact that the inversion solves for the fluxes at the transport model spatial resolution and regarding the control of the average conditions for each of the 4 lateral boundaries). We need to guess it from the numerical derivation of the size of the control vector or wait for sections 2.5 to get clearer details. The situation is similar regarding the fact that 1-month inversions are conducted to cover the 13 month period.

- there is a problem with the order of the citations (see Tarantola (2005) and Enting (2002) and Lauvaux et al., 2016; Bréon et al., 2015 on page 5)

- the percentile filtering technique at Cape point and its impact on the timeseries at this site is not well detailed (e.g. on which time windows, at which timescales is it applied ?), while the station can be influenced by Cape Town, and by the NEE in the region covered by the modeling domain. This is perturbing since the system controls the North and East boundary conditions that are inland (and thus separated from the Cape point station by large areas of NEE and potentially influenced by even larger areas of NEE outside the modeling domain) and since it uses the data filtered at Cape point to provide a prior value with a low prior uncertainty to these conditions.

- p20: the discussion on the representation error ignores the part of this error due to the difference of spatial representativeness between the measurements and the model ("We did not account any further for aggregation or representation errors as we did in the network design, as we were running the inversion at the same spatial scale as the transport model.")

———————————————

---

## Referee Comment (RC2) · Anonymous Referee #2 · 20 Oct 2017

The manuscript by Nickless et al. describes a Bayesian inversion study over the city of Cape Town which aims to improve CO2 flux estimates. Prior estimates for the biosphere fluxes are derived from a land surface model coupled to a regional climate model running at 1km resolution. Prior estimates of fossil fuel emissions are based on a comprehensive inventory analysis. Two monitoring sites were set up recently for this study, whereas the third monitoring site is an established site within the GAW network.

The inversion was able to reduce total flux uncertainties by up to over 90% at a grid cell level. Biosphere fluxes show the largest reduction in uncertainty, mainly due to larger prior uncertainties assigned. The authors conclude that the success of the inversion

largely depends on the quality of the prior estimates for fossil fuel and biosphere fluxes. If they could provide smaller prior uncertainties for the biosphere fluxes then fossil fuel emissions could also be better constrained.

General comments:

Overall I think this work makes an important contribution and should therefore be of interest for readers of ACP. Atmospheric inversions over cities are challenging and I appreciate the amount of work that has gone into this study. However, I think the manuscript can still be improved. The results section for example contains (too) many details including figures and tables. This makes it really difficult for the reader to focus on the main findings. Sentences are sometimes overly long and therefore hard to understand. Figure quality could also be improved.

Specific comments:

Abstract:

- please consider rephrasing the sentence stretching from L6 to L11 on P1.

- information such as "interquartile range" should be removed from the abstract (P1: L15, L16, L17, L18)

Main text:

P5, L9: "CABLE was dynamically coupled to CCAM" I don't know how many times I read this phrase in the manuscript. Please avoid repetitions.

P6, L19: surface source grid "point" ?

P10, L5-L23: you talk about the generation of sensitivity matrix H, but you also mention sensitivity matrix T, which is confusing

P11, L14-L16: I don't understand this sentence. Please rephrase using shorter sentences.

P11, L23: "For the network design ..." ? It is confusing that you refer to a network design here.

Figs.3+4: This figures need improvement. Please label the sub-plots. The information is provided in the caption, but it would be much easier to have it in the figure directly. The used colour scheme is probably not that helpful either.

Section 3.1: I found this section far too detailed. Maybe some of the content could be moved to the supplementary material?

Fig6: Please provide a legend for symbols and colours within the figure.

Fig7: Is there a reason why Oct12, Dec12, Jan13 are missing? I am not sure if the diurnal cycle for all months is required in the manuscript?

Table 2: Consider to move this table to supplementary material.

P34, L8-L9: NEE or NEP?

Figs. 9+10: Please provide a legend for symbols and colours within the figure.

Fig.12: Please label the sub-plots (i.e. north, east, south, west)

P42, L1-L3: Please rephrase this sentence.

Figs.13+14: Add labels and improve colour scheme.

Table 3: Is this level of detail really required in the manuscript?

P49: I have difficulties understanding the whole section on this page.

Figs. 15+16: I cant see the value of those figures. I don't think they are even referenced in the text?

Section 3.2.3: Can you please check the units? You refer to fluxes, but units are in ktCO2.

Fig. 17: Please add a colour legend. Check units.

Section 3.2.4: Units not consistent. I think, they are all fluxes.

Table 5: move to supplementary material?

---

## Author Comment (AC2) · 30 Nov 2017

**Response to interactive comment by Anonymous Referee 2 on "Estimates of CO2 fluxes over the City of Cape Town, South Africa, through Bayesian inverse modelling" by Alecia Nickless et al.**

Alecia Nickless, Peter J. Rayner, Francois Engelbrecht, Ernst-Günther Brunke, Birgit Erni, and Robert J. Scholes

We would to thank the referee for their consideration of the paper and for their support of this work.

Referee 2 states: "*Overall I think this work makes an important contribution and should therefore be of interest for readers of ACP. Atmospheric inversions over cities are challenging and I appreciate the amount of work that has gone into this study. However, I think the manuscript can still be improved. The results section for example contains (too) many details including*

5  *figures and tables. This makes it really difficult for the reader to focus on the main findings. Sentences are sometimes overly long and therefore hard to understand. Figure quality could also be improved.*"

The manuscript has been largely rewritten to improve the focus of the paper and improve clarity. Figures have been improved to be clearer, use better colour schemes, and to provide more information through legends rather than overly long and complicated captions. Certain sections have been removed or merged. For example, the supplementary section on the validation of

10  the wind product from CCAM has been removed as the sites were generally not equipped or suited for this purpose. Instead more information on the general wind patterns within the domain has been provided, which is of more use in this context. The section 3.2.3 and 3.2.4 have been merged.

The referee makes the following specific comments:

"*Abstract:*

15  *- please consider rephrasing the sentence stretching from L6 to L11 on P1.*"

This has been re-written.

Previous: "Prior estimates of the fossil fuel emissions were obtained from an inventory analysis specifically carried out for this inversion exercise, making use of vehicle count data, population census data, fuel usage at industrial point sources, and aviation and shipping vessel counts. The inversion solved for the actual concentration measurements at each site, which was

20  made possible by the use of the Cape Point background site to provide information on the boundaries, and was necessary due to the effect of topography on the atmospheric transport, affecting particularly the sensitivity of the Robben Island site to the surface fluxes."

Revised: "Estimates for the prior fossil fuel fluxes were obtained through a bespoke inventory analysis carried out specifically for this inversion study. This inventory analysis made use of a traffic model producing vehicle kilometres travelled on each

25  stretch of road, population census data, fuel usage at industrial point sources, and aviation and shipping vessel counts. The inversion solves for the working week and weekend fossil fuel fluxes, the weekly net ecosystem exchange, and the average

weekly concentrations at the domain boundaries through modelling the concentrations of $CO_2$ at the Robben Island and Hangklip measurement sites."

"- *information such as "interquartile range" should be removed from the abstract (P1: L15, L16, L17, L18)*"
This has been removed from the abstract.

5 "*Main text:*
*P5, L9: "CABLE was dynamically coupled to CCAM" I don't know how many times I read this phrase in the manuscript. Please avoid repetitions.*"
As in response to the remarks by Referee 1, these sorts of redundancies have been removed through a thorough revision of the manuscript.

10 "*P6, L19: surface source grid "point" ?*"
What was meant here was "surface pixel"

"*P10, L5-L23: you talk about the generation of sensitivity matrix H, but you also mention sensitivity matrix T, which is confusing*"
This was a typo. It has been corrected in the revised manuscript. There is no matrix T.

15 "*P11, L14-L16: I don't understand this sentence. Please rephrase using shorter sentences.*"
This paragraph has been rewritten:

Previous: "Since the surface sources are expressed as fluxes of carbon, the contribution to the concentration at the measurement site is expressed in the amount of carbon seen at the measurement site from a particular source. In the case of the boundary sources (or contributions from outside of the domain) which are given as concentrations, their contributions to the concentra-
20 tion at the measurement site are expressed as a proportion of their concentration, dependent on their influence at the receptor site. Ziehn et al. (2014) shows that by calculating the Jacobian which provides the sensitivities of observed concentrations to boundary concentrations, the boundary contribution can then be written as:"

Revised: "The fluxes from the surface pixels are expressed in $\mathrm{kg}\ CO_2\mathrm{m}^{-2}\,\mathrm{week}^{-1}$ and are transformed through $\mathbf{H}$ into contributions to the concentration at the measurement site in units of $\mathrm{ppm}$. The inversion solves for the concentrations at the
25 boundary of the domain. The sensitivity of each site to each of these boundary concentrations (North, East, South and West) is given as a proportion of the concentration at the boundary, so that the sum of the boundary sensitivities for any given hour add to one. Ziehn et al. (2014) shows that by calculating the Jacobian which provides the sensitivities of observed concentrations to boundary concentrations, the boundary contribution can then be written as:"

"*P11, L23: "For the network design ..." ? It is confusing that you refer to a network design here.*"
30 This has been removed.

"*Figs.3+4: This figures need improvement. Please label the sub-plots. The information is provided in the caption, but it would be much easier to have it in the figure directly. The used colour scheme is probably not that helpful either.*"

These plots have been revised to make better use of subplot titles and legends to reduce the amount of text in the captions, and the text has been made clearer. The colours have been slightly altered to improve clarity. The colour scheme was deliberately chosen so that 1.) positive, negative and zero fluxes would be distinct. 2.) And the "jagged" graduation in colours so that the very large fossil fuel fluxes at point sources could be distinct, but also the smaller fluxes, such as those for transport and residential sources. These are smaller, but more numerous than the point sources, and we wanted these to be distinguishable from each other as well.

"*Section 3.1: I found this section far too detailed. Maybe some of the content could be moved to the supplementary material?*"

We agree. This section has been moved to the supplementary material.

"*Fig6: Please provide a legend for symbols and colours within the figure.*"

Legends for the symbols and colours used in these plots has been provided, instead of including this information in the caption.

"*Fig7: Is there a reason why Oct12, Dec12, Jan13 are missing? I am not sure if the diurnal cycle for all months is required in the manuscript?*"

These are plots of the observed diurnal pattern in the concentrations. Unfortunately observations were not available for all months. The monthly plots have been moved to the supplementary material, and instead the average diurnal plot over all periods has been improved in the manuscript (Figure 8 in the original manuscript).

"*Table 2: Consider to move this table to supplementary material.*"

Agreed. This will be moved together with Section 3.1. to the supplementary material.

"*P34, L8-L9: NEE or NEP?*"

The net primary productivity was used as the uncertainty in the net ecosystem exchange. This has been corrected in the manuscript. Generally, for most parts of South Africa, the NEE flux itself is very small, but could be resulting for relatively large respiration and photosynthesis fluxes. If the uncertainty in the NEE estimate was based on the NEE value itself, it would be unrealistically small. This discussion has been made clearer in the revised manuscript.

"*Figs. 9+10: Please provide a legend for symbols and colours within the figure.*"

These figures have been changed altogether to ensure that the time series is not squashed, and to separate out the day and night concentrations and residuals. Better use has been made of legends. Please see response to Referee 1 for an example of the new time series plots.

"*Fig.12: Please label the sub-plots (i.e. north, east, south, west)*"

Thank you. This has been done.

"*P42, L1-L3: Please rephrase this sentence.*"

This sentence has been revised.

5    Previous: "The innovation of the inversion can be observed through the differences between the prior and posterior flux estimates, which we refer to as the innovations, and through the change in the uncertainty estimates."

Revised: "We refer to the difference between the prior and posterior flux estimates as the innovations. The impact of the inversion on the flux estimates can be assessed through the size and direction of these innovations and through the reduction in the uncertainty estimates."

10    "*Figs.13+14: Add labels and improve colour scheme.*"

These plots have been improved. As discussed earlier, the colour scheme was selected to ensure that positive, negative and zero fluxes were distinct. The jagged colour graduation was also selected so that large fluxes could be displayed together with small fluxes, while still showing subtle differences between the smaller fluxes.

"*Table 3: Is this level of detail really required in the manuscript?*"

15    The purpose of including this table was so that the impact of the inversion could be assessed from month to month. This table attempts to summarise the distribution of the fluxes over the domain, and show how that distribution is changed by the inversion. This section has been largely rewritten. The table has been converted into a time series of box plots to show the distribution of these pixel-level fluxes.

"*P49: I have difficulties understanding the whole section on this page.*"

20    This section has been largely rewritten. The main point that this section makes is that, under the current inversion framework, the posterior off-diagonal covariances are very small. This is because the uncertainties in the modelled concentrations that are attributed to the flux contributions ($\mathbf{H}\mathbf{C_{s_0}}\mathbf{H}^T$) are small relative to the uncertainties specified in the observation error covariance matrix ($\mathbf{C_c}$). If the diagonal elements in $\mathbf{C_c}$ could be made smaller, then the posterior covariances would be much larger in magnitude. When we wish to estimate the difference between the fossil fuel and NEE fluxes $s_{f,i} - s_{NEE,i}$, in other

25    words, to distinguish between the fossil fuel and NEE fluxes in the same pixel, the variance of this estimate is determined by the sum of the variances of these two fluxes plus twice the covariance between them: $C_{s(f,i;f,i)} + C_{s(NEE,i;NEE,i)} - 2 \times C_{s(f,i;NEE,i)}$ where $C_{s(f,i;NEE,i)}$ will be negative. If the posterior covariances were larger in magnitude (resulting in large negative correlations), then the inversion would be less able to distinguish between these two fluxes. At the moment the dominant term is the large uncertainty prescribed to the NEE estimates. Therefore, even though the negative correlations are

30    small, the posterior variance of $s_{f,i} - s_{NEE,i}$ is still large, therefore the inversion does not distinguish well between NEE and fossil fuel fluxes under the current configuration.

But when we aggregate the fluxes from the same pixel, the variance of NEE plus the fossil fuel flux from the cell would be $C_{s(f,i;f,i)} + C_{s(NEE,i;NEE,i)} + 2 \times C_{s(f,i;NEE,i)}$ where $C_{s(f,i;NEE,i)}$ will be negative. The inversion always results in reductions in the uncertainty of the individual fluxes (or at worst they remain the same). If posterior covariances between the flux uncertainties were larger, which would occur if the uncertainties prescribed to the prior fossil fuel and NEE fluxes were smaller or if the elements of the observation error covariance matrix were smaller, then there would be a much smaller uncertainty in the aggregated flux estimates. The negative posterior covariances which are produced by the inversion mean that the uncertainty of the aggregated flux is always smaller than the sum of the uncertainty of the individual components. But by how much smaller depends on the prior uncertainties and the skill of the atmospheric transport model.

"*Figs. 15+16: I cant see the value of those figures. I don't think they are even referenced in the text?*"

Only Figure 15 is retained. It was an oversight that these were not originally reference, and this has been corrected. The discussion has been expanded as explained above. See also the response to Referee 1 on their comment marked B.

"*Section 3.2.3: Can you please check the units? You refer to fluxes, but units are in ktCO2.*"

The units are $\mathrm{kt\,CO_2\,week^{-1}}$ emitted from the domain of the inversion. This has been clarified.

"*Fig. 17: Please add a colour legend. Check units.*"

This has been clarified and legend added.

"*Section 3.2.4: Units not consistent. I think, they are all fluxes.*"

This section has been merged with section 3.2.3 which now considers the aggregated fluxes. The units have be made clear to reflect that for those fluxes referred to in the original section 3.2.4, the units are $\mathrm{kt\,CO_2\,month^{-1}}$.

"*Table 5: move to supplementary material?*"

This whole section has been moved to the supplementary material.

**References**

Ziehn, T., Nickless, A., Rayner, P. J., Law, R. M., Roff, G., and Fraser, P.: Greenhouse gas network design using backward Lagrangian particle dispersion modelling – Part 1: Methodology and Australian test case, Atmos. Phys. Chem., 14, 9363–9378, doi: 10.5194/acp-14-9363-2014, 2014.

140

---

## Author Response (AR1)

**Response to interactive comment by Anonymous Referee 1 on "Estimates of CO2 fluxes over the City of Cape Town, South Africa, through Bayesian inverse modelling" by Alecia Nickless et al.**

Alecia Nickless, Peter J. Rayner, Francois Engelbrecht, Ernst-Günther Brunke, Birgit Erni, and Robert J. Scholes

We would like to thank the referee for their time and thorough consideration of the paper. We would like to thank the referee for their support of this work. With regards to the issues raised concerning the flow of the manuscript and redundancies highlighted by the referee, we have taken these comments on board, and performed a thorough rewriting of the manuscript which presents a more cohesive and focused presentation of the results of this reference inversion. The main purpose of this

5 paper is to introduce the inversion framework used for the Cape Town inversion and to present the initial results for the reference inversion. The manuscript was rewritten to reinforce this focus. The sensitivity analyses will be presented in a related paper, and will rely on this introduction of the methodology. This paper focuses on the specification of the covariance matrices. Originally we had intended to include this work together with the reference inversion, but it soon became apparent that it would be too much content. Some of the discontinuities in the original manuscript are likely due to this change in focus of the paper. This

10 was corrected in the revised manuscript.

As the manuscript has largely been rewritten to improve clarity, we will not focus on individual sentences identified by the referee, as many of these will be entirely changed or dropped from the manuscript.

Firstly we will address the main scientific concerns (labelled A to D by the referee).

The referee states *"A.)Figures 9a,c and 10a,c indicate very large misfits between the model and the measurements when*

15 *using the prior fluxes while, as highlighted by the manuscript itself, one does not expect a very large variability of the CO2 in the area. How is it possible to get such an amount of misfits larger than 30 ppm (and in the range 100 to 200 ppm) while the data nearly never reach an excess of 30 ppm over their baseline, and while the measurement stations seem to be quite distant from the city major point sources? The manuscript ignores that such prior misfits strongly question the reliability of the atmospheric transport modeling framework, and thus of the inversion system.*

20 *The authors say that the model "shows ability to track local events at the sites" but it is impossible to assess on Figures 9 and 10. Furthermore, given the very large size of the control vector (and thus the very large number of degrees of freedom in the inversion system), it is not really surprising to see that the inversion manages to fit the data to a far better extent. I find it difficult to take it as a demonstration that the atmospheric transport model is reliable. In particular, opposed to what is said on page 1 of the supplementary material ("some evidence to provide confidence in the modelled meteorology is provided in*

25 *this section"), the figures 1 and 2 of this supplementary material strongly question this reliability, displaying very large misfits between the modeled and measured timeseries of wind speed, with a weak correlation between them. The difference between*

*the height of the measurement sites and the model vertical representativeness can hardly explain such misfits and even less such a low correlation. This requires a deeper analysis or better insights regarding the skill of the 1 km resolution CCAM model at city scale from studies like Engelbrecht et al. (2009, 2011).*

*By gathering the night-time and daytime data both in their analysis and in the inversion system, the authors do not help investigating this issue. We can assume that the largest misfits are obtained at night. However, opposed to what is said on page 1 line 12 ("Night-time observations were included, but allocated much larger errors compared to the daytime observations"), figure 5 indicates that the increase of the observation error at night is likely far from sufficient to cope with the increase of model errors at night. I think that the authors have overestimated their ability to assimilate night-time data. Analysis of the misfits between the modeled and measured $CO_2$ and of the corrections to the fluxes applied by the inversion at night vs day could help investigate this topic."*

The issues raised here are concerning the poor match between the prior modelled concentrations and the observed concentrations, as well as the results of the attempt at validating the wind information from CCAM in the supplementary material. Lastly the justification of the allocated uncertainties for the night-time observations is questioned.

The referee is concerned with the reliability of the atmospheric transport modelling framework. In retrospect, presenting the adhoc validation of the CCAM wind fields was unwise. Unfortunately the wind measurements we have available to us were not adequate to compare with the modelled winds from CCAM. The locations of the weather stations were far from the sites, except for Cape Point. All the locations for the weather station sites would not have been representative of the 1 km by 1 km pixel from which the modelled winds would have been extracted, or were situated in locations that would have been strongly influenced by the local topography or built environment. The anemometer located at the top of the Cape Point measurement tower is subject to winds influenced by the surrounding topography (Figure 1). As stated in Whittlestone et al. (2009), the single point measurements at Cape Point were strongly influenced by local topography and measured wind directions showed little correlation with the true source of the air mass: "Two superficially attractive selection criteria proved to be ineffective. One, wind direction, was so perturbed by local topography that there was no correlation of the measured wind direction with the bearing of the source of more distant trace gases from the critical north to east sector. The other, back trajectories, were effective in determining if contact with the southern African continent had occurred, but the indicated time and location of land contact was highly inaccurate. We make use of the Cape Point site for background concentration information only, and the inversion does not rely on correctly modelling transport to this particular site. Our two sites are slightly less extreme in the surrounding topography, but nonetheless, still pose significant challenges for obtaining wind speed and direction measurements that would be representative of a grid square produced by a regional climate model. All the weather station sites are located near the shoreline, therefore subject to sea-breeze variations.

[Figure]

**Figure 1.** Image of topography around Cape Point GAW station. http://www.imk-ifu.kit.edu/319.php

To justify the use of CCAM to provide modelled winds, we rely on previous studies which have used this model for atmospheric transport modelling in our target area (Whittlestone et al., 2009), and studies which have validated CCAM at different spatial resolutions (Engelbrecht et al., 2009; Roux , 2009; Engelbrecht et al., 2011, 2013, 2015). In particular, CCAM has been able to satisfactorily recreate present-day rainfall totals and the rainfall seasonal cycle, as well as circulation patterns over
5 South Africa (Engelbrecht et al., 2009), and has been able to simulate with some success mid-tropospheric closed-lows and extreme rainfall events (Engelbrecht et al., 2015). CCAM has been validated over the Stellenbosch wine producing area, which falls within the domain of this inversion, with respect to temperature, relative humidity and wind speed at six different stations within this region (Roux , 2009). Validating the wind product from CCAM in a rigorous manner is beyond the scope of this paper. Our interest is in the impact of the estimates of the uncertainties on the inversion results, and we have focussed our
10 sensitivity analyses on this question. The discussion in the manuscript reviewing CCAM's capabilities has been expanded to give a more thorough examination of this literature (section 2.4 in the revised manuscript).

Whittlestone et al. (2009) used CCAM to generate the wind data as well as to perform the transport modelling over Cape Point for their investigation. We instead relied only on the wind and other climate variables from CCAM, such as temperature which is well validated (Engelbrecht et al., 2009), and used the well known Lagrangian Particle Dispersion Model (LPDM) (Uliasz, 1994; Lauvaux et al., 2009; Lauvaux et al., 2016) for the transport modelling. This model has been used for several

5   inversion exercises at various spatial scales.

The more likely candidate for the poor agreement between the observed and prior modelled concentrations is the specification of the prior fossil fuel and net ecosystem exchange (NEE) fluxes. The information available to perform a spatially and temporally disaggregated fossil fuel emission inventory for the City of Cape Town is limited. In addition, Cape Town is a city with stark inequalities between the population subgroups. For cities in developed countries it can generally be assumed that

10   almost all heating and lighting will be generated through electricity consumption, but for many of the communities in Cape Town, this is not the case. These communities will rely heavily on raw fossil fuel burning for heating and lighting. Accessing this information and converting this into emissions based on the assumed behaviour of people is not factored into standard inventory analyses and is beyond the scope of this paper. Therefore, the residential emissions are a large contributor to the fossil fuel emission budget as well as one of the largest contributors to the uncertainties in the fossil fuel flux. The relevant

15   question is whether we have successfully captured these uncertainties in specification of the prior uncertainty. The discussion of this issue has been included in section 2.9.1.

Another significant uncertainty is the simulation of the Fynbos biome. This biome is biodiverse, with many endemic species, but covers a relatively small area in South Africa, but a significant area within the domain of the inversion. The fynbos biome is poorly represented by dynamic vegetation models. The land atmosphere exchange model CABLE was selected to couple

20   with CCAM due to its use and development in regions of Australia which share similar characteristics to the savanna biome in South Africa, which has a coverage of over 50%. Its ability to simulate respiration and photosynthesis in the fynbos region is largely untested. This discussion has been expanded in the revised manuscript in sections 2.7 and 2.9.2.

We can test whether our assumed uncertainties are consistent with the prior misfit in concentration (see Michalak et al. (2005) for details). The magnitude of the discrepancies that we obtained between the observed concentrations and the modelled

25   concentrations are expected. If we calculate the matrix $\mathbf{HC_{s_0}H}^T + \mathbf{C_c}$, and assume $\mathbf{H}$ is correct, the uncertainty in prior fluxes is projected into uncertainty in modelled concentrations. The square root of the diagonal elements have a similar distribution to the absolute mismatches between the observed and prior modelled concentrations. This has been included at the beginning of section 3.1.

To confirm that the prior information is the major cause of the disagreement (although not having perfect transport modelling

30   is to blame as well), we performed a sensitivity analysis where the NEE estimates were averaged over the domain, and the prior estimates for NEE set as the overall average for all pixels. Similarly, the uncertainty of the NEE estimates was set as the overall average of the net primary productivity (NPP) estimates. At the Hangklip site, which is dominated by the NEE contributions, this led to far smaller discrepancies between modelled and observed concentrations (compare Figures 2 to 3). Therefore it appears that CABLE may be overestimating the amount of photosynthesis or respiration (or both) that are taking place in

35   the region. On the other hand, Robben Island, which is far more influenced by the Cape Town fossil fuel emissions, did

not show any improvement in the discrepancies between the prior and observed concentrations (compare Figures 4 to 5). In the sensitivity analysis paper, we present further tests based on the specification of the fossil fuel emissions and NEE fluxes and their uncertainties. These timeseries plots have been included in revised manuscript (Figures 6 and 7), as well as in the supplementary material in section 1.4.

5     The plots of the residuals between the modelled and observed concentrations are not very different between day and night (see the Figures 2 and 4). Therefore it does not appear that the difficulty in modelling transport at night is leading to significantly larger discrepancies between the observed and modelled concentrations, as suggested by the referee. Specifying uncertainties that are too small for the night time model errors does not appear to be leading to the large discrepancies we are observing. The time series plots have been given for only a short period in order to avoid squashing the time axis, and to better demonstrate the

10 inversions ability to obtain posterior modelled concentrations that are much closer to the observations. Although this is entirely expected, we wanted to demonstrate that as far as optimising the flux estimates to better match the observed concentrations, the inversion was at least achieving this. The expanded time series plots better show the ability of the inversion to track local "pollution" events.

    The revised manuscript now includes more discussion on the discrepancy between the observed and prior modelled concen-

15 trations. To deal with the problems related to viewing the observed and modelled concentration data in Figures 9a,c and 10a,c, the time series plots have been altered. They are expanded over several panels in order to stretch out the time axis, allowing the reader to better assess the discrepancies and similarities between the observed and modelled concentrations, as shown in Figures 2 to 5. We included the diurnal plots to demonstrate how the prior and posterior modelled concentrations were differing from the observed at different hours of the day.

[Figure]

**Figure 2.** A time series of the observed and modelled $CO_2$ concentrations for April and May 2012 at Hangklip separated by day and night

[Figure]

**Figure 3.** A time series of the observed and modelled $CO_2$ concentrations for April and May 2012, where the prior estimates of NEE have been set as the average NEE over all pixels, at Hangklip separated by day and night

[Figure]

**Figure 4.** A time series of the observed and modelled $CO_2$ concentrations for April and May 2012 at Robben Island separated by day and night

[Figure]

**Figure 5.** A time series of the observed and modelled $CO_2$ concentrations for April and May 2012, where the prior estimates of NEE have been set as the average NEE over all pixels, at Robben Island separated by day and night

The second major issue that the referee raised was:

*"B) One of the main discussion of the manuscript is related to the lack of distinction between the anthropogenic and natural fluxes in the inversion results. What puzzles me is that the analysis of the posterior error covariance matrices should be a very helpful tool to feed such a discussion. The authors display correlations between uncertainties in the emissions at a given pixel and the NEE field in Figure 16 but ignore them when conducting this discussion.*

*Actually, Figure 16 shows correlations that are very close to 0, which undermines the assumption of the lack of distinction. When looking at the station locations and at the situation of the city vs. the areas of high NEE (which are also the areas of high prior uncertainties in the NEE and of high corrections to the NEE in the inversion), it is difficult to understand why the separation between the NEE and the anthropogenic emissions should be so problematic for the inversion. The authors have used very large prior uncertainties in the NEE, and the NEE dominates the mean diurnal cycles of the stations. This explains why, on the first order, the inversion focuses on the NEE rather than on the anthropogenic emissions (from that point of view, it would not be a problem of distinction between NEE and anthropogenic emissions, but rather a problem of detection of the emissions despite the dominating signal from NEE). But, according to figures 3 and 4, this should not prevent the inversion from getting a signal that is dominated by the anthropogenic emissions when the wind blows roughly from one station to the other one through Cape town. Paradoxically, the type of "gradient approach" that the author assume to be useless for their study case (p13 line18 p61 lines7-9) may help them to cope with the NEE signal. All of this needs to be better analyzed. The analysis of the variations of the modeled contribution from the NEE vs that from Cape Town at the measurement sites could be very useful."*

The referee is concerned about the conclusions regarding the difficulty of separating NEE and fossil fuel fluxes correctly in the inversion. This is an important concern although we do not agree with the metric the referee chooses. We believe that the correct metric is the uncertainty of the difference between the two flux components, not the correlation. To see this, imagine the case where two flux components from the same pixel are given independent priors (no prior correlation) and are not all visible to the observation network. Nothing will have been learned about the fluxes, including their difference, but the posterior correlation will be zero. Unfortunately, this is close to our present case for many pixels. There is not enough information available to the inversion for it to correctly assign contributions from NEE and fossil fuel fluxes. We solve for these contributions separately in the inversion so the flexibility exists for adjustments to be made to both of these fluxes while attempting to obtain sensible adjustments to the overall flux within the pixel. As the referee also mentions, this is made more difficult by the large uncertainty that we have assigned to the NEE estimates (more on this in the third issue raised).

The covariance between fossil fuel and NEE flux uncertainties are small because the uncertainties in the prior modelled concentrations that are attributed to the flux contributions ($\mathbf{H}\mathbf{C_{s_0}}\mathbf{H}^T$) are small relative to the uncertainties specified for the modelled concentration errors ($\mathbf{C_c}$). This is not because our prior uncertainty is small but because the transport Jacobian only projects fluxes from individual pixels weakly into modelled concentrations. As the uncertainty in the modelled concentration errors is decreased, the size of the posterior off-diagonal covariance elements between the fossil fuel and NEE flux uncertainties from the same pixel increases. This can easily be confirmed through the use of a toy inversion system using typical values for $\mathbf{H}$, $\mathbf{C_{s_0}}$ and $\mathbf{C_c}$ from our inversion system. The posterior variance of any linear combination of terms from the source vector

of the fluxes (including the difference between two fluxes from the same pixel) will always be reduced or (at worst) left unchanged relative to the prior variance of the same linear combination of elements (Jackson , 1979; Jackson and Matsu'ura , 1985). Observations may very well introduce correlations between flux components but this does not mean a reduction in the ability to distinguish between them.

5   If we define the distinction between the fossil fuel flux and NEE flux within the same pixel $i$ as the variance of the difference between the fossil fuel and NEE fluxes $s_{f,i} - s_{NEE,i}$, this will be equal to the sum of the variances of these two fluxes minus twice the covariance between them: $C_{s(f,i;f,i)} + C_{s(NEE,i;NEE,i)} - 2 \times C_{s(f,i;NEE,i)}$ where $C_{s(f,i;NEE,i)}$ will be negative. Therefore the posterior uncertainty of the difference in these fluxes is always going to be larger than the sum of the individual posterior flux uncertainties, but smaller than the prior uncertainty of this linear combination of terms. Therefore, although the

10  off-diagonal terms may be small, the ability of the inversion under the current framework to detect between NEE and fossil fuel fluxes is limited as the posterior uncertainties are still large, and therefore the uncertainty of $s_{f,i} - s_{NEE,i}$ is large. If the covariance terms are small because, relative to the errors in the modelled concentrations the contribution of the uncertainty in the fluxes to the uncertainty in the modelled concentration is small, then the variance of $s_{f,i} - s_{NEE,i}$ is still going to be large due to the dominance of uncertainty in $s_{NEE,i}$.

15  On the other hand, when we aggregate these fluxes from the same pixel to get the total flux within a pixel $s_{f,i} + s_{NEE,i}$, the uncertainty of this term is equal to $C_{s(f,i;f,i)} + C_{s(NEE,i;NEE,i)} + 2 \times C_{s(f,i;NEE,i)}$ where $C_{s(f,i;NEE,i)}$ is negative. We already know that the sum of any linear combination of sources will have a smaller uncertainty after the inversion, but when we aggregate fluxes from the same pixel, the uncertainty of this total is smaller due to the smaller posterior variances and also because the covariances are negative. This demonstrates that the value of the inversion is to reduce the uncertainty on each of

20  the individual fluxes and to additionally reduce the uncertainty of the aggregation of the NEE and fossil fuel flux within the same pixel. The reduction in the uncertainty of the sum of fluxes within the same pixel is going to depend on the size of the uncertainty of the NEE flux, which is usually the larger uncertainty. To improve the ability of the inversion to estimate the total flux within a pixel, we need to improve the skill of the atmospheric transport and we need to reduce the uncertainty in the estimates of the NEE. As it stands, with a large prior uncertainty in the estimation of the NEE fluxes from the CABLE

25  model which remains a large posterior estimate after the inversion, the distinction between the fossil fuel and NEE flux from the same pixel is not very different from the prior estimate of the variance of the difference between the fossil fuel and NEE flux, $C_{s_0(f,i;f,i)} + C_{s_0(NEE,i;NEE,i)}$.

We would like to thank the referee for the suggestion of adding the investigation of what contributions NEE and fossil fuel fluxes make to the modelled concentrations at each site. This analysis has now been included into the manuscript. This

30  discussion is at the end of section 3.2. Figures 9 and 10 have now been included which provide information on the corrections that inversion is making to the contribution to the modelled concentration from the fossil fuel and NEE fluxes. An addition discussion is provided in the supplementary material (section 1.8).

We still have a long way to go before we can reliably estimate the fossil fuel fluxes from the City of Cape Town using this inversion framework, but our intention is to provide the building blocks for such an inversion system which would allow better

35  products for fossil fuel emissions and NEE fluxes to be slotted in as they become available, while we work towards a reduction

in the misfit between the prior and posterior modelled concentrations. For example, we could replace the bespoke inventory analysis used here for the global $1 \times 1$ km ODIAC fossil fuel product (Oda et al. , 2017) which has recently been used for Indianapolis. This has been performed for the sensitivity analysis paper.

The referee mentions that the gradient approach may have been more useful in the Cape Town inversion than implied in the original manuscript. The main concern we had about using the gradient method in these circumstances was that the direction of travel of an air parcel between the two sites would not necessarily be in a straight line due to the topography of the Cape Town region. Therefore selecting observation pairs based on the wind direction would not necessarily have given the true gradient in concentration between the sites. We supplied a map of the sensitivities of the observations sites to the surface fluxes to show this in the original manuscript (now Figure 17 in the revised manuscript). To further add to this argument, we provided figures of the average wind speed and direction for the domain for each month in the supplementary material (section 1.3). These figures show that, in general, the wind direction would not be in our favour, and with only two sites, that would leave very little information to constraint the surface fluxes. When the wind is blowing from the Hangklip site, it curves northwards towards the interior and away from Cape Town. When the Robben Island site is observing clean marine air before coming into the Cape Town area from the Atlantic side, such as June 2013, the wind changes from the north westerly direction once it passes over Cape Town to a more northerly direction, missing the Hangklip site. These wind plots have replaced the original attempt at validating the CCAM wind data.

The third major issued related to the NEE uncertainty estimates. These were set to be large (relative to the NEE estimate), and we used the estimates of the productivity associated with the NEE estimate as the uncertainty. More precisely, which is made clearer in the manuscript, as the uncertainties were scaled so that the $\chi$-squared goodness-of-fit statistic was closer to 1, the productivity values (NPP) were used to define the relative uncertainties between the NEE estimates. There is a typo in the original manuscript which states we used Net Ecosystem Productivity for the uncertainties where it should be Net Primary Productivity. The NPP fluxes were squared to give initial estimates of the variances. The fossil fuel and NEE variance estimates were then doubled so that the Chi-squared goodness-of-fit statistic was closer to 1, and these scaled variances provided the final uncertainties assigned to the prior fluxes. These scaled uncertainties were correctly displayed in prior information figures, but not made clear in the text.

*"C) Regarding the prior uncertainties in the NEE, the relative values discussed in section 2.9.2 can be very large. They deserve some justification based on the CABLE validation studies, especially since they will be amplified by the multiplication of the prior uncertainties by a factor of 2 in section 2.11 (Figure 4 being misleading). I understand that when aggregating them over the modeling domain, we get much smaller relative values due to using a small spatial correlation length scale. However, could not it be an issue for the results at the pixel scale and, implicitly, for the control of the highly localized anthropogenic emissions (see the strange correction patterns in figure 13 and 14) ?"*

We certainly agree that the large uncertainty we have in the NEE estimates produced by CABLE is a limitation and inhibits the ability of the inversion to apply corrections to the fossil fuel fluxes, in which we are the most interested. Modelling NEE in this region is uncertain. Wang et al. (2011) have shown that unless CABLE is closely calibrated for a specific system, it can lead to significant errors in the estimation of NEE. There are many land types in the region, which include the endemic fynbos biota, different types of agriculture, as well as fallow land. Although a great deal of work is being carried out to validate CABLE over the savanna biome, which covers much of South Africa, the information available for how well CABLE behaves for fynbos is limited. In the sensitivity analysis we will consider other models of NEE for this region, but for the reference inversion, we think that a conservative estimate of the error is best. Sensitivity analyses will consider the impact of reducing the uncertainty estimates to a smaller fraction of the NPP estimates from CABLE.

The final major scientific issue that the referee raises concerns the inventory analysis:

*"D) I am not sure to understand the distribution of the emissions from Cape Town according to the author's inventory. p11 says "But of the carbon emissions due to energy usage, only 27% were attributed to the transport sector as a result of the carbon intensive usage of coal for electricity generation to provide almost all of the energy to the residential and commercial sectors in South Africa, which emit approximately 29% and 28%, respectively, of the total carbon emissions of CT (City of Cape Town, 2011)." Paying much attention to the terms "electricity generation" and "almost all of the energy", my understanding of this sentence is that there is a large number of coal power plants within the city bounds (otherwise the part of the emissions within the city due to the transport would be very high), which represent almost 60% of the city CO2 emissions, while the direct emissions from the residential and commercial areas should be very low. However, this seems strongly at odd with the figures*

*and discussions of this manuscript and this would be highly problematic for the atmospheric inversion. Could the authors clarify this point "*

The referee is concerned about contributions to the fossil fuel budget from the different sectors of the city. The percentages that are quoted here come from a report on the energy consumption of the city, and do not entirely relate to the direct fossil fuel emissions from the city. Emissions from coal are small because most of the power generation capacity through coal occurs in the north eastern provinces of South Africa. Residential emissions are not negligible in Cape Town because many of the communities still rely on burning raw fossil fuels for heating, cooking and lighting. This discussion has been made clearer in the revised manuscript. Instead of discussing these energy statistics which are already discussed in the inventory paper, the discussion in the revised manuscript now reflects the percentage contributions as reflected in the inventory data available for this inversion. Using these statistics, the emissions from industry (based on the available fuel usage data) are 12.0%, 34.6% from vehicle road transport, 51.0% from the residential sector, and 2.4% from the airport and harbour. This discussion has been updated in section 2.6.

The referee lists a number of additional issues which have been corrected in the revised manuscript.

*"Section 2.1 makes a rough account of the traditional theoretical framework of the inversions (e.g. sentences like "If we assume a Gaussian error distribution for the surface fluxes and concentrations we obtain the following cost function for our least squares problem" on page 6). Throughout the manuscript, the covariances between uncertainties in fluxes are often called covariances between fluxes."*

The description of the Bayesian inversion framework has been rewritten in the revised manuscript (section 2.1). With regards to the covariances, the wording has been clarified.

*"- in the abstract and first sections, the text introduces the concept of "boundary concentrations" without specifying that the boundaries relate to the modeling framework. This becomes problematic when explicitly speaking about the sensitivities of the measurements to the boundary concentrations (e.g. l23-24 p5)"*

We have made this description clearer in the manuscript to reflect that we are referring to the concentrations at the boundaries of the modelling domain when we refer to "boundary concentrations".

*"- the text often uses the terms "sources" and "emissions" while speaking about (natural) fluxes that can be negative"*
This has been corrected in the manuscript.

*"- p10 line17-19 the sensitivity is not the influence, H is not HTtranspose"*
This has been corrected and the description of the methodology made clearer".

*"- p11 "and allowed for small scale transport features to be maintained in H": using a coarse resolution control vector does not remove the small scale transport features in H"*

This has been removed from the sentence.

*""- Section 2.11 states that the error covariance matrix that is underestimated in the first configuration according to the chi test is necessarily B ("and values greater than one indicate that the variance prescribed is lower than it should be and therefore the posterior estimates will be over-constrained by the prior fluxes") while it could be R (and actually some of the results favor the assumption that it is R)."*

As discussed above in response to the referees concerns regarding the atmospheric transport modelling, we think that the prior information is the most uncertain in our inversion system. We consider other configurations of the covariance matrix for the errors in the modelled concentrations in the sensitivity analysis paper.

The individual sentences identified by the referee on pages C6 to C8, and similarly problematic sentences elsewhere in the manuscript, have been amended or cut from the revised version.

*"- The design of the figures should be improved. The location and name of the sites are hardly visible in figure 1. Labels are too small and the fields are fuzzy (mainly due to the choice of the colorbars) in figures like Figure 3. The choice of the colors in figures like Figure 8 is poor: on my screen, it is really hard to distinguish between the different curves. In figures like Figure 9, it is impossible to analyze the different timeseries since they are compressed along the x axis (with nighttime and daytime data mixed together) and most of the measurements are hidden behind the model patches. In most of the figures, there is a lack of subtitle and legends to help the reader while the captions are sometimes quite complex (e.g. the legend of figure 15). Therefore, in general, the figures are very difficult to read. In Figure 15: it is difficult to see the pixel against which covariances are computed."*

Figures in the manuscript have been replotted to address the concerns highlighted above. The time series have been expanded to multiple panels as explained early, and the day and night-time concentration and residuals separated. Better use has been made of legends and subtitles in the figures, to avoid including too much information in the caption. The maps have been improved so that labels are easier to read. The colour scheme has been amended to the rainbow colour scheme. In the original manuscript, the purpose of deliberately not using a smooth colour scheme was to allow very large fluxes, like those from point sources, to be distinct from smaller fluxes, while still allowing differences in these smaller fluxes to be distinguished, such as subtle differences in the residential and transport fossil fuel fluxes. We have now used the log scale where necessary, instead of customised graduations in the colour bar. The pixel against which covariances are computed in Figures 15 and 16 has been clearly identified (now just Figure 15 in the revised manuscript).

*"- The notations used in several equations are not really optimal. At least, they do not help understand the meaning of the variables, e.g. Etrans in equation 11 which refers to a subcomponent of the transport error, while Eobs refers to another part of the transport error, and not to the observation error (which is the sum of the transport and measurement errors). Eq 8 is not really adapted to equations 1 and 6. Eq 9 and 11 are informal."*

The notation for these equations has been amended to be more consistent with the rest of the manuscript. These are now equations 9, (10 and 11) and 13.

*"- There are too many significant digits in tables 3 and 4 which makes these tables difficult to read. Would not it be better to show the content of these tables using plots ? I do not understand why the authors produce distinct sections (3.2.3 and 3.2.4)*
5     *and tables (3 and 4) for the variations of the 1-week mean and 1-month mean flux budgets. This is a source of redundancies and I don not think that they manage to bring specific insights for each of the two timescales."*

Table 3 has been converted into a time series of box plots to show the prior and posterior distribution of pixel-level fluxes over the domain. These fluxes are also presented as maps in the supplementary material (section 1.7). Table 3 was produced to deliberately consider the summary statistics of the pixel-level flux estimates within the domain, and to show how these
10     summary statistics differ between months. The objective of section 3.2.2 was to address how the inversion was updating fluxes at the pixel level, and from month to month. In this section we were comparing the prior estimates $\mathbf{s_0}$ with the posterior estimates $\mathbf{s}$. Therefore the unit of the fluxes are $\mathrm{kg\ CO_2 m^{-2}\ week^{-1}}$, which is unit of the fluxes solved for by the inversion. Sections 3.2.3 and 3.2.4 were aimed at addressing what impact the inversion had on the total flux of $CO_2$ over the whole domain. These two sections have been merged. The purpose of the month estimates was originally for the sensitivity analysis
15     to compare between configurations.

As discussed earlier, the inversion has the most impact in reducing uncertainty when aggregating over fluxes in the domain, and the main objective of Table 4 is to show this uncertainty reduction. When aggregating fossil fuel and NEE fluxes, and aggregating over all pixels, the reduction in the posterior variances and the negative posterior covariances brought about by the inversion can be taken advantage of to produce posterior total estimates which have associated uncertainties which are much
20     smaller than those of the prior total estimates.

*"- The acronyms CT and CBD are not defined explicitly."*
This has been corrected.

*"- Section 2.2. and Equation (6) are confusing regarding the composition of the control vector (regarding the fact that the inversion solves for the fluxes at the transport model spatial resolution and regarding the control of the average conditions for*
25     *each of the 4 lateral boundaries). We need to guess it from the numerical derivation of the size of the control vector or wait for sections 2.5 to get clearer details. The situation is similar regarding the fact that 1-month inversions are conducted to cover the 13 month period."*
This has been made clearer in the revised manuscript.

*"- there is a problem with the order of the citations (see Tarantola (2005) and Enting (2002) and Lauvaux et al., 2016; Bréon*
30     *et al., 2015 on page 5)"*
This has been corrected.

*"- the percentile filtering technique at Cape point and its impact on the timeseries at this site is not well detailed (e.g. on which time windows, at which timescales is it applied ?), while the station can be influenced by Cape Town, and by the NEE in the region covered by the modeling domain. This is perturbing since the system controls the North and East boundary conditions that are inland (and thus separated from the Cape point station by large areas of NEE and potentially influenced by even larger areas of NEE outside the modeling domain) and since it uses the data filtered at Cape point to provide a prior value with a low prior uncertainty to these conditions."*

More discussion has been added on the percentile filtering technique (seciont 2.8 of the revised manuscript). A detailed description of this technique is provided in (Brunke et al., 2004): "The background signal at Cape Point, obtained from a percentile filtering technique (Brunke et al., 2004), was used as the prior estimate of the concentrations at each of the four domain boundaries. The percentile filtering technique removes data influenced by the continent or anthropogenic emissions. Two 11-day moving percentiles, which are adjustable by tuneable factors, control the upper and lower threshold limits. This results in a subset of background measurements from Cape Point represented by a narrow concentration band contained within these limits. This filter, when applied to the Cape Point $CO_2$ measurements, selects approximately 75% of the data. The percentile-filtering technique has been shown to compare well with the more robust method of using contemporaneous radon ($^{222}Rn$) measurements to differentiate between marine and continental air.".

The purpose of the percentile filtering technique is to remove those measurements which are strongly influenced by either anthropogenic emissions from Cape Town (which are observed very seldom by the tower) or those measurements strongly influenced by biospheric uptake of $CO_2$. This discussion has been expanded in the manuscript. We have deliberately made the margin of the domain around the City of Cape Town large, which allows the inversion system to solve for fluxes at large distances from the City, rather than relying on estimates of the concentrations at the boundary. Therefore when the air arriving at the measurement site originates from the north the inversion can account for uptake by correcting the far-field fluxes and leave the concentration at the far North or East boundaries relatively unchanged. In this way, the boundary $CO_2$ concentrations act more like baseline concentrations.

*"- p20: the discussion on the representation error ignores the part of this error due to the difference of spatial representativeness between the measurements and the model ("We did not account any further for aggregation or representation errors as we did in the network design, as we were running the inversion at the same spatial scale as the transport model.")"*

The representation error is accounted for in the 2 ppm and 4 ppm assigned to daytime and night-time concentrations respectively. The discussion in the manuscript has been altered to better reflect this. The representation error occurs due to errors in the transport modelling. The distinction between aggregation error and representation error has been corrected in the manuscript (section 2.10).

As in response to the remarks by Referee 1, these sorts of redundancies have been removed through a thorough revision of the manuscript. The only details which are repeated in a few places relate to the structure of the control vector, as this is a distinctive feature of the inversion framework, and results in the dimensions of the matrices $\mathbf{C_{s_0}}$ and $\mathbf{H}$.

*"P6, L19: surface source grid "point" ?"*

What was meant here was "surface pixel"

*"P10, L5-L23: you talk about the generation of sensitivity matrix H, but you also mention sensitivity matrix T, which is confusing"*

This was a typo. It has been corrected in the revised manuscript. There is no matrix T.

*"P11, L14-L16: I don't understand this sentence. Please rephrase using shorter sentences."*

This paragraph has been rewritten:

Previous: "Since the surface sources are expressed as fluxes of carbon, the contribution to the concentration at the measurement site is expressed in the amount of carbon seen at the measurement site from a particular source. In the case of the boundary sources (or contributions from outside of the domain) which are given as concentrations, their contributions to the concentration at the measurement site are expressed as a proportion of their concentration, dependent on their influence at the receptor site. Ziehn et al. (2014) shows that by calculating the Jacobian which provides the sensitivities of observed concentrations to boundary concentrations, the boundary contribution can then be written as:"

Revised: "The fluxes from the surface pixels are expressed in $\mathrm{kg\ CO_2 m^{-2}\ week^{-1}}$ and are transformed through $\mathbf{H}$ into contributions to the concentration at the measurement site in units of $\mathrm{ppm}$. The inversion solves for the concentrations at the boundary of the domain. Ziehn et al. (2014) shows that the Jacobian which provides the sensitivities of observed concentrations to boundary concentrations can be calculated as:"

*"P11, L23: "For the network design ..." ? It is confusing that you refer to a network design here."*

This has been removed.

*"Figs.3+4: This figures need improvement. Please label the sub-plots. The information is provided in the caption, but it would be much easier to have it in the figure directly. The used colour scheme is probably not that helpful either."*

These plots have been revised to make better use of subplot titles and legends to reduce the amount of text in the captions, and the text has been made clearer. The colour scheme has been altered to improve clarity. The colour scheme in the original manuscript was deliberately chosen so that 1.) positive, negative and zero fluxes would be distinct; 2.) and the "jagged" graduation in colours so that the very large fossil fuel fluxes at point sources could be distinct, but also the smaller fluxes, such as those for transport and residential sources. These are smaller, but more numerous than the point sources, and we wanted these to be distinguishable from each other as well. The colour scheme has been changed to a rainbow scheme, which shows up the differences more distinctly. Use of a log scale has also been used where necessary.

*"Section 3.1: I found this section far too detailed. Maybe some of the content could be moved to the supplementary material?"*

We agree. This section has been moved to the supplementary material.

*"Fig6: Please provide a legend for symbols and colours within the figure."*

Legends for the symbols and colours used in all plots have been provided, instead of including this information in the caption. This particular figure has been dropped, and replaced with an expanded time series where day and night concentrations have been separated.

*"Fig7: Is there a reason why Oct12, Dec12, Jan13 are missing? I am not sure if the diurnal cycle for all months is required in the manuscript?"*

These are plots of the observed diurnal pattern in the concentrations. Unfortunately observations were not available for all months, and there were large gaps on the concentration data during these three months. The monthly plots have been moved to the supplementary material, and instead the average diurnal plot over all periods has been improved in the manuscript (Figure 8).

*"Table 2: Consider to move this table to supplementary material."*

Agreed. This will be moved together with Section 3.1. to the supplementary material.

*"P34, L8-L9: NEE or NEP?"*

The net primary productivity was used as the uncertainty in the net ecosystem exchange. This has been corrected in the manuscript. Generally, for most parts of South Africa, the NEE flux itself is very small, but could be resulting from relatively large respiration and photosynthesis fluxes. If the uncertainty in the NEE estimate was based on the NEE value itself, it would be unrealistically small. This discussion has been made clearer in the revised manuscript.

This paragraph has been removed in the revised manuscript and these details now appear in section 2.9.2. "The uncertainty in the biogenic prior fluxes was set at the absolute value of the net primary productivity (NPP) as produced by CABLE. This is a very large error relative to the prior estimate itself, but there is a great deal of uncertainty in both the productivity and respiration fluxes contributing to thee NEE flux (Wang et al. 2011)."

*"Figs. 9+10: Please provide a legend for symbols and colours within the figure."*

These figures have been changed altogether to ensure that the time series is not squashed, and to separate out the day and night concentrations and residuals. Better use has been made of legends. Please see response to Referee 1 for an example of the new time series plots. These are now figures 6 and 7.

*"Fig.12: Please label the sub-plots (i.e. north, east, south, west)"*

Thank you. This has been done. This is now Figure 11.

*"P42, L1-L3: Please rephrase this sentence."*

This sentence has been revised.

Previous: "The innovation of the inversion can be observed through the differences between the prior and posterior flux estimates, which we refer to as the innovations, and through the change in the uncertainty estimates."

Revised: "We refer to the difference between the prior and posterior flux estimates as the innovations. The impact of the inversion on the flux estimates can be assessed through the size and direction of these innovations and through the reduction in
5  the flux uncertainties."

"*Figs.13+14: Add labels and improve colour scheme.*"

These plots have been improved. As discussed earlier, the colour scheme was selected to ensure that positive, negative and zero fluxes were distinct. The jagged colour graduation was also selected so that large fluxes could be displayed together with small fluxes, while still showing subtle differences between the smaller fluxes. The colour scheme has been changed to the
10  rainbow scheme to make differences between pixels more distinctive. These are now Figures 12 and 13.

"*Table 3: Is this level of detail really required in the manuscript?*"

The purpose of including this table was so that the impact of the inversion could be assessed from month to month. This table attempts to summarise the distribution of the fluxes over the domain, and show how that distribution is changed by the inversion. This section has been largely rewritten. The table has been converted into a time series of box plots to show the
15  distribution of these pixel-level fluxes (Figure 14).

"*P49: I have difficulties understanding the whole section on this page.*"

This section has been largely rewritten. The main point that this section makes is that, under the current inversion framework, the posterior off-diagonal covariances are very small. This is because the uncertainties in the modelled concentrations that are attributed to the flux contributions ($\mathbf{HC_{s_o}H}^T$) are small relative to the uncertainties specified in the observation error
20  covariance matrix ($\mathbf{C_c}$). If the diagonal elements in $\mathbf{C_c}$ could be made smaller, then the posterior covariances would be much larger in magnitude. When we wish to estimate the difference between the fossil fuel and NEE fluxes $s_{f,i} - s_{NEE,i}$, in other words, to distinguish between the fossil fuel and NEE fluxes in the same pixel, the variance of this estimate is determined by the sum of the variances of these two fluxes minus twice the covariance between them: $C_{s(f,i;f,i)} + C_{s(NEE,i;NEE,i)} - 2 \times C_{s(f,i;NEE,i)}$ where $C_{s(f,i;NEE,i)}$ will be negative. If the posterior covariances were larger in magnitude (resulting in
25  large negative correlations), then the inversion would be less able to distinguish between these two fluxes. At the moment the dominant term is the large uncertainty prescribed to the NEE estimates. Therefore, even though the negative correlations are small, the posterior variance of $s_{f,i} - s_{NEE,i}$ is still large, therefore the inversion does not distinguish well between NEE and fossil fuel fluxes under the current configuration.

But when we aggregate the fluxes from the same pixel, the variance of NEE plus the fossil fuel flux from the cell would
30  be $C_{s(f,i;f,i)} + C_{s(NEE,i;NEE,i)} + 2 \times C_{s(f,i;NEE,i)}$ where $C_{s(f,i;NEE,i)}$ will be negative. The inversion always results in reductions in the uncertainty of the individual fluxes (or at worst they remain the same). If posterior covariances between the flux uncertainties were larger, which would occur if the uncertainties prescribed to the prior fossil fuel and NEE fluxes were smaller or if the elements of the observation error covariance matrix were smaller, then there would be a much smaller

uncertainty in the aggregated flux estimates. The negative posterior covariances which are produced by the inversion mean that the uncertainty of the aggregated flux is always smaller than the sum of the uncertainty of the individual components. But by how much smaller depends on the prior uncertainties and the skill of the atmospheric transport model.

This discussion appears on page 45 of the revised manuscript, at the end of section 3.2.

5 "*Figs. 15+16: I cant see the value of those figures. I don't think they are even referenced in the text?*"

Only Figure 15 is retained (which is Figure 15 in the revised manuscript). It was an oversight that these were not originally reference, and this has been corrected. The discussion has been expanded as explained above. See also the response to Referee 1 on their comment marked B.

"*Section 3.2.3: Can you please check the units? You refer to fluxes, but units are in ktCO2.*"

10 The units are $\mathrm{kt}\ CO_2\ \mathrm{week}^{-1}$ emitted from the domain of the inversion. This has been clarified.

"*Fig. 17: Please add a colour legend. Check units.*"

This has been clarified and legend added. This is Figure 16 in the revised manuscript.

"*Section 3.2.4: Units not consistent. I think, they are all fluxes.*"

This section has been merged with section 3.2.3 which now considers the aggregated fluxes. The units in the original

15 manuscript were $\mathrm{kt}\ CO_2\ \mathrm{month}^{-1}$. We now consider only the aggregated weekly fluxes.

"*Table 5: move to supplementary material?*"

This whole section has been moved to the supplementary material.

**References**

[revised manuscript text omitted]

**1.4 Modelled Concentrations**

The time series for the modelled concentrations is discussed in Section 3.1 of the main paper.

[Figure]

[Figure]

[Figure]

[Figure]

**Figure 5.** The top 4 panels provide a time series of the observed, prior and posterior modelled concentrations at the Robben Island site. The time series is separated into day and night-time periods. The residuals between the modelled and observed concentrations, defined as the difference between the observed and modelled concentrations, are provided in the lower panel 4 panels. The first two months are presented in the main paper in section 3.1.

[Figure]

[Figure]

[Figure]

[Figure]

**Figure 6.** The top 4 panels provide a time series of the observed, prior and posterior modelled concentrations at the Hangklip site. The time series is separated into day and night-time periods. The residuals between the modelled and observed concentrations, defined as the difference between the observed and modelled concentrations, are provided in the lower panel 4 panels. The first two months are presented in the main paper in section 3.1.

**1.5 Diurnal Cycle**

The observed, prior and posterior modelled diurnal cycle, separated into working week and weekend $CO_2$ concentrations, are provided for each site and for each month in Figures 7 and 8. For all months, the diurnal cycle of the posterior modelled concentrations is relatively flat in comparison with the observed diurnal cycle, and usually sits at a higher mean level in the
5   case of Robben Island, and at a lower mean level in the case of Hangklip. Compared with the prior modelled concentrations, the posterior diurnal cycle matches better with the observed concentrations in terms of the peaks and troughs of the cycle and in terms of the mean level of the concentrations at each hour, although the posterior cycle still appears relatively flat in comparison to the observed cycle.

April 2013 at the Robben Island site provides an example where the prior modelled concentrations had working week
10   concentrations that were above those for the weekend during the early morning hours, whereas the observed concentrations showed the opposite situation. After the inversion, the posterior estimates had mean concentrations for the weekend that were above those for the working week during the early morning hours, matching better with the observed diurnal cycle.

Therefore the inversion does show an ability to improve estimates of the diurnal cycle, despite only separating the sources into day and night sources over a week period, and further separating the fossil fuel sources into weekend and week sources.

[Figure]

[Figure]

[Figure]

**Figure 7.** Diurnal cycle of the observed, prior modelled and posterior modelled $CO_2$ concentrations (ppm) at Robben Island, separated into working week (black) and weekend concentrations (grey), for each month with 95% confidence intervals, where the standard error is calculated over all measurements available for that hour of the day during that particular month.

[Figure]

[Figure]

[Figure]

**Figure 8.** Diurnal cycle of the observed, prior modelled and posterior modelled $CO_2$ concentrations ($ppm$) at Hangklip, separated into working week (black) and weekend concentrations (grey), for each month with 95% confidence intervals, where the standard error is calculated over all measurements available for that hour of the day during that particular month.

**1.6 Fossil Fuel and NEE Contributions**

The contributions of the fossil fuel and NEE fluxes to the modelled concentrations are provided for Robben Island and Hangklip for the months July 2012 to June 2013. As for March 2012 to June 2012, the posterior estimates of the NEE fluxes are increased in such a way that the uptake of $CO_2$ cancels out the emission of $CO_2$ due to fossil fuel sources. Little adjustment is made to the contribution from the fossil fuel fluxes to the modelled concentration by the inversion.

[revised manuscript text omitted]

**1.8 Toy Inversion**

Let us consider an hourly measurement at a single site, with a fossil fuel flux daytime source, a fossil fuel flux night-time source, an NEE flux from the same location, and an NEE flux from a neighbouring pixel. We wish to solve for these four fluxes and the covariance matrix of the uncertainties in these fluxes. Selecting some of the most extreme values for the uncertainties and for the sensitivities for the current inversion framework we could get the following:

$$\mathbf{H} = \quad (0.0, 0.0126, 0.00902, 0.0032); \quad \mathbf{C_{s_0}} = \begin{pmatrix} 233 & 0 & 0 & 0 \\ 0 & 78 & 0 & 0 \\ 0 & 0 & 1578 & 1220 \\ 0 & 0 & 1220 & 1578 \end{pmatrix}; \quad \mathbf{C_c} = 4$$

Solving for the posterior covariance matrix of the flux uncertainties using:

$$\mathbf{C_s} = \left(\mathbf{H}^T \mathbf{C_c}^{-1} \mathbf{H} + \mathbf{C_{s_0}}^{-1}\right)^{-1} \tag{1}$$

$$= \mathbf{C_{s_0}} - \mathbf{C_{s_0}} \mathbf{H}^T \left(\mathbf{H} \mathbf{C_{s_0}} \mathbf{H}^T + \mathbf{C_c}\right)^{-1} \mathbf{H} \mathbf{C_{s_0}}. \tag{2}$$

gives

$$\mathbf{C_s} = \begin{pmatrix} 233 & 0 & 0 & 0 \\ 0 & 77.8 & -4.2 & -3.7 \\ 0 & -4.2 & 1500.2 & 1151.1 \\ 0 & -3.7 & 1151.1 & 1517.0 \end{pmatrix}; \quad \rho_{\mathbf{matrix}} = \begin{pmatrix} 1 & 0 & 0 & 0 \\ 0 & 1 & -0.01 & -0.01 \\ 0 & -0.01 & 1 & 0.76 \\ 0 & -0.01 & 0.76 & 1 \end{pmatrix};$$

Although the sensitivity of the concentration measurement to the fossil fuel and NEE fluxes are not that different, the posterior covariances are small because the transport Jacobian only projects fluxes from individual pixels weakly into modelled concentrations. The uncertainties in the prior modelled concentrations that are attributed to the flux contributions ($\mathbf{H} \mathbf{C_{s_0}} \mathbf{H}^T$) are small relative to the uncertainties specified for the modelled concentration errors ($\mathbf{C_c}$). If we reduce the elements of $\mathbf{C_c}$ then the posterior covariances increase. For example, If $\mathbf{C_c} = 1$ then

$$\mathbf{C_s} = \begin{pmatrix} 233 & 0 & 0 & 0 \\ 0 & 77.2 & -14.5 & -12.9 \\ 0 & -14.5 & 1310.0 & 982.8 \\ 0 & -12.9 & 982.8 & 1368.0 \end{pmatrix}; \quad \rho_{\mathbf{matrix}} = \begin{pmatrix} 1 & 0 & 0 & 0 \\ 0 & 1 & -0.04 & -0.04 \\ 0 & -0.04 & 1 & 0.73 \\ 0 & -0.04 & 0.73 & 1 \end{pmatrix};$$

**References**

Lamigueiro, O., P.: Visualization methods for raster data, ver. 0.41. http://oscarperpinan.github.io/rastervis

---

## Author Response (AR2)

**Response to interactive comment by Anonymous Referee 2 on "Estimates of CO2 fluxes over the City of Cape Town, South Africa, through Bayesian inverse modelling" by Alecia Nickless et al.**

Alecia Nickless, Peter J. Rayner, Francois Engelbrecht, Ernst-Günther Brunke, Birgit Erni, and Robert J. Scholes

**The comments from Referee 2 are:**

*The responses provided by the authors have addressed the main issues raised by the reviewers. It is important to note here that the authors have produced new figures and results to support their responses. The only answer that lacks some clarity in the response is related to the first point raised by the referee. The discussion related to the large errors in NEE from CABLE*

5 *seems to explain the problem at the Hangklip site. It is less obvious at the Robben Island site where fossil fuel dominates the observed variability. My main concern, directly related to the reviewer's comment, is about the improvement after inversion, better illustrated in the latest manuscript by the residuals shown in Figures 4 and 5. The original residuals look as large as the actual variability at the site. However, the inversion seems to correct most of the initial mismatch, including the high-frequency variability. To the extent that the inversion should be able to reduce the mismatches, the actual improvement corresponds*

10 *to a high degree of confidence in the system. While the mean fluxes should clearly match the observations better, it is quite surprising to note the absence of peaks after inversion. One could worry about the degree of freedom in the system, allowing for these corrections after the optimization of the fluxes. The reviewer was concerned by the prior mismatches, comparable to the signals themselves, being reduced to an excellent match.*

*Your response here should focus on demonstrating that the observed improvement in the residuals is actually physically*

15 *realistic, and not a consequence of being over-confident in your inversion system. The reference to a companion paper is welcome but additional details in your prior error structures, or at least the degree to which your system can adjust to observed mismatches, would be a convincing argument to the observed improvement. The reference to Michalak et al. (2005) demonstrates that your errors are reasonable and in agreement with the observed mismatches. But one could argue that such an improvement after inversion may be the consequence of a high DFS, with innovations in space and time allowing to adjust*

20 *the mixing ratios with too much confidence, as described in the paper: "The inversion was able to reduce uncertainty of the total flux within a pixel by up to 97.7%". This discussion should be added in the same section, to understand if such an error reduction is realistic.*

**Response**: We would like to thank the referee for their support of the work, and we are happy to add the additional clarification to the discussion.

Previously this section was: "The inversion was able to substantially improve the agreement between the prior and posterior modelled concentrations, with posterior modelled concentrations tracking most of the local events observed in the measurements. The most notable corrections to the pixel-level fluxes by the inversion were made to those with the largest industrial point sources, to pixels located on Robben Island where activities unaccounted for in the inventory were taking place, and to the areas dominated by NEE fluxes and located relatively close to the measurement sites. This evidence suggests that the inversion framework used here has had some success in capturing information regarding the $CO_2$ fluxes in the CT domain, and has applied reasonable corrections to the sources considered.

The inversion was able to reduce uncertainty of the total flux within a pixel by up to 97.7%, and was able to reduce the uncertainty in the total weekly flux over the whole domain by up to 50.5%. The largest innovation to a fossil fuel flux was applied to a pixel which contained an important point source in the domain - a crude oil refinery. This facility can process up to 100,000 barrels of crude oil per day. Unlike most industrial sources in the area which would be expected to have fairly consistent activity from day to day, activity at the crude oil refinery would depend on oil supply and on the global oil prices. During the period of March 2012 to June 2013, the global monthly oil price deviated between $117.29 in March 2012 and $90.73 in June 2012, ending on $99.74 in June 2013 (World Bank commodity prices). In addition, the consumption of liquid petroleum gas and heavy furnace oils may have decreased during this period (City of Cape Town, 2015). As this is a source with a large amount of expected variability, it is unlikely that the inversion method with distant measurement sites will be able to adequately estimate the flux in this pixel. In order for this to occur, the measurement site would have to consistently view the source during periods of both high and low activity in order to provide an unbiased estimate. An inversion could be used to estimate this particular source if a ring of instruments were placed around the site in order to capture information from the site at all times, regardless of the prevailing wind direction, such as suggested for the Otway $CO_2$CRC carbon capture project (Cook, 2012).

Compared with the fossil fuel emissions, relative innovations to the NEE fluxes were much larger, due to the large uncertainty prescribed to these fluxes. The largest innovations were made to natural areas near the CBD of CT, as well as to agricultural regions within the domain, particularly those close to the measurement sites. The prior estimates are dependent on the CABLE land-atmosphere interaction model and, although driven by the CCAM regional climate model, which in turn was driven by reliable reanalysed observations of the climate from NCEP, is still under ongoing development for use over South Africa. There is a great deal of uncertainty in its ability to simulate fluxes over the fynbos biome, as there is for most dynamic vegetation models (Moncrieff et al., 2015). In general, the inversion tended to increase the NEE fluxes so that the total flux was less negative compared with the priors, indicating that the amount of productivity estimated by CABLE may be overestimated."

We have updated this discussion to include an assessment of how realistic the posterior fluxes are. This large percentage uncertainty reduction is mainly due to the large prior uncertainty prescribed to NEE fluxes, but does not translate to very dramatic innovations to the prior fluxes themselves. We did calculate the degrees of freedom for the signal (DFS) (Rodgers, 2000), and this varied between 38% and 65% of the maximum DFS for a monthly inversion of 1344. Therefore, the posterior fluxes were constrained both by the priors and by the observations. We calculated the DFS separately for the different groups

of fluxes, and noted that the DFS were very small for the night-time NEE fluxes, therefore the observations were not providing much information to constrain these estimates. This section has been modified as follows:

[revised manuscript text omitted]